# GPEX, A Framework For Interpreting Artificial Neural Networks

**Amir Akbarnejad**
Department of Computing Science
University of Alberta
Edmonton, AB, Canada
ah8@ualberta.ca

**Gilbert Bigras**
Department of Laboratory Medicine and Pathology
University of Alberta
Edmonton, AB, Canada
Gilbert.Bigras@albertaprecisionlabs.ca

**Nilanjan Ray**
Department of Computing Science
University of Alberta
Edmonton, AB, Canada
nray1@ualberta.ca

## Abstract

The analogy between Gaussian processes (GPs) and deep artificial neural networks (ANNs) has received a lot of interest, and has shown promise to unbox the blackbox of deep ANNs. Existing theoretical works put strict assumptions on the ANN (e.g. requiring all intermediate layers to be wide, or using specific activation functions). Accommodating those theoretical assumptions is hard in recent deep architectures, and those theoretical conditions need refinement as new deep architectures emerge. In this paper we derive an evidence lower-bound that encourages the GP's posterior to match the ANN's output without any requirement on the ANN. Using our method we find out that on 5 datasets, only a subset of those theoretical assumptions are sufficient. Indeed, in our experiments we used a normal ResNet-18 or feed-forward backbone with a single wide layer in the end. One limitation of training GPs is the lack of scalability with respect to the number of inducing points. We use novel computational techniques that allow us to train GPs with hundreds of thousands of inducing points and with GPU acceleration. As shown in our experiments, doing so has been essential to get a close match between the GPs and the ANNs on 5 datasets. We implement our method as a publicly available tool called GPEX: https://github.com/amirakbarnejad/gpex. On 5 datasets (4 image datasets, and 1 biological dataset) and ANNs with 2 types of functionality (classifier or attention-mechanism) we were able to find GPs whose outputs closely match those of the corresponding ANNs. After matching the GPs to the ANNs, we used the GPs' kernel functions to explain the ANNs' decisions. We provide more than 200 explanations (around 30 explanations in the paper and the rest in the supplementary) which are highly interpretable by humans and show the ability of the obtained GPs to unbox the ANNs' decisions.

## 1 Introduction

Artificial neural networks (ANNs) are widely adopted in machine learning. Despite their benefits, ANNs are known to be black-box to humans, meaning that their inner mechanism for making predictions is not necessarily interpretable/explainable to humans. ANN's black-box property impedes its deployment in safety-critical applications like medical imaging or autonomous driving, and makes them hard-to-troubleshoot for machine learning researchers.

37th Conference on Neural Information Processing Systems (NeurIPS 2023).

Attribution-based explanation methods like LIME[31], SHAP[21] and most gradient-based explanation methods like DeepLIFT [5] presume a linear surrogate model. Given a test instance $x_{test}$, this simpler surrogate model is encouraged to have the same output "locally" around $x_{test}$. Because of this "local assumptions", explanations from these methods might be unreliable, and can be easily manipulated by an adversary model [11][28]. Moreover, these models may produce discordant explanations for a fixed model and test instance [16].

Considering Gaussian processes (GPs) [26] as the explainer model is beneficial, because: 1. Gaussian processes are highly interpretable. 2. Researchers have long known that GP's posterior has the potential to match an ANN's output "globally". More precisely, given an ANN and some requirements on it [24][8], there might exist a GP whose posterior matches the ANN's output all over the input-space $\mathbb{X}$ (as opposed to the local explanation models for which the match happens only locally around a test instance $x_{test} \in \mathbb{X}$). Not many explainer models can globally match the ANN's output. Among gradient-based methods, with the best of our knowledge only Integrated Gradients [33] has a weak sense of ANN's global behaviour over the input space. Having some conditions on an ANN, representer point selection [37] finds a "globally faithfull" explainer model that, similar to GPs, works with a kernel function. As we will elaborate upon in Sec. 4.3 and Sec. S6 of the supplementary, the GP's kernel that we find in this paper is superior due to a technical point in the formulation of representer point selection [37]. All in all, using GPs to explain ANNs is quite promising and has advantages over other approaches to explain ANNs.

The contributions of this paper are as follows:

- Theoretical results on ANN-GP analogy impose some restrictions on ANNs under which the ANN will be equivalent to a GP. These conditions are too restrictive for recently used deep architectures. Moreover, those theoretical conditions need refinement as new deep architectures emerge. In this paper we derive an ELBO for training GPs which encourages GP's posterior to match ANN's output. Our formulation and method doesn't impose any restriction on the ANN and the method used to train it.

- Using our method, we empirically show that on 5 datasets (4 image datasets, and 1 biological dataset) and ANNs with 2 types of functionality (classifier or attention-mechanism) the ANN needs to fulfill only a subset of those theoretical conditions. Indeed, in our experiments we used a normal ResNet-18 or feed-forward backbone with a single wide layer in the end.

- Scalability is a major issue in training GPs. To address this issue, we adopted computational techniques recently used for fast spectral clustering [13] as well as a novel method to learn the GPs using mini-batches of inducing points and training instances. These computational techniques allow us to train GPs with hundreds of thousands of inducing points. According to our analysis, increasing the inducing points has been essential to get a good match between the trained GPs and ANNs. Indeed, without many inducing points GPs posterior cannot be a complex function (a function with many ups and downs [36]) and fails to match ANNs' output.

- With the best of our knowledge, our work is the first method that performs knowledge distillation between GPs and ANNs.

- We implement our method as a public python library called GPEX (Gaussian Processes for EXplaining ANNs). GPEX takes in an arbitrary PyTorch module, and replaces any ANN submodule of choice by GPs. Our package makes use of GPU-accelaration, and enables effortless application of GPs without getting users involved in details of the inference procedure. GPEX can be used by machine learning researchers to interpret/troubleshoot their artificial neural networks. Moreover, GPEX can be used by researchers working on the theoretical side of ANN-GP analogy to empirically test their hypotheses.

## 2 Proposed Method

### 2.1 Notation

In this article the function $g(.)$ always denotes an ANN. The kernel of a Gaussian process is denoted by the double-input function $\mathcal{K}(.,.)$. We assume the kernel similarity between two instances $x_i$ and $x_j$ is equal to $f(x_i)^T f(x_j)$, where $f(.)$ maps the input-space to the kernel space. In this paper $u$ (resp. $v$) denotes a vector in the kernel-space (resp. the posterior mean) of a GP. In some sense $u$ and $v$ denote

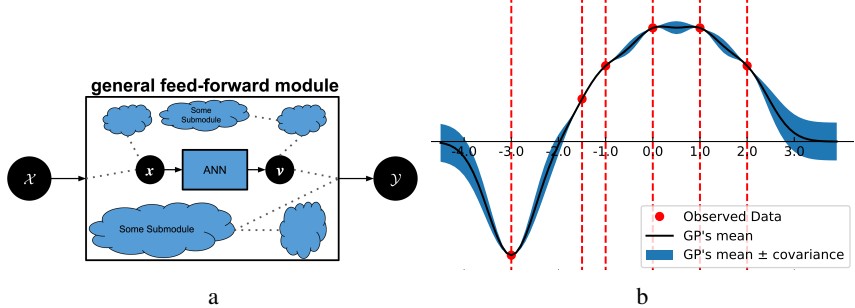

Figure 1: a) A general feed-forward pipeline, with an ANN sub-module to be explained by GPEX. b) Typical behaviour of Guassian process posterior given a set of observed values.

the input and the output of a GP, respectively. We have that: $\mathcal{K}(\boldsymbol{x_i}, \boldsymbol{x_j}) = f(\boldsymbol{x_i})^T f(\boldsymbol{x_j}) = \boldsymbol{u_i}^T \boldsymbol{u_j}$. The number of GPs is equal to the number of the outputs from the ANN. In other words, we consider one GP per scalar output-head from the ANN. We use index $\ell$ to specify the $\ell$-th GP as follows: $\mathcal{K}_\ell(\boldsymbol{x_i}, \boldsymbol{x_j}) = f_\ell(\boldsymbol{x_i})^T f_\ell(\boldsymbol{x_j}) = \boldsymbol{u_i}^{(\ell)T} \boldsymbol{u_j}^{(\ell)}$. We parameterize the $\ell$-th GP by a set of $M$ inducing points $\{(\tilde{\boldsymbol{u}}_m^{(\ell)}, \tilde{v}_m^{(\ell)})\}_{m=1}^M$. The tilde in $(\tilde{\boldsymbol{u}}_m^{(\ell)}, \tilde{v}_m^{(\ell)})$ indicates that $\tilde{\boldsymbol{u}}$ is one of the $M$ inducing points in the kernel space. However, $\boldsymbol{u}$ (without tilde) can be an arbitrary point in the continuous kernel-space.

## 2.2 The Proposed Framework

To make our framework as general as possible, we consider a general feed-forward pipeline that contains an ANN as a submodule. In Fig. 1a the bigger square illustrates the general module. The input-output of the general pipeline are denoted in Fig. 1a by $\mathcal{X}$ and $\mathcal{Y}$. The general pipeline has at least one ANN submodule to be explained by GPEX. Fig. 1a illustrates this ANN by the small blue rectangle within the general pipeline. The input-output of the ANN are denoted in Fig. 1a by $\boldsymbol{x}$ and $\boldsymbol{v}$. Note that $\mathcal{X}$ and $\mathcal{Y}$ can be anything, including without any limitation, a set of vectors, labels, and meta-information. However, input-output of the ANN (i.e. $\boldsymbol{x}$ and $\boldsymbol{y}$) are required to be in tensor format. The exact requirements are provided in the online documentation for GPEX. Moreover, the general module can have other arbitrary submodules, which are depicted by the blue clouds. The relations between the submodules, as illustrated by the dotted-lines in Fig. 1a, can also be quite general. Our probabilistic formulation only needs access to the conditional distributions $p(\boldsymbol{x}|\mathcal{X})$ and $p(\mathcal{Y}|\boldsymbol{x}, \mathcal{X})$. Similarly, the proposed GPEX is completely agnostic about the general pipeline and it only requires the ANN's input-output to be in the tensor format. Given a PyTorch module, the proposed GPEX tool automatically grabs the distributions $p(\boldsymbol{x}|\mathcal{X})$ and $p(\mathcal{Y}|\boldsymbol{x}, \mathcal{X})$ from the main module it is given.

The inducing points $\{\tilde{\boldsymbol{u}}_m^{(\ell)}, \tilde{v}_m^\ell\}_{m=1}^M$ parameterize the $\ell$-th GP. Note that $\tilde{\boldsymbol{u}}_m^{(\ell)} = f_\ell(\tilde{\boldsymbol{x}}_m)$. A feature point like $\boldsymbol{x}$ is first mapped to the kernel-space as $\boldsymbol{u}^{(\ell)} = f_\ell(\boldsymbol{x})$. Note that the kernel functions $\{f_\ell(.)\}_{\ell=1}^L$ are implemented as separate neural networks, or for the sake of efficiency as a single neural network backbone with $L$ different heads. Afterwards, the GP's posterior on $\boldsymbol{x}$ depends on the kernel similarities between $\boldsymbol{u}^{(\ell)}$ and the inducing points $\{\tilde{\boldsymbol{u}}_m^{(\ell)}\}_{m=1}^M$. More precisely, the posterior of the $\ell$-th GP on $\boldsymbol{x}$ is a random variable $v^{(\ell)}$ whose distribution is as follows [26]:

$$p(v^{(\ell)}|\boldsymbol{u}^{(\ell)}, \tilde{\mathbf{u}}_{1:M}^{(\ell)}, \tilde{v}_{1:M}^{(\ell)}) = \mathcal{N}\Big(v^{(\ell)} \; ; \; \mu_v(\boldsymbol{u}^{(\ell)}, \tilde{\mathbf{u}}_{1:M}^{(\ell)}, \tilde{v}_{1:M}^{(\ell)}), \; cov_v(\boldsymbol{u}^{(\ell)}, \tilde{\mathbf{u}}_{1:M}^{(\ell)}, \tilde{v}_{1:M}^{(\ell)})\Big), \quad (1)$$

where $\mu_v(., ., .)$ and $cov_v(., ., .)$ are the mean and covariance of a GP's posterior computed as:

$$\mu_v(\boldsymbol{u}^{(\ell)}, \tilde{\mathbf{u}}_{1:M}^{(\ell)}, \tilde{v}_{1:M}^{(\ell)}) = \mathcal{K}(\boldsymbol{u}^{(\ell)}, \tilde{\mathbf{u}}_{1:M}^{(\ell)}) \big[\mathcal{K}(\tilde{\mathbf{u}}_{1:M}^{(\ell)}, \tilde{\mathbf{u}}_{1:M}^{(\ell)}) + \sigma_{gp}^2 \mathbf{I}_{M \times M}\big]^{-1} \tilde{v}_{1:M}^{(\ell)} \quad (2)$$

and

$$cov_v(\boldsymbol{u}^{(\ell)}, \tilde{\mathbf{u}}_{1:M}^{(\ell)}, \tilde{v}_{1:M}^{(\ell)}) = \mathcal{K}(\boldsymbol{u}^{(\ell)}, \boldsymbol{u}^{(\ell)}) -$$
$$\mathcal{K}(\boldsymbol{u}^{(\ell)}, \tilde{\mathbf{u}}_{1:M}^{(\ell)}) \times \big[\mathcal{K}(\tilde{\mathbf{u}}_{1:M}^{(\ell)}, \tilde{\mathbf{u}}_{1:M}^{(\ell)}) + \sigma_{gp}^2 \mathbf{I}_{M \times M}\big]^{-1} \mathcal{K}(\tilde{\mathbf{u}}_{1:M}^{(\ell)}, \boldsymbol{u}^{(\ell)}). \quad (3)$$

As the variables $\{v_m^{(\ell)}\}_{m=1}^M$ and $v$ are latent or hidden, we train the model parameters by optimizing a variational lower-bound. We consider the following variational distributions:

$$q_1(v^{(\ell)}|\boldsymbol{x}) = \mathcal{N}\big(v^{(\ell)} \; ; \; g_\ell(\boldsymbol{x}) \, , \, \sigma_g^2\big), \qquad q_2\big(\tilde{v}_m^{(\ell)}\big) = \mathcal{N}\big(\tilde{v}_m^{(\ell)} \; ; \varphi_m^{(\ell)}, \sigma_\varphi^2\big). \quad (4)$$

In Eq. 4, the function $g_\ell(.)$ is the $\ell$-th output from the ANN. Note that as the set of hidden variables $\{\tilde{v}_m^{(\ell)}\}_{m=1}^M$ is finite, we have parameterized their variational distribution by a finite set of numbers $\{\varphi_m^{(\ell)}\}_{m=1}^M$. However, as the variables $\boldsymbol{x}$ can vary arbitrarily in the feature space, the variable $\boldsymbol{u}^{(\ell)}$ varies arbitrarily in the kernel space. Therefore, the set of values $v^{(\ell)}$ may be infinite. Accordingly, the variational distribution for $v^{(\ell)}$ is conditioned on $\boldsymbol{x}$ and is parameterized by the ANN $g(.)$.

## 2.3 The Derived Evidence Lower-Bound (ELBO)

Due to space limitation, the derivation of the lower-bound is moved to Sec. S1 of the supplementary material. In this section we only introduce the derived ELBO and discuss how it relates the GP, the ANN and the training cost of the main module in an intuitive way. The ELBO terms containing the GP parameters (i.e. the parameters of the kernel function $f(.)$) is denoted by $\mathcal{L}_{gp}$. According to Eq. S9 of the supplementary material $\mathcal{L}_{gp}$ is as follows:

$$
\begin{aligned}
\mathcal{L}_{gp} = &-\frac{1}{2} \, \mathbb{E}_{\sim q}\Big[ \sum_{\ell=1}^L \frac{(\mu_v(\boldsymbol{u}^{(\ell)}, \tilde{\mathbf{u}}_{1:M}^{(\ell)}, \tilde{v}_{1:M}^{(\ell)}) - g_\ell(\boldsymbol{x}))^2 + \sigma_g^2}{cov_v(\boldsymbol{u}^{(\ell)}, \tilde{\mathbf{u}}_{1:M}^{(\ell)}, \tilde{v}_{1:M}^{(\ell)})} \Big] \\
&-\frac{1}{2} \, \mathbb{E}_{\sim q}\Big[ \sum_{\ell=1}^L \log\big(\frac{cov_v(\boldsymbol{u}^{(\ell)}, \tilde{\mathbf{u}}_{1:M}^{(\ell)}, \tilde{v}_{1:M}^{(\ell)})}{\sigma_g^2}\big) \Big] + (\text{const.}),
\end{aligned} \tag{5}
$$

where $q(.)$ is the variational distribution that factorizes to the $q_1(.)$ and $q_2(.)$ distributions defined in Eq. 4. In the first term of Eq. 5, the numerator encourages the GP and the ANN to have the same output. More precisely, for a feature point $\boldsymbol{x}$ we can compute the corresponding point in the kernel space as $\boldsymbol{u}^{(\ell)} = f_\ell(\boldsymbol{x})$ and then compute the GP's posterior mean based on kernel similarities between $\boldsymbol{u}$ and the inducing points to get the GP's mean $\mu_v$. In Eq. 5 the GP's mean $\mu_v$ is encouraged to match the ANN's output $g_\ell(\boldsymbol{x})$. In Eq. 5, because of the denominator of the first term, the ANN-GP similarity is not encouraged uniformly over the feature-space. Wherever the GP's uncertainty is low, the term $cov_v(\boldsymbol{u}^{(\ell)}, \tilde{\mathbf{u}}_{1:M}^{(\ell)}, \tilde{v}_{1:M}^{(\ell)})$ in the denominator becomes small. Therefore, the GP's mean is highly encouraged to match the ANN's output. On the other hand, in regions where the GP's uncertainty is high, the GP-ANN analogy is less encouraged. This formulation is quite intuitive according to the behaviour of Gaussian processes. Fig. 1b illustrates the posterior of a GP with radial-basis kernel for a given set of observations. In regions like $[3, \infty)$ and $(-\infty, -4]$ there are no nearby observed data. Therefore, in these regions the GP is highly uncertain and the blue uncertainty margin is thick in such regions. Intuitively, our derived ELBO in Eq. 5 encourages the GP-ANN analogy only when GP's uncertainty is low and gives less importance to regions similar to $[3, \infty)$ and $(-\infty, -4]$ in Fig. 1b. Note that this formulation makes no difference for the ANN as ANNs are known to be global approximators. However, this formulation makes a difference when training the GP, because the GP is not required to match the ANN in regions where there are no similar training instances. The ELBO terms containing the ANN parameters is denoted by $\mathcal{L}_{ann}$. According to Sec. S1.2 of the supplementary material, $\mathcal{L}_{ann}$ is as follows:

$$
\mathcal{L}_{ann} = -\frac{1}{2} \, \mathbb{E}_{\sim q}\Big[ \sum_{\ell=1}^L \frac{(\mu_v(\boldsymbol{u}^{(\ell)}, \tilde{\mathbf{u}}_{1:M}^{(\ell)}, \tilde{v}_{1:M}^{(\ell)}) - g_\ell(\boldsymbol{x}))^2}{cov_v(\boldsymbol{u}^{(\ell)}, \tilde{\mathbf{u}}_{1:M}^{(\ell)}, \tilde{v}_{1:M}^{(\ell)})} \Big] + \mathbb{E}_{\sim q}\big[ \log p(\mathcal{Y}|\boldsymbol{y}, \mathcal{X}) \big]. \tag{6}
$$

In the above objective the first term encourages the ANN to have the same output as the GP. Similar to the objective of Eq. 5, the denominator of the first term gives more weight to ANN-GP analogy when GP's uncertainty is low. In the right-hand-side of Eq. 6, the second term is the likelihood of the pipeline's output(s), i.e. $\mathcal{Y}$ in Fig. 1a. This term can be, e.g., the cross-entropy loss when $\mathcal{Y}$ contains class scores in a classification problem, or the mean-squared error when $\mathcal{Y}$ is the predicted value for a regression problem, or a combination of those costs in a multi-task setting.

## 3 Algorithm

We consider a separate Gaussian process for each output head of an ANN. In other words, given an ANN we have as many GPs as the number of the ANN's output heads. To explain an ANN, we find the explainer GPs by optimizing the objective in Eq. 5 w.r.t. to the kernel mappings $\{f_\ell(.)\}_{\ell=1}^L$. To do so, we need to have $\mu_v$ which in turn means we need to have all kernel-space representations

---

**Algorithm 1** Method Optim_KernMappings

---

**Input:** Input instance $\boldsymbol{x}$ and inducing instance $\tilde{\boldsymbol{x}}$, list of matrices $\mathbf{U}$, list of vectors $\mathbf{V}$.
**Output:** Kernel-space mappings $[f_1(.), ..., f_L(.)]$.

  *Initialisation* : $loss \leftarrow 0$.
1: $\mu$, $cov \leftarrow$ forward_GP($\mathbf{x}$, $\tilde{\boldsymbol{x}}$, $\mathbf{U}$, $\mathbf{V}$) //feed $\mathbf{x}$ to GPs, "forward_GP" is Alg.S1 in supplementary.
2: $\mu_{ann} \leftarrow g(\mathbf{x})$ //feed $\mathbf{x}$ to ANN.
3: **for** $\ell = 1$ to $L$ **do**
4:     $loss \leftarrow loss + \frac{(\mu[\ell] - \mu_{ann}[\ell])^2 + \sigma_g^2}{cov[\ell]} + \log(cov[\ell])$. //Eq.5.
5: **end for**
6: $\boldsymbol{\delta} \leftarrow \frac{\partial\ loss}{\partial\ params\big([f_1(.), ..., f_L(.)]\big)}$.//the gradient of loss.
7: $params\big([f_1, ..., f_L]\big) \leftarrow params\big([f_1, ..., f_L]\big) - lr \times \boldsymbol{\delta}$     //update the parameters.
8: $lr \leftarrow$ updated learning rate
9: **return** $[f_1(.), ..., f_L(.)]$

---

---

**Algorithm 2** Method Explain_ANN

---

**Input:** Training dataset $ds\_train$, and the inducing dataset $ds\_inducing$.
**Output:** Updated kernel-space mappings $[f_1(.), ..., f_L(.)]$, and the other GP parameters $\mathbf{U}$ and $\mathbf{V}$.

  *Initialisation* : $\mathbf{U}, \mathbf{V} \leftarrow$ Init_GPparams($ds\_inducing$) //Alg.S3 in supplementary.
1: **for** $iter = 1$ to $max\_iter$ **do**
2:     $\boldsymbol{x} \leftarrow randselect(ds\_train)$.
3:     $\tilde{\boldsymbol{x}} \leftarrow randselect(ds\_inducing)$
4:     $[f_1(.), ..., f_L(.)] \leftarrow$ Optim_KernMapings($\boldsymbol{x}, \tilde{\boldsymbol{x}}, \mathbf{U}, \mathbf{V}$).
5:     $\tilde{\boldsymbol{x}} \leftarrow randselect(ds\_inducing)$.
6:     **for** $\ell = 1$ to $L$ **do**
7:         //update kernel-space representations.
8:         $U[\ell][\tilde{\boldsymbol{x}}.index] \leftarrow f_\ell(\tilde{\boldsymbol{x}})$
9:     **end for**
10: **end for**
11: **return** $[f_1(.), ..., f_L(.)]$, $\mathbf{U}$, $\mathbf{V}$

---

$\{\tilde{\boldsymbol{u}}_m^{(\ell)}\}_{m=1}^M$. However, it is computationally prohibitive to feed thousands of inducing instances to the kernel mappings as $\tilde{\boldsymbol{u}}_m^{(\ell)} = f_\ell(\tilde{\boldsymbol{x}_m})$ for $m \in \{1, 2, ..., M\}$ in each gradient-descent iteration. On the other hand, as the kernel-space mappings $\{f_\ell(.)\}_{\ell=1}^L$ keep changing during training, we need to somehow track how the inducing points $\{\tilde{\boldsymbol{u}}_m^{(\ell)}\}_{m=1}^M$ change during training. To this end, we put the kernel-space representations of the inducing points in matrices denoted by $\mathbf{U}$. During training, these matrices are repeatedly updated by feeding mini-batches of inducing instances to the kernel-mappings.

Alg. 2 optimizes the objective of Eq. 5 w.r.t. the kernel mappings $\{f_\ell(.)\}_{\ell=1}^L$. First, a single training instance $\boldsymbol{x}$ and a single inducing point $\tilde{\boldsymbol{x}}$ are selected (line 2-3). Afterwards, the procedure of Alg. 1 is called to update the kernel mappings (line 4 of Alg. 2). To update the kernel-mappings, the GP posterior is computed via the matrices $\mathbf{U}$ (line 1 of Alg.1). The "forward_GP" procedure (called in line 1 of Alg. 1) is provided in Alg. S1 of the supplementary, and uses the matrices $\mathbf{U}$ to compute GP's posterior. Only the rows of $\mathbf{U}$ that correspond to the selected inducing point $\tilde{\boldsymbol{x}}$ are computed using the kernel-mappings, so that the gradient w.r.t. the kernel-mappings can be computed in the backward pass (lines 5-6 of Alg. S1 in the supplementary). Finally, the matrices $\mathbf{U}$ are updated (lines 6-8 of Alg. 2). Due to the lack of space, the routines "forward_GP" and "Init_GPparams" and more details are moved to Sec. S2 of the supplementary. Of course instead of a single training/inducing instance, we used a mini-batch of multiple training/inducing instances.

One difficulty of training GPs is the matrix inversion of Eqs.2 and 3, which has $\mathcal{O}(M^3)$ complexity using standard matrix inversion methods. To address this issue, we adopted computational techniques recently used for fast spectral clustering [13]. Let $\boldsymbol{A}$ be an arbitrary $M \times D$ matrix where $M >> D$. Moreover, let $\boldsymbol{b}$ be a $M$-dimensional vector and let $\sigma$ be a scalar. The computational techniques [13] allow us to efficiently compute: $(\mathbf{A}\mathbf{A}^T + \sigma^2 \boldsymbol{I}_{M \times M})^{-1} \boldsymbol{b}$. (Note the similar terms in the right hand side of Eqs. 2 and 3.) The idea is that $\mathbf{A}\mathbf{A}^T$ is of rank $D$. Therefore, from linear algebra it follows

that $(\mathbf{A}\mathbf{A}^T + \sigma^2 \boldsymbol{I}_{M \times M})^{-1}$ has $M - D$ eigen-values all of which are equal to $\sigma^{-2}$. Therefore, in the space of those eigen-vectors, the transformation on $\boldsymbol{b}$ is simply a scaling by $\sigma^{-2}$. The details and more computational techniques are provided in Sec. S2.1 of the supplementary. These computational techniques allow us to efficiently compute the GP-posterior for hundreds of thousands of inducing points in each gradient descent iteration.

**A note on the used datasets in Alg. 2:** According to our analysis of Sec. S7 in the supplementary material, "ds_inducing" should be as large as possible so the GP posteriors can be flexible enough to match the ANNs. Therefore a good practice is to include all training instances (without data augmentation) in "ds_inducing". By doing so, the following issue arises. An instance from "ds_train" like $\boldsymbol{x}$ is an augmented version of an inducing instance $\tilde{\boldsymbol{x}}$. Because $\boldsymbol{x}$ and $\tilde{\boldsymbol{x}}$ are close, their kernel-space representations $f(\boldsymbol{x})$ and $f(\tilde{\boldsymbol{x}})$ also become close regardless of parameters of $f(.)$. Consequently, regardless of $f(.)$, GP's posterior mean will be roughly equal for both $\boldsymbol{x}$ and $\tilde{\boldsymbol{x}}$. Indeed, in this case Alg. 2 fails to find the kernel mappings $\{f_\ell(.)\}_{\ell=1}^L$. To avoid this issue, we sample $\boldsymbol{x}$ in line 2 of Alg. 2 as follows: $\boldsymbol{x}_1$ and $\boldsymbol{x}_2$ are randomly selected from "ds_train", and $\alpha \sim uniform(-1, 2)$, and $\boldsymbol{x} = \alpha\boldsymbol{x}_1 + (1 - \alpha)\boldsymbol{x}_2$. The rest of Alg. 2 after line 2 is run as before.

## 4 Experiments

We conducted several experiments on four publicly available datasets: MNIST [9], Cifar10 [19], Kather [15], and DogsWolves [34]. For MNIST [9] and Cifar10 [19] we used the standard split to training and test sets provided by the datasets. For Kather [15] and DogsWolves [34] we randomly selected 70% and 80% of instances as our training set. The exact parameter settings for running Alg. 2 are elaborated upon in Sec. S5 of the supplementary. We trained the ANNs as usual rather than using Eq. 6, because our proposed GPEX should be applicable to ANNs which are trained as usual.

### 4.1 Measuring Faithfulness of GPs to ANNs

We trained a separate convolutional neural network (CNN) on each dataset to perform the classification task. For MNIST [9], Cifar10 [19], and Kather [15] we used a ResNet-18 [12] backbone followed by some fully connected layers. DogsWolves [34] is a relatively small dataset, and very deep architectures like ResNet [12] overfit to training set. Therefore, we used a convolutional backbone which is suggested in the dataset website [34]. For all datasets, we set the width (i.e. the number of neurons) of the second last fully-connected layer to 1024. Because according to theoretical results on GP-ANN analogy, the second last layer of ANN should be wide. We used an implementation of ResNet [12] which is publicly available online [2]. We trained the pipelines for 20, 200, 20, and 20 epochs on MNIST [9], Cifar10 [19], Kather [15], and DogsWolves [34], respectively. For Cifar10 [19], we used the exact optimizer suggested by [2]. For other datasets we used an Adam [17] optimizer with a learning-rate of 0.0001. The test accuracies of the models are equal to 99.56%, 95.43%, 96.80%, and 80.50% on MNIST [9], Cifar10 [19], Kather [15], and DogsWolves [34], respectively. We also applied our proposed GPEX to a state-of-the-art cell-embedding method called scArches [20]. We ran a tutorial notebook [32] and applied GPEX to the decoder whose job is to predict expression of some genes given scArches [20] cell embeddings. More details are provided in our public github repository (repository link is provided in page 1).

We explained each classifier ANN using our proposed GPEX framework (i.e. Alg.2). As discussed in Sec. 3, given an ANN we have as many kernel-spaces (and as many GPs) as the number of ANN's output heads. The exact parameter settings and practical considerations for training the GPs is elaborated upon in Sec. S5 of the supplementary material. To measure the faithfulness of GPs to ANNs, we compute the Pearson correlation coefficient for each ANN head and the mean of the corresponding GP posterior on unseen test instances. The results are provided in Fig. 3. In Fig. 3, the first five groups of bars (i.e. the groups labeled as Cifar10 (classifier), MNIST (classifier), Kather (classifier), DogsWolves (classifier), and scArches (classifier)) correspond to applying the proposed GPEX to the five classifier ANNs trained on the four datasets and scArches embeddings. According to Fig. 3, our trained GPs almost perfectly match the corresponding ANNs. Only for DogsWovles [34], as illustrated by the 4-th bar group in Fig. 3, the correlation coefficients are lower compared to other datasets. We hypothesize that this is because the DogsWolves dataset [34] has very few images. GP posterior mean can be changed only by moving the inducing points in the kernel-space.

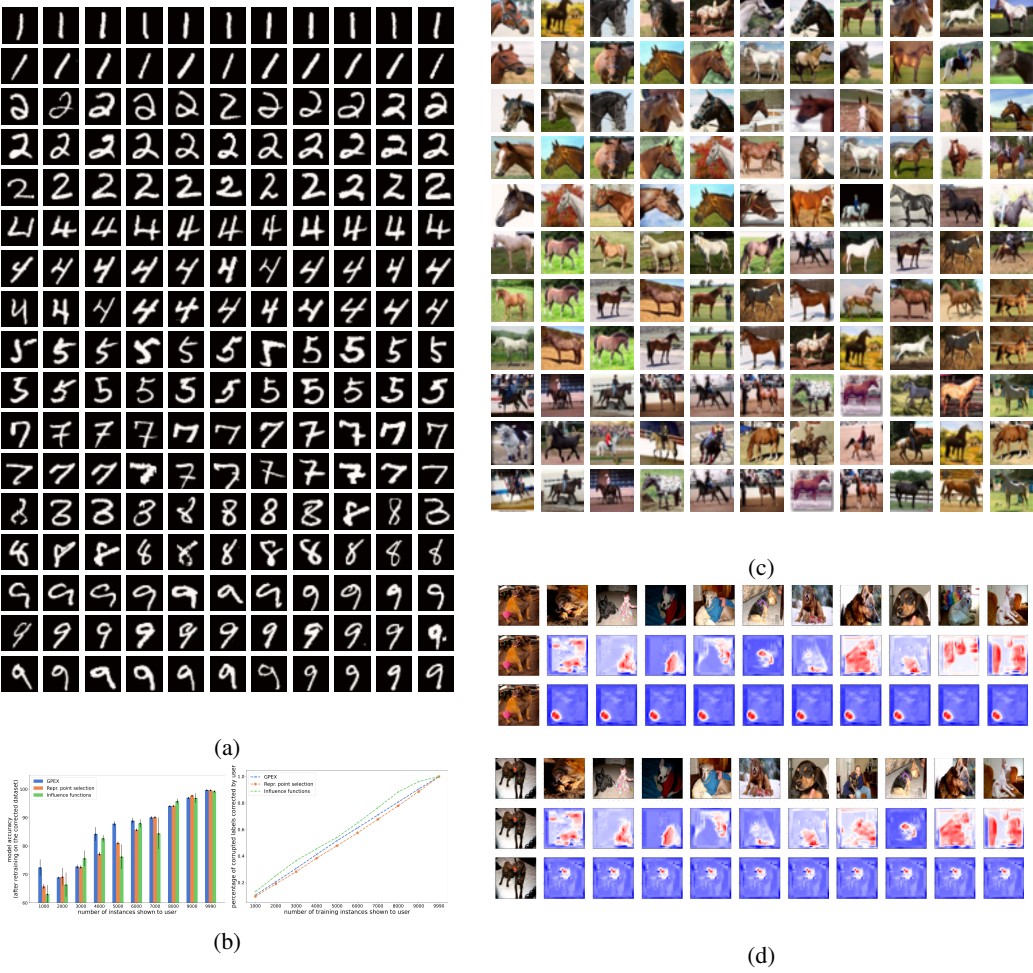

Figure 2: (a,c,d) Sample explanations for MNIST, Cifar10, and DogsWolves. In each row a test instance is shown in the first column, and the 10 nearest neighbours (in the kernel-space of the GP that corresponds to the output-head with maximum value at the test instance) is shown in columns 2-11. (b) Evaluating our proposed method, representer point selection [37], and influence functions [18] in dataset debugging task.

Therefore, when very few inducing points are available GP posterior mean is less flexible [36]. This is consistent with our parameter analysis in Sec. S7 of supplementary material.

In Fig. 1a we discussed that GPEX is not only able to explain a classifier ANN, but it can explain any ANN which is a subcomponent of any feed-forward pipeline. To evaluate this ability, we trained three classifiers with an attention mechanism [22]. Each classifier has two ResNet-18 [12] backbones: one extracts a volumetric map containing deep features, and the other produces a spatial attention mask. For each attention backbone, we set the width of the second last layer to 1024, followed by a linear layer and sigmoid activation. We applied our proposed GPEX (i.e. Alg.2) to each classifier, but this time the ANN to be explained (i.e. the box called "ANN" in Fig. 1a) is set to be the attention submodule. Note that each attention backbone produces a spatial attention mask of size $h$ by $w$. We think of each attention backbone as an ANN which has $h \times w$ output heads. We trained three classifier pipelines with attention mechanism on Cifar10 [19], MNIST [9], and Kather [15] with the same training procedure as previous part. In Fig. 3, 6-th, 7-th, and 8-th bar groups show the correlation coefficients between the attention backbones and the corresponding GPs on unseen test instances. According to Fig. 3, our proposed GPEX is able find GPs which are faithful to attention subcomponents of the classifier pipelines. Note that we didn't include all attention heads, because

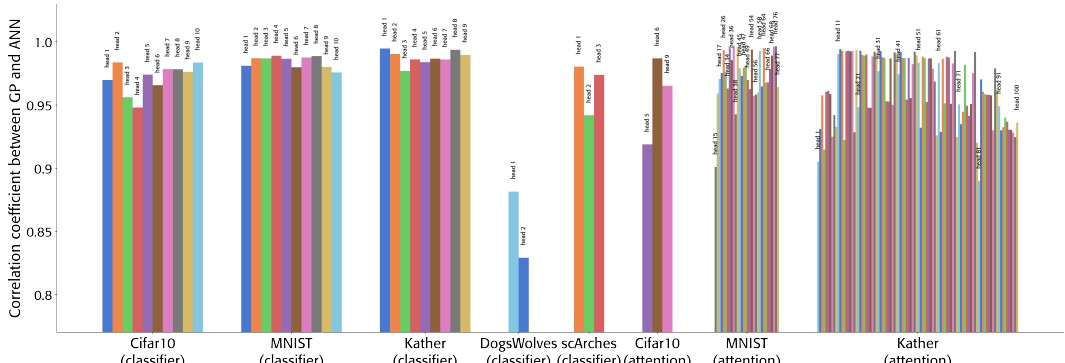

Figure 3: Faithfulness of GPs to ANNs measured by the Pearson correlation coefficient.

some pixels in attention masks are always off. In Sec. S3 of the supplementary material we have included more information and insights about the faithfulness of GPs to ANNs.

## 4.2 Explaining ANNs' Decisions

In Sec. 4.1 we trained four CNN classifiers on Cifar10 [19], MNIST [9], Kather [15], and DogsWolves [34], respectively. Afterwards, we applied our proposed explanation method to each CNN classifier. In this section, we are going to explain the decisions made by the classifiers via the obtained GPs found by Alg. 2. We explain the decision made for a test instance like $x_{test}$ as follows. We consider the GP and the kernel-space that correspond to the ANN's head with maximum value (i.e. the ANN's head that relates to the predicted label). Consequently, among the instances in the inducing dataset, we find the 10 closest instances to $x_{test}$, like $\{x_{i1}, x_{i2}, ..., x_{i10}\}$. Intuitively the ANN has labeled $x_{test}$ in that way because it has found $x_{test}$ to be similar to $\{x_{i1}, x_{i2}, ..., x_{i10}\}$.

For MNIST digit classification, some test instances and nearest neighbours in training set are shown in Fig. 2a. In this figure each row corresponds to a test instance. The first column depicts the test instance itself and columns 2 to 11 depict the 10 nearest neighbours. For example, in Fig. 2a the image in row3-col1 depicts a test instance $x_{test}$ and the images in row3, cols2-11 depict the nearest neighbours $\{x_{i1}, x_{i2}, ..., x_{i10}\}$. According to rows 1 and 2 of Fig. 2a, the classifier has labeled the two images as digit 1 because it has found 1 digits with similar inclinations in the training set (in Fig. 2a in row 1 all digits are vertical but in row 2 all digits are inclined). We see the model has also taken the inclination into account for the test instances of rows 7, 8, 15, 16, and 17 of Fig. 2a. In Fig. 2a, according to rows 3, 4, and 5 the test instances are classified as digit 2 because 2 digits with similar styles are found in the training set. We see the model has also taken the style into account for the test instances of rows 6, 7, 8, 9, 10, 12, 13, 14, 15, 16, and 17 of Fig. 2a. For instance, the test instance in row 6 of Fig. 2a is a 4 digit with a short stand and the two nearest neighbours are alike. Or for the test instances in rows 13, 14, and 15 of Fig. 2a the test instances have incomplete circles in the same way as their nearest neighbours. More explanations are provided in the supplementary material in Sec. S4.

Fig. 2c illustrates some sample explanations for Cifar10 [19]. Like before, each row corresponds to a test instance, the first column depicts the test instance itself and columns 2 to 11 depict the 10 nearest neighbours. In Fig. 2c, the test instances of rows 1, 2, 3, 4, and 5 are captured from horses' heads from closeby, and the nearest neighbours are alike. However, in rows 6, 7, 8, 9, 10, and 11 of Fig. 2c the test images are taken from faraway and the found similar training images are also taken from faraway. Intuitively, as the classifier is not aware of 3D geometry, it finds training images which are captured from the same distance. In rows 9, 10, and 11 of Fig. 2c, we see that the testing images contain riders. Similarly, the nearest neighbours also tend to have riders. Therefore, in rows 9, 10, and 11 of Fig. 2c the model has made use of the riders or other context information to classify the test instances as horse. More explanations are provided in the supplementary material in Sec. S4.

Besides finding the nearest neighbours, we provide CAM-like [38] explanations as to why $x_{test}$ and an instance like $x_{ij}, 1 \leq j \leq 10$ are considered similar by the model (according to the procedure of Sec. S2.2 in the supplementary material). Fig.2d illustrates some sample explanations for DogsWolves [34] dataset. In row 1 of Fig. 2d, the first column depicts the test instance itself and columns 2 to 11 depict the 10 nearest neighbours. The second and third rows highlight the pixels

that contribute the most to the similarities. The second and third rows highlight the pixels of $x_{test}$ and $\{x_{i1}, x_{i2}, ..., x_{i10}\}$ respectively. According to row 3 of Fig. 2d, the pink object next to the dog's leg has contributed the most to the similarities. According to row 2 of Fig. 2d regions like the baby in column 3, the dog colar or costume in columns 4, 5, and 6, human finger in column 9, and the background in columns 10 and 11 have contributed the most to their similarity to the test instance. These are patterns that usually happen for dogs images. Indeed, since the training set has been small (1600 images), to detect dogs the model is making use of patterns that normally exist in indoor scenes and do not normally appear in wolves images. We see a similar pattern for the test instance in row 6 of Fig. 2d and also several explanations in Sec.S4 of the supplementary.

### 4.3 Comparing GPEX to Representer-Point Selection and Influence Functions

In Sec. S6 of the supplementary material we qualitatively compare GPEX explanations to those of representer point selection [37]. According to the experiments and detailed discussions of Sec. S6 in the supplementary, the GP's kernel that we find in this paper is superior due to a technical point in the formulation of representer point selection [37]. Besides the analysis of Sec. S6, we compared our proposed GPEX with representer point selection [37] and influence functions [18] in dataset debugging task. In these experiments we only selected images from Cifar10 [19] that are labeled as either automobile or horse. To corrupt the labels, we randomly selected 45% of training instances and changed their labels. Afterwards, we trained a classifier CNN with ResNet18 [12] backbone with the same training procedure explained in Sec. 4.1. In dataset debugging task, training instances are shown to a user in some order. After seeing an instance, the user checks the label of the instance and corrects it if needed. One can use explanation methods to bring the corrupted labels to the user's attention more quickly. Given an explanation method, we repeatedly select a test instance which is misclassified by the model. Afterwards, we show to the user the closest training instance (of course among the training instances which are not yet shown to the user). We repeat this process for test instances in turn until all training instances are shown to the user. We compared our proposed GPEX to representer point selection [37] and influence functions [18] in dataset debugging task. We used an implementation of influence functions [18] based on LiSSA [4] with 10 steps for each instance. The implementation is publicly available [1]. For representer point selection [37] we used the implementation by authors which is publicly available [3]. The result is shown in Fig. 2b. According to the plot on the left in Fig. 2b, when correcting the dataset by GPEX, the model accuracy becomes close to 90% after showing about 4000 instances to user. But when using representer point selection [37] or influence functions [18], this happens when the user has seen about 7000 training instances. With noisy labels model training becomes unstable. Therefore, in the plot on the left of Fig. 2b we repeat the training 5 times and we report the standard errors by the lines in top of the bars. According to the plot on the right of Fig. 2b, after showing a fixed number of training instances to the user, when using the proposed GPEX more corrupted labels are shown to the user. Indeed, GPEX brings the corrupted labels to the user's attention quicker than representer point selection [37] does. Interestingly, according to the plot in the right hand side of Fig. 2b influence functions [18] is quicker at spotting incorrect labels, but the instances found by our proposed method are more effective in increasing the accuracy quicker.

## 5 Related Work

The first theoretical connection between ANNs and GPs was that under some conditions, a random single-layer neural network converges to the mean of a Gaussian process [23] as the width of that single layer goes to infinity. This connection was later proven for ANNs with many layers [8], and for ANNs trained with gradient descent [14]. The theoretical requirements are usually too restrictive. For example, [8] requires all intermediate layers to be wide and also requires the dataset to be countable (so data-augmentations like color-jitter are not allowed). Or [24] requires the ANN to be trained with MSE loss and requires all intermediate layers to be wide. In this paper we do not presume any conditions on the ANN and simply distill knowledge from a neural network to some GPs. Of note, those theoretical conditions may facilitate knowledge distillation and improve the Pearson correlation coefficient between the ANNs and the GPs obtained by our method.

Scalability is a major issue when training GPs, and including a few inducing points may limit the flexibility of GP's posterior [36]. Here we review some previous methods to tackle the computational challenges of training GPs. SV-DKL [27] derives a lower-bound for training a GP with a deep

kernel. In this method, a grid of inducing points are considered in the kernel-space (like the vectors $\{(\tilde{\boldsymbol{u}}_m^{(\ell)}, \tilde{v}_m^{(\ell)})\}_{m=1}^M$ with the notation of this paper). Afterwards, each input instance is firstly mapped to the kernel-space and the output is computed based on similarities to the grid points in the kernel-space. Since the GP posterior is computed via the grid points, SV-DKL [27] is scalable. But unfortunately the number of grid points cannot be increased to above 1000 even for Cifar10 [19] and with a RTX 3090 GPU. Therefore, this may limit the flexiblity of the GP's posterior [36].

A more recent framework called GPytorch [29] provides GPU acceleration. However, its computational complexity is quadratic in number of inducing points. Other approaches to improve scalability of GPs include: considering structured kernel matrices [7], kernel interpolation [35], and imposing grid-structure on including points [27]. Stack of Gaussian processes are shown to be connected to ANNs [10][30][24]. By stacking kernels, GP kernels work on intermediate representations and therefore are not necessarily interpretable to humans. But in our method the GPs' kernels work directly on the input-space itself. Knowledge distillation (KD) is closely related to this work. With the best of our knowledge and according to the authors, [6] is the first work that applies KD to GPs. But the distinction of our work is that we distill knowledge from ANN to GP, as opposed to the self-distillation of [6] that distills knowledge from a GP to another GP.

**Limitations and Outlook**: In this work we used Eq. 5 to distill knowledge from ANN to GP. One may use Eq. 6 to distill knowledge from GP to ANN in order to, e.g., transfer GP's good generalization to the ANN. Our method scales very well, and Alg. 2 runs without memory/computational issues even on imagenet with more than 1M inducing points (i.e. images) and Resnet-18 [12] when a few output-heads are selected, but on imagenet we failed to match the GPs to ANN in a 2-3 day runtime. The issue is that the $\mathbf{U}$ matrices have to be updated very often (the update of line 8 of Alg. 2) so that GPs' kernels are updated according to an accurate estimate of kernel-space representations. Otherwise the convergence may not happen especially for millions of inducing points and a small batch-size (as required for, e.g., CNNs). We used control-variate [25], but one may use more advanced heuristics [35] to achieve convergence for datasets like imagenet and with a reasonable computation time. In this paper we analyzed the effect of number of inducing points, the width of the second last layer, and the number of epochs for which the ANN is trained. One can use the proposed tool to answer other questions, like, is the GP kernel required to have more parameters than the ANN itself? May it so happen that a test instance is equally close to hundreds of training instances thereby limiting a human's ability to understand ANNs decision? Is the uncertainty provided by the GP correlated to the understandability of the explanations to humans or to the ANN's failures?

# 6 Acknowledgements

The experiments of this paper were enabled in part by the Digital Research Alliance of Canada. This work was supported in part by the NSERC Discovery Grant.

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
