# Supplementary Material for
# GPEX, A Framework For Interpreting Artificial Neural Networks

Amir Akbarnejad, Gilbert Bigras, Nilanjan Ray

---◆---

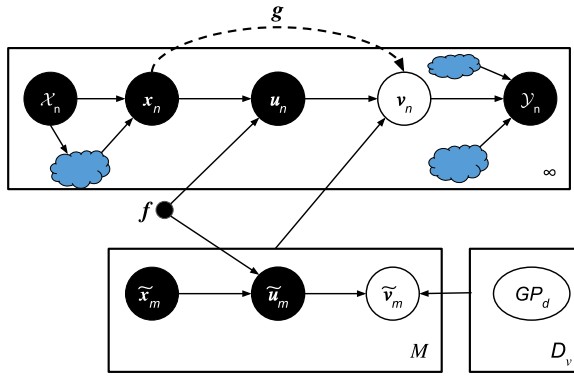

Fig. S1: The proposed framework as a probabilistic graphical model.

## S1  DERIVING THE VARIATIONAL LOWER-BOUND

In this section we derive the variational lower-bound introduced in Sec.2.3 of the main article. We firstly introduce Lemmas 1 and 2 as they appear in our derivations.

*Lemma 1.* The KL-divergence between two normal distributions $\mathcal{N}_1(.\ ;\ \boldsymbol{\mu}_1, \boldsymbol{\Sigma}_1)$ and $\mathcal{N}_2(.\ ;\ \boldsymbol{\mu}_2, \boldsymbol{\Sigma}_2)$ can be computed as follows:

$$KL\Big(\mathcal{N}_1 \,\|\, \mathcal{N}_2\Big) = \frac{1}{2}\Big( \log(\frac{|\boldsymbol{\Sigma}_2|}{|\boldsymbol{\Sigma}_1|}) \ - \ D \ + \ trace\{\boldsymbol{\Sigma}_2^{-1}\boldsymbol{\Sigma}_1\}$$
$$+\ (\boldsymbol{\mu}_2 - \boldsymbol{\mu}_1)^T\boldsymbol{\Sigma}_2^{-1}(\boldsymbol{\mu}_2 - \boldsymbol{\mu}_1)\Big).\blacksquare$$

$$(S1)$$

*Lemma 2.* Let $p_1$ and $p_2$ be two normal distributions:

$$p_1(x) = \mathcal{N}\big(x\ ;\ \mu_1, \sigma_1^2\big),$$
$$p_2(x) = \mathcal{N}\big(x\ ;\ \mu_2, \sigma_2^2\big).$$

We have that

$$\mathbb{E}_{x \sim p_2}\big[ \log\ p_1(x\ ;\ \mu_1, \sigma_1^2)\big] =$$
$$-\frac{(\mu_1 - \mu_2)^2 + \sigma_2^2}{2\sigma_1^2} - \frac{1}{2}\log(\sigma_1^2) - \frac{1}{2}\log(2\pi).\blacksquare$$

$$(S2)$$

Fig.S1 illustrates the framework as a probabilistic graphical model. A general feed-forward pipeline takes in a set of

input(s) $\mathcal{X}$ and produces a set of output(s) $\mathcal{Y}$. The general pipeline is required to have at least one ANN as a submodule. The ANN submodule is required to take in only one input $\boldsymbol{x}$ and to produce only one output $\boldsymbol{v}$, where $\boldsymbol{x}$ and $\boldsymbol{v}$ are tensors of arbitrary sizes. As illustrated in Fig.S1, the ANN's input $\boldsymbol{x}$ can depend arbitrarily on some other intermediate variables in the pipeline. This relation is modeled by the conditional distribution $p\big(\boldsymbol{x}_n|Parent(\boldsymbol{x}_n)\big)$ where $Parent(\boldsymbol{x}_n)$ is the set of all variables which are connected to $\boldsymbol{x}_n$. Similarly, as illustrated in Fig.S1 the pipeline's output $\mathcal{Y}$ can arbitrarily depend on some intermediate variables in the pipeline. This relation is modeled by the conditional distribution $p(\mathcal{Y}_n|Parent(\mathcal{Y}_n))$. In Fig.S1 the lower boxes are the inducing points and other variables that determine the GPs' posterior. More precisely, in Fig.S1 $\{\tilde{\boldsymbol{x}}_m\}_{m=1}^M$ are some inducing points (e.g. some training images). Vectors in the kernel space are denoted by $\tilde{\boldsymbol{u}}$ and $\boldsymbol{u}$. Moreover, the observed values are denoted by $v$ and $\tilde{v}$. Informally, $\boldsymbol{u}$ and $v$ denote the input/output of the GPs. When referring to one of the $M$ inducing points a "tilde" is used (as $(\tilde{\boldsymbol{u}}, \tilde{v})$), however $(\boldsymbol{u}, v)$ corresponds to a point that can be anywhere in the kernel-space.

The inducing instances $\{\tilde{\boldsymbol{x}}_m\}_{m=1}^M$ are mapped to the kernel-spaces by the kernel mappings $\{f_1(.), ..., f_L(.)\}$. In Fig.S1 the variables $\{\tilde{\boldsymbol{u}}_m\}_{m=1}^M$ are the kernel-space representations of the inducing points $\{\tilde{\boldsymbol{x}}_m\}_{m=1}^M$. Moreover, $\{\tilde{\boldsymbol{v}}_m\}_{m=1}^M$ are the GP's output values at the inducing points. Given an instance $\boldsymbol{x}_n$, it is firstly fed to the kernel mappings $\{f_1(.), ..., f_L(.)\}$ and the kernel-space representations $\boldsymbol{u}_n$ are obtained. Afterwards, the GPs' outputs on $\boldsymbol{u}_n$ depend on $\boldsymbol{u}_n$ as well as all other inducing points because the inducing points actually determine the GPs' posterior on all kernel-space points including $\boldsymbol{u}_n$. Therefore, in Fig.S1 the variable $\boldsymbol{v}_n$ is not only connected to $\boldsymbol{u}_n$ but it is also connected to the box at the bottom (i.e. all inducing points and other variables associated with them).

As usual, the variational lower-bound is equal to

$$\mathcal{L} = \mathbb{E}_{\sim q}\big[ \log p(\text{all variables})\big] - \mathbb{E}_{\sim q}\big[ \log q(\text{hidden variables})\big].$$
$$(S3)$$

The likelihood of all variables in Eq.S3 factorizes as the product of conditional distributions of each variable given

its parents. Therefore

$$p(\text{all variables}) = \prod_{variable\ t} p(t|Parent(t)). \quad \text{(S4)}$$

In Eq.S4 only some conditional distributions appear in our derivations which are discussed at the following.

- The variable $\boldsymbol{x}_n$: the ANN's input $\boldsymbol{x}_n$ can depend arbitrarily on some other intermediate variables in the pipeline. In our derivations we leave this conditional distribution as $p(\boldsymbol{x}_n|Parent(\boldsymbol{x}_n))$.
- The variable $\boldsymbol{u}_n$: Given a training instance $\boldsymbol{x}_n$, the kernel-space representations $\boldsymbol{u}_n$ are deterministically obtained by feeding the instance to the kernel-mappings $[f_1(.), ..., f_L(.)]$.
- The variable $\boldsymbol{v}_n$: The ANN's output is required to depend only on the input, so

$$p(\boldsymbol{v}_n|Parent(\boldsymbol{v}_n)) = p(\boldsymbol{v}_n|\boldsymbol{u}_n, \boldsymbol{x}_n, \{\tilde{\boldsymbol{x}}_m, \tilde{\boldsymbol{u}}_m, \tilde{v}_m\}_{m=1}^{M}).$$
$$\text{(S5)}$$

  The above distribution is actually the GPs' posterior at $\boldsymbol{u}_n$ (i.e. the normal distribution of Eq.1 of the main article).
- The variable $\tilde{\boldsymbol{x}}_m$: the inducing point $\tilde{\boldsymbol{x}}_m$ can depend arbitrarily on some other intermediate variables in the pipeline. In our derivations we leave this conditional distribution as $p(\tilde{\boldsymbol{x}}_m|Parent(\tilde{\boldsymbol{x}}_m))$.
- The variable $\tilde{\boldsymbol{u}}_m$: Given an inducing point $\tilde{\boldsymbol{x}}_m$, the kernel-space representations $\tilde{\boldsymbol{u}}_m$ are deterministically obtained by feeding the inducing point $\tilde{\boldsymbol{x}}_m$ to the kernel-mappings $[f_1(.), ..., f_L(.)]$.
- The variables $\hat{\boldsymbol{v}}_m$: Given the kernel-space representations $\{\tilde{\boldsymbol{u}}_m^{(\ell)}\}_{m=1}^{M}$, the variables $\{\tilde{v}_1^{(\ell)}, ..., \tilde{v}_M^{(\ell)}\}$ follow a $M$-dimensional Gaussian distribution with zero mean and a covariance matrix determined by the GP prior covariance among the variables $\{\tilde{\boldsymbol{u}}_m^{(\ell)}\}_{m=1}^{M}$.
- The variable $\mathcal{Y}_n$: the pipeline's output $\mathcal{Y}$ can arbitrarily depend on some intermediate variables in the pipeline. In our derivations we leave this conditional distribution as $p(\mathcal{Y}_n|Parent(\mathcal{Y}_n))$.

According to Eq.S4, the likelihood of all variables factorizes as

$$p(\text{all variables}) = \prod_{variable\ t} p(t|Parent(t))$$
$$= \big(\prod_n p(\boldsymbol{x}_n|Parent(\boldsymbol{x}_n))\big) \times \big(\prod_n p(\boldsymbol{u}_n|\boldsymbol{x}_n)\big) \times$$
$$\big(\prod_n \prod_\ell p(v_n^{(\ell)}|\boldsymbol{u}_n, \boldsymbol{x}_n, \{\tilde{\boldsymbol{x}}_m, \tilde{\boldsymbol{u}}_m, \tilde{v}_m\}_{m=1}^{M})\big) \times$$
$$\big(\prod_m p(\tilde{\boldsymbol{x}}_m|Parent(\tilde{\boldsymbol{x}}_m))\big) \times \big(\prod_m p(\tilde{\boldsymbol{u}}_m|\tilde{\boldsymbol{x}}_m)\big) \times$$
$$\big(\prod_\ell p(\tilde{v}_{1:M}^{(\ell)}|\boldsymbol{0}, \mathcal{K}_{prior}(\tilde{\boldsymbol{u}}_{1:M}^{(\ell)}, \tilde{\boldsymbol{u}}_{1:M}^{(\ell)}))\big) \times$$
$$\big(\prod_n p(\mathcal{Y}_n|Parent(\mathcal{Y}_n))\big) \times \big(\prod_{other\ vars\ t} p(t|Parent(t))\big).$$
$$\text{(S6)}$$

Now we derive the lower-bound $\mathcal{L}$ with respect to each parameter separately.

## S1.1 Deriving the Lower-bound With Respect to the Kernel-mappings

In the right-hand-side of Eq.S6 only the following terms are dependant on the kernel-mappings $[f_1(.), ..., f_L(.)]$:

$$\big[\prod_m p(\tilde{\boldsymbol{u}}_m|\tilde{\boldsymbol{x}}_m) \times \prod_\ell p(\tilde{v}_m^{(\ell)}|\boldsymbol{0}, \mathcal{K}_{prior}(\tilde{\boldsymbol{u}}_{1:M}^{(\ell)}, \tilde{\boldsymbol{u}}_{1:M}^{(\ell)}))\big] \times$$
$$\big[\prod_n p(\boldsymbol{u}_n|\boldsymbol{x}_n) \times \prod_\ell p(v_n^{(\ell)}|\boldsymbol{u}_n, \boldsymbol{x}_n, \{\tilde{\boldsymbol{x}}_m, \tilde{\boldsymbol{u}}_m, \tilde{v}_m\}_{m=1}^{M})\big].$$
$$\text{(S7)}$$

Note that in the above equation the terms $p(\tilde{\boldsymbol{u}}_m|\tilde{\boldsymbol{x}}_m)$ and $p(\boldsymbol{u}_n|\boldsymbol{x}_n)$ are equal to 1 because $\tilde{\boldsymbol{u}}_m$ and $\boldsymbol{u}_n$ are deterministically obtained from $\tilde{\boldsymbol{x}}_m$ and $\boldsymbol{x}_n$. Therefore, in Eq.S3 the terms containing the kernel mappings $[f_1(.), ..., f_L(.)]$ are as follows:

$$\mathcal{L}_f = \mathbb{E}_{\sim q}\big[\sum_\ell \log p(v^{(\ell)}|\boldsymbol{u}, \boldsymbol{x}, \{\tilde{\boldsymbol{x}}_m, \tilde{\boldsymbol{u}}_m, \tilde{v}_m\}_{m=1}^{M})\big] +$$
$$\sum_\ell \mathbb{E}_{\sim q}\big[\log p(\tilde{v}_{1:M}^{(\ell)}|\boldsymbol{0}, \mathcal{K}_{prior}(\tilde{\boldsymbol{u}}_{1:M}^{(\ell)}, \tilde{\boldsymbol{u}}_{1:M}^{(\ell)}))\big] -$$
$$\sum_\ell \mathbb{E}_{\sim q}\big[\log q_2(\tilde{v}_{1:M}^{(\ell)})\big]$$
$$= \mathbb{E}_{\sim q}\big[\sum_\ell \log p(v^{(\ell)}|\boldsymbol{u}, \boldsymbol{x}, \{\tilde{\boldsymbol{x}}_m, \tilde{\boldsymbol{u}}_m, \tilde{v}_m\}_{m=1}^{M})\big] -$$
$$\sum_{\ell=1} \mathbb{E}_{\sim q}\big[KL\big(q_2(\tilde{v}_{1:M}^{(\ell)}) \ || \ p(\tilde{v}_{1:M}^{(\ell)}|\boldsymbol{0}, \mathcal{K}_{prior}(\tilde{\boldsymbol{u}}_{1:M}^{(\ell)}, \tilde{\boldsymbol{u}}_{1:M}^{(\ell)}))\big)\big].$$
$$\text{(S8)}$$

We simplify the two terms on the right-hand-side of Eq.S8. The first term is the expected log-likelihood of a Gaussian distribution (i.e. the conditional log-likelihood of $\tilde{v}^\ell$ as in Eq.1 of the main article). Also the variational distribution $q(.)$ is Gaussian. Therefore, we can use Lemma.2 to simplify the first term:

$$\mathbb{E}_{\sim q}\big[\sum_{\ell=1}^{L} \log p(v^{(\ell)}|\boldsymbol{u}, \boldsymbol{x}, \{\tilde{\boldsymbol{x}}_m, \tilde{\boldsymbol{u}}_m, \tilde{v}_m\}_{m=1}^{M})\big] =$$
$$\sum_{\ell=1}^{L} \mathbb{E}_{\sim q}\big[\log p(v^{(\ell)}|\boldsymbol{u}, \boldsymbol{x}, \{\tilde{\boldsymbol{x}}_m, \tilde{\boldsymbol{u}}_m, \tilde{v}_m\}_{m=1}^{M})\big] =$$
$$\sum_{\ell=1}^{L}\Big[ -\frac{\big(\mu_v(\boldsymbol{u}^{(\ell)}, \tilde{\mathbf{u}}_{1:M}^{(\ell)}, \tilde{v}_{1:M}^{(\ell)}) - g_\ell(\boldsymbol{x})\big)^2 + \sigma_g^2}{cov_v(\boldsymbol{u}^{(\ell)}, \tilde{\mathbf{u}}_{1:M}^{(\ell)}, \tilde{v}_{1:M}^{(\ell)})}$$
$$-\frac{1}{2}\log\big(cov_v(\boldsymbol{u}^{(\ell)}, \tilde{\mathbf{u}}_{1:M}^{(\ell)}, \tilde{v}_{1:M}^{(\ell)})\big)$$
$$-\frac{1}{2}\log(2\pi)\Big]. \quad \text{(S9)}$$

Note that the two terms of Eq.S9 are the two terms which were presented and discussed in Eq.5 of the main article.

Now we simplify the KL-term on the right-hand-side of

Eq.S8. According to Lemma.1 we have that

$$
\begin{aligned}
KL\Big(q_2(\tilde{\boldsymbol{v}}_{1:M}^{(\ell)}) \;\|\; p(\tilde{\boldsymbol{v}}_{1:M}^{(\ell)}|\mathbf{0}, \mathcal{K}_{prior}(\tilde{\boldsymbol{u}}_{1:M}^{(\ell)}, \tilde{\boldsymbol{u}}_{1:M}^{(\ell)}))\Big) = \\
+ 0.5\big(\log(\frac{\sigma_{gp}^2}{\sigma_\varphi^2})\big) \\
- 0.5M \\
+ \frac{\sigma_\varphi^2}{\sigma_{gp}^2} \\
+ \frac{\boldsymbol{\varphi}_{1:M}^{(\ell)T}\boldsymbol{\varphi}_{1:M}^{(\ell)}}{\sigma_{gp}^2},
\end{aligned}
$$
(S10)

where $\varphi$ are the variational parameters of $q_2(.)$ as in Eq.4 of the main article. Therefore, the KL-term of Eq.S8 is a constant with respect to the kernel mappings $[f_1(.), ..., f_L(.)]$ and can be discarded. All in all, the lower-bound for optimizing the kernel-mappings is equal to the right-hand-side of Eq.S9 which was introduced and discussed in Sec.2.3. of the main article.

## S1.2 Deriving the Lower-bound With Respect to the ANN Parameters

According to Eq.4 of the main article, in our formulation the ANN's parameters appear as some variational parameters. Therefore, the likelihood of all variables (Eq.S6) does not generally depend on the ANN's parameters. But according to the general ELBO formulation in Eq.S3 the ELBO $\mathcal{L}$ depends on ANN's parameters, because when computing the expectation the variables are drawn from the variational distribution $q(.)$. We estimated the ELBO of Eq.S3 by the average over few samples. More precisely, given a training instance $\boldsymbol{x}$, we firstly computed the kernel-space representations as:

$$
\boldsymbol{u}^{(\ell)} = f_\ell(\boldsymbol{x}), \quad 1 \le \ell \le L.
$$
(S11)

Afterwards, we used the reparametrization trick for Eq.1 of the main article to draw a sample for $\boldsymbol{v}^{(\ell)}$ as follows:

$$
\begin{aligned}
z_{q2}^{(\ell)} &\sim \mathcal{N}(0,1), \\
v^{(\ell)} &\sim \mu_v(\boldsymbol{u}^{(\ell)}, \tilde{\mathbf{u}}_{1:M}^{(\ell)}, \tilde{v}_{1:M}^{(\ell)}) + z_{q2}^{(\ell)} cov_v(\boldsymbol{u}^{(\ell)}, \tilde{\mathbf{u}}_{1:M}^{(\ell)}, \tilde{v}_{1:M}^{(\ell)}),
\end{aligned}
$$
(S12)

where $\mu_v(.,.,.)$ and $cov_v(.,.,.)$ are defined in Eqs.2 and 3 of the main article. Moreover, we continue the forward pass of the original pipeline to get a sample $\mathcal{Y}$. Having drawn $\boldsymbol{x}$, $\boldsymbol{u}$, $v$, and $\mathcal{Y}$ from the variational distribution, we estimate the ELBO of Eq.S3 by these samples.

$$
\begin{aligned}
\mathcal{L} = \\
\mathbb{E}_{\sim q}\big[\log p(\text{all variables})\big] - \mathbb{E}_{\sim q}\big[\log q(\text{hidden variables})\big] \\
\approx \log p(\text{all variables})\Big|_{\boldsymbol{x},\boldsymbol{u},v,\mathcal{Y}} - \sum_m^M \sum_\ell^L \mathbb{E}_{\sim q_2}\big[\log q_2(\tilde{v}_m^{(\ell)})\big]
\end{aligned}
$$
(S13)

In the above equation, the second term on the right-hand-side is the entropy of a normal distribution and it only depends on the variance of the $q_2$ distribution. As we let

the variance of $q_2$ be fixed ($\sigma_g^2$ in Eq.4 of the main article), the second term is a constant. Therefore,

$$
\mathcal{L} \approx \log p(\text{all variables})\Big|_{\boldsymbol{x},\boldsymbol{u},v,\mathcal{Y}}.
$$
(S14)

Among the likelihood term on the right-hand-side of Eq.S6 the conditional distribution of all variables before $\boldsymbol{u}_n$ (e.g. $\boldsymbol{x}_n$ and $\mathcal{X}_n$) are independent of the ANN's parameters (i.e. the parameters of the function $g(.)$). On the other hand, for all variables that appear after $\boldsymbol{u}_n$, the conditional distribution depends on the ANN's parameters. Indeed, according to Eq.S14

$$
\begin{aligned}
\mathcal{L}_{ann} \approx \big[\sum_{\ell=1}^L \log p(v^{(\ell)}|\boldsymbol{u}, \boldsymbol{x}, \{\tilde{\boldsymbol{x}}_m, \tilde{\boldsymbol{u}}_m, \tilde{v}_m\}_{m=1}^M)\big]\Big|_{\boldsymbol{x},\boldsymbol{v}} + \\
\log p(\mathcal{Y}|Parent(\mathcal{Y}))\Big|_{\boldsymbol{x},\boldsymbol{v},\mathcal{Y}} + \\
\big(\sum_{\text{other vars after } \boldsymbol{u}_n} \log p(t|Parent(t))\big)\Big|_{\boldsymbol{x},\boldsymbol{v},\mathcal{Y}}.
\end{aligned}
$$
(S15)

In the above equation, the first term on the right-hand-side is the log-likelihood of the normal distribution of Eq.1:

$$
\begin{aligned}
\log p(v^{(\ell)}|\boldsymbol{u}, \boldsymbol{x}, \{\tilde{\boldsymbol{x}}_m, \tilde{\boldsymbol{u}}_m, \tilde{v}_m\}_{m=1}^M) = \\
-\frac{1}{2}\big[\sum_{\ell=1}^L \frac{(\mu_v(\boldsymbol{u}^{(\ell)}, \tilde{\mathbf{u}}_{1:M}^{(\ell)}, \tilde{v}_{1:M}^{(\ell)}) - g_\ell(\boldsymbol{x}))^2}{cov_v(\boldsymbol{u}^{(\ell)}, \tilde{\mathbf{u}}_{1:M}^{(\ell)}, \tilde{v}_{1:M}^{(\ell)})}\big] \\
+ (\text{some terms independent from } g(.)).
\end{aligned}
$$
(S16)

In Eq.S15 the term $p(\mathcal{Y}|Parent(\mathcal{Y}))$ is the likelihood of the output(s) of the whole pipeline as illustrated by Fig.1a of the main article, given the ANN's output and all other intermediate variables on which the final output $\mathcal{Y}$ depends. This likelihood turns out to be equivalent to commonly-used losses like the cross-entropy loss or the mean-squared loss. Here we elaborate upon how this happens. Let the task be a classification, and let $\hat{\mathcal{Y}} \in \mathbb{R}^L$ be the pipeline's output. The final model prediction $\mathcal{Y}$ is done as follows:

$$
\mathcal{Y} \sim \mathcal{Categorical}(\hat{\mathcal{y}}_K, ..., \hat{\mathcal{y}}_K)
$$
(S17)

Therefore we have that

$$
p(\mathcal{Y}|Parent(\mathcal{Y})) = (\hat{\mathcal{y}}_1)^{I[\mathcal{Y}==1]} \times ... \times (\hat{\mathcal{y}}_K)^{I[\mathcal{Y}==K]},
$$
(S18)

where $I[.]$ is the indicator function. So, we have that

$$
\begin{aligned}
\log p(\mathcal{Y}|Parent(\mathcal{Y})) = \\
I[\mathcal{Y} == 1]\log(\hat{\mathcal{y}}_1) + ... + I[\mathcal{Y} == K]\log(\hat{\mathcal{y}}_K).
\end{aligned}
$$
(S19)

Therefore, when the pipeline is for classification, $\log p(\mathcal{Y}|\mathbf{v}, \ etc.)$ will be equal to the cross-entropy loss. This conclusion was introduced and discussed in Eq.6 of the main article. We can draw similar conclusions when the pipeline is for other tasks like regression, or even a combination of tasks.

In the general pipeline of Fig.S1, if all stages after $\boldsymbol{v}$ are deterministic (of course except the final stage which is probabilistic like Eq.S17), the third term on the right-hand-side of Eq.S15 becomes 1. Therefore, the right-hand-side of Eq.S15 is equal to Eq.6 of the main article. As we discussed in Sec.2.3 of the main article, $\mathcal{L}_{ann}$ has two terms: the first terms encourages the GP-ANN analogy and the second term seeks to lower the task-loss.

---

**Algorithm S1** Method Forward_GP

---

**Input:** Input instance $\boldsymbol{x}$ and inducing instance $\tilde{\boldsymbol{x}}$, list of matrices $\mathbf{U}$, list of vectors $\mathbf{V}$.
**Output:** List of GP posterior means $\mu$, and covariances $cov$.
    *Initialisation* : $\mu = list(L)$, $cov = list(L)$.
  1: **for** $\ell = 1$ to $L$ **do**
  2:     $u = f_\ell(\boldsymbol{x})$ //map $\boldsymbol{x}$ to the kernel space of the $\ell$-th GP.
  3:     $\mathbf{U}_\ell \leftarrow \mathbf{U}[L]$ //get the inducing points of the $\ell$-th GP.
  4:     $\mathbf{V}_\ell \leftarrow \mathbf{V}[L]$ //observed values at the inducing points.
  5:     **if** training **then**
  6:         $\mathbf{U}_\ell[\tilde{\boldsymbol{x}}.index] \leftarrow f_\ell(\tilde{\boldsymbol{x}})$ //to pass gradient w.r.t. $f_\ell(.)$
  7:     **end if**
  8:     $\mu[\ell] \leftarrow \boldsymbol{u}^T \mathbf{U}_\ell^T \left( \mathbf{U}_\ell \mathbf{U}_\ell^T + \sigma_{gp}^2 \mathbf{I} \right)^{-1} \mathbf{V}_\ell$.
  9:     $cov[\ell] \leftarrow \boldsymbol{u}^T \boldsymbol{u} - \boldsymbol{u}^T \mathbf{U}_\ell^T \left( \mathbf{U}_\ell \mathbf{U}_\ell^T + \sigma_{gp}^2 \mathbf{I} \right)^{-1} \mathbf{U}_\ell \boldsymbol{u}$.
10: **end for**
11: **return** $\mu$ and $cov$

---

**Algorithm S2** Method Optim_KernMappings

---

**Input:** Input instance $\boldsymbol{x}$ and inducing instance $\tilde{\boldsymbol{x}}$, list of matrices $\mathbf{U}$, list of vectors $\mathbf{V}$.
**Output:** Kernel-space mappings $[f_1(.), ..., f_L(.)]$.
    *Note the important modifiactions to Alg.S2 which are explained in Sec.S5.*
    *Initialisation* : $loss \leftarrow 0$.
  1: $\mu, cov \leftarrow$ forward_GP($\mathbf{x}, \tilde{\boldsymbol{x}}, \mathbf{U}, \mathbf{V}$) //feed $\mathbf{x}$ to GPs.
  2: $\mu_{ann} \leftarrow g(\mathbf{x})$ //feed $\mathbf{x}$ to ANN.
  3: **for** $\ell = 1$ to $L$ **do**
  4:     $loss \leftarrow loss + \frac{(\mu[\ell] - \mu_{ann}[\ell])^2 + \sigma_g^2}{cov[\ell]} + \log(cov[\ell])$. //Eq.5.
  5: **end for**
  6: $\boldsymbol{\delta} \leftarrow \frac{\partial \ loss}{\partial \ params\left([f_1(.), ..., f_L(.)]\right)}$. //the gradient of loss.
  7: $params\left([f_1, ..., f_L]\right) \leftarrow params\left([f_1, ..., f_L]\right) - lr \times \boldsymbol{\delta}$
    //update the parameters.
  8: $lr \leftarrow$ updated learning rate
  9: **return** $[f_1(.), ..., f_L(.)]$

---

### S1.3 Deriving the Lower-bound With Respect to $q_2(.)$ Parameters

In Eq.4 of the main article we considered the variational parameters $\{\varphi_m^{(\ell)}\}_{m=1}^M$ for the hidden variables $\{\tilde{v}_m^{(\ell)}\}_{m=1}^M$. The ELBO of Eq.S3 can be optimized with respect to $\{\varphi_m^{(\ell)}\}_{m=1}^M$ as well. But we noticed that optimizing $\{\varphi_m^{(\ell)}\}_{m=1}^M$ is computationally unstable. Therefore, we set $\{\varphi_m^{(\ell)}\}_{m=1}^M$ according to the following rule:

$$\varphi_m^{(\ell)} = g_\ell(\tilde{\boldsymbol{x}}_m), \qquad (S20)$$
$$1 \le m \le M, \ \ 1 \le \ell \le L.$$

We set $\{\varphi_m^{(\ell)}\}_{m=1}^M$ as above because $\tilde{v}_m^{(\ell)}$ is simply the $\ell$-th GP posterior mean at the inducing point $\tilde{\boldsymbol{x}}_m$. To make the GP's posterior mean equal to the ANN's output, $\tilde{v}_\ell^{(m)}$ should be equal to the ANN's (i.e. $g(.)$'s output at the $m$-th inducing point.

### S2 ALGORITHM DETAILS

During training, to compute GP's posterior we firstly need to have the $M$ inducing points $\{(\tilde{\boldsymbol{u}}_m^{(\ell)}, \tilde{v}_m^{(\ell)})\}_{m=1}^M$.

---

**Algorithm S3** Method Init_GPparams

---

**Input:** Dataset of inducing points $[\tilde{\boldsymbol{x}}_1, ..., \tilde{\boldsymbol{x}}_M]$.
**Output:** List of matrices $\mathbf{U}$, list of vectors $\mathbf{V}$.
    *Initialisation* : $\mathbf{U} = list(L)$, $\mathbf{V} = list(L)$.
  1: **for** $\ell = 1$ to $L$ **do**
  2:     $\mathbf{V}[\ell] \leftarrow [g(\tilde{\boldsymbol{x}}_1)[\ell], ..., g(\tilde{\boldsymbol{x}}_M)[\ell]]$.
  3: **end for**
  4: **for** $\ell = 1$ to $L$ **do**
  5:     $\mathbf{U}[\ell] \leftarrow [f_\ell(\tilde{\boldsymbol{x}}_1), ..., f_\ell(\tilde{\boldsymbol{x}}_M)]$.
  6: **end for**
  7: **return** $\mathbf{U}$ and $\mathbf{V}$

---

**Algorithm S4** Method Explain_ANN

---

**Input:** Training dataset $ds\_train$, and the inducing dataset $ds\_inducing$.
**Output:** Kernel-space mappings $[f_1(.), ..., f_L(.)]$, and the other GP parameters $\mathbf{U}$ and $\mathbf{V}$.
    *Initialisation* : $\mathbf{U}, \mathbf{V} \leftarrow$ Init_GPparams($ds\_inducing$).
  1: **for** $iter = 1$ to $max\_iter$ **do**
  2:     $\boldsymbol{x} \leftarrow randselect(ds\_train)$.
  3:     $\tilde{\boldsymbol{x}} \leftarrow randselect(ds\_inducing)$
  4:     $[f_1(.), ..., f_L(.)] \leftarrow$ Optim_KernMapings($\boldsymbol{x}, \tilde{\boldsymbol{x}}, \mathbf{U}, \mathbf{V}$).
  5:     $\tilde{\boldsymbol{x}} \leftarrow randselect(ds\_inducing)$.
  6:     **for** $\ell = 1$ to $L$ **do**
  7:         //update kernel-space representations.
  8:         $U[\ell][\tilde{\boldsymbol{x}}.index] \leftarrow f_\ell(\tilde{\boldsymbol{x}})$
  9:     **end for**
10: **end for**
11: **return** $[f_1(.), ..., f_L(.)]$, $\mathbf{U}$, $\mathbf{V}$

---

It is computationally prohibitive to repeatedly update $\{\tilde{\boldsymbol{u}}_m^{(\ell)}\}_{m=1}^M$ by mapping all $M$ instances to the kernel space as $\tilde{\boldsymbol{u}}_m^{(\ell)} = f_\ell(\tilde{\boldsymbol{x}}_m)$. On the other hand, as the kernel-space mappings $\{f_\ell(.)\}_{\ell=1}^L$ keep changing during training, we need to somehow track how the inducing points $\{\tilde{\boldsymbol{u}}_m^{(\ell)}\}_{m=1}^M$ change during training. To this end, we consider a matrix whose $m$-th row contains the value of $f_\ell(\tilde{\boldsymbol{x}}_m)$ at some point during training, where $\tilde{\boldsymbol{x}}_m$ is the $m$-th inducing instance. During training, we keep updating the rows of this matrix by feeding mini-batches of instances to $f_\ell(.)$. Note that we have as many GPs as the number of ANN's output heads. Therefore, for each GP we consider a separate matrix containing the representations of the inducing instances in the $\ell$-th kernel space. In Algs.S1, S2, S3, and S4 the variable $\mathbf{U}$ is a list containing all of the the aforementioned matrices. To explain a given ANN, we let the ANN to be fixed and we only train the GPs' parameters. This procedure is explained in Alg.S4. In each iteration, the kernel-mappings are updated according to the objective function of Eq.5 (line 3 of Alg.S2). Afterwards, to make the matrices in $\mathbf{U}$ track the changes in $[f_1(.), ..., f_L(.)]$, we map an inducing instance (or a mini-batch of inducing instances) to the kernel spaces, and we update the corresponding matrices and rows in $\mathbf{U}$ according to the newly obtained kernel-space representations. Updating $\mathbf{U}$ is done in line 8 of Alg.S4. The method in Alg.S1 computes the GPs' posterior means and covariances at any instance like $\boldsymbol{x}$, given the observed inducing points as specified by $\mathbf{U}$ and $\mathbf{V}$. Note that this

---

**Algorithm S5** Method Efficiently_Compute_AATinvb

---

**Input:** Matrix $\mathbf{A}$ of size $M \times D$, vector $\boldsymbol{b}$ of size $M \times 1$, and positive scalar $\sigma$.
**Output:** The vector $\mathbf{output} = (\mathbf{A}\mathbf{A}^T + \sigma^2\mathbf{I})^{-1}\boldsymbol{b}$.
1: $\tilde{\mathbf{E}}, \tilde{\boldsymbol{\lambda}} \leftarrow eigendecomp(\mathbf{A}^T\mathbf{A} + \sigma^2\mathbf{I})$.
2: $[\tilde{\boldsymbol{e}}_1, ..., \tilde{\boldsymbol{e}}_D] \leftarrow \tilde{\mathbf{E}}$
3: $[\tilde{\lambda}_1, ..., \tilde{\lambda}_D] \leftarrow \tilde{\boldsymbol{\lambda}}$
4: $[\boldsymbol{e}_1, ..., \boldsymbol{e}_D] \leftarrow [\mathbf{A}\tilde{\boldsymbol{e}}_1, ..., \mathbf{A}\tilde{\boldsymbol{e}}_D]$
5: $[\lambda_1, ..., \lambda_D] \leftarrow [\tilde{\lambda}_1, ..., \tilde{\lambda}_D]$
6: $\mathbf{E} \leftarrow [\boldsymbol{e}_1, ..., \boldsymbol{e}_D]$
7: $\boldsymbol{\Lambda} \leftarrow diagonal(\frac{1}{\lambda_1 + \sigma^2}, ..., \frac{1}{\lambda_D + \sigma^2})$
8: $\mathbf{output} \leftarrow \mathbf{E}\boldsymbol{\Lambda}\mathbf{E}^T\boldsymbol{b} + \frac{1}{\sigma^2}(\boldsymbol{b} - \mathbf{E}\mathbf{E}^T\boldsymbol{b})$ //according to //Eq.S21 in supplementary material
9: **return output**

---

method returns two outputs, because a GP's posterior at $\mathbf{x}$ is a normal distribution described by its mean and variance. In Alg.S1 lines 8 and 9 correspond to the equations of GP posterior (i.e. Eqs. 1 and 2 of the main article). The method in Alg.S1 is used both during training and testing. During training, this method is called whenever ANN's output and GP's posterior are encouraged to be close. During training, according to line 6 of Alg.S1 only the matrix row(s) corresponding to the fed inducing instance(s) are the result of mapping the inducing instance(s) via the kernel-mapping, and all other rows are kept fixed. Line 6 of Alg.S1 allows for computing the gradient of loss with respect to kernel-mappings $[f_1(.), ..., f_L(.)]$. During testing we call Alg.S1 to get the GP's posterior at a test instance like $\boldsymbol{x}_{test}$. Alg.S3 initializes the GP parameters $\mathbf{U}$ and $\mathbf{V}$. For the $\ell$-th GP, the vector $\mathbf{V}[\ell]$ is initialized to the $\ell$-th output head of the ANN at all inducing images. In Alg.S3, the vector $\mathbf{V}[\ell]$ is initialized in line 2. Moreover, for the $\ell$-th GP the matrix $\mathbf{U}[\ell]$ is initialized by mapping all inducing instances to the $\ell$-th kernel-space via the mapping $f_\ell(.)$. In Alg.S3 the matrix $\mathbf{U}[\ell]$ is initialized in line 5. The method in Alg.S3 is called only once before training the GP. For instance, when explaining an ANN in Alg.S4, the initialisation is done once at the beginning of the procedure.

## S2.1 Efficiently Computing Gaussian Process Posterior

Let $\boldsymbol{A}$ be an arbitrary $M \times D$ matrix where $M >> D$. Moreover, let $\boldsymbol{b}$ be a $M$-dimensional vector and let $\sigma$ be a scalar. The computational techniques [10] allow us to efficiently compute:

$$(\mathbf{A}\mathbf{A}^T + \sigma^2\boldsymbol{I}_{M \times M})^{-1}\boldsymbol{b}.$$

The idea is that $\mathbf{A}\mathbf{A}^T$ and therefore its inverse are of rank $D$. Therefore, $(\mathbf{A}\mathbf{A}^T)^{-1}$ has $D$ non-zero eigenvalues like $\{\lambda_1, ..., \lambda_D\}$ and the rest of its eigenvalues are zero. Let the corresponding eigenvectors be $\{\boldsymbol{e}_1, ..., \boldsymbol{e}_D\}$. To compute $(\mathbf{A}\mathbf{A}^T)^{-1}\boldsymbol{b}$ we can simply project $\boldsymbol{b}$ to the $D$-dimensional space of the eigenvectors. By doing so, we avoid the $\mathcal{O}(M^3)$ computational complexity. Let $\{\lambda_1, ..., \lambda_D\}$ be the non-zero eigenvalues of $\mathbf{A}\mathbf{A}^T$ and let $\{\boldsymbol{e}_1, ..., \boldsymbol{e}_D\}$ be the corresponding eigenvectors. From linear algebra, it follows that for $\mathbf{A}\mathbf{A}^T + \sigma^2\mathbf{I}_{M \times M}$ the eigenvalues and the eigenvectors are

$\{\lambda_1 + \sigma^2, ..., \lambda_D + \sigma^2, \sigma^2, ..., \sigma^2\}$ and $\{\boldsymbol{e}_1, ..., \boldsymbol{e}_D\}$, respectively. Note that $M - D$ eigenvectors are added all of which are equal to $\sigma^2$. Similarly, from linear algebra it follows that for the inverse of $\mathbf{A}\mathbf{A}^T + \sigma^2\mathbf{I}_{M \times M}$ the eigenvalues and eigenvectors are $\{\frac{1}{\lambda_1 + \sigma^2}, ..., \frac{1}{\lambda_D + \sigma^2}, \frac{1}{\sigma^2}, ..., \frac{1}{\sigma^2}\}$ and $\{\boldsymbol{e}_1, ..., \boldsymbol{e}_D, \boldsymbol{e}_{D+1}, ..., \boldsymbol{e}_M\}$ respectively. Note that although there are $M$ eigenvectors, only the first $D$ eigenvectors appear in our computations. More precisely, let $\mathbf{E} \in \mathbb{R}^{M \times D}$ be a matrix whose columns are $\{\boldsymbol{e}_1, ..., \boldsymbol{e}_D\}$. Let $\boldsymbol{\Lambda}$ be a diagonal matrix whose diagonal is formed by $\{\frac{1}{\lambda_1 + \sigma^2}, ..., \frac{1}{\lambda_D + \sigma^2}\}$. In the space of the $D$ eigenvectors the linear transformation on any vector like $\boldsymbol{b}$ is equal to $\mathbf{E}\boldsymbol{\Lambda}\mathbf{E}^T\boldsymbol{b}$, meaning that multiplication by $\mathbf{E}^T$ transforms $\boldsymbol{b}$ to the space of the $D$ eigenvectors, multiplication by $\boldsymbol{\Lambda}$ performs the transformation in that space, and multiplication by $\mathbf{E}$ transforms the result back to the original space. The $(M - D)$ eigenvalues that correspond to the rest of the eigenvectors are all the same and are equal to $\frac{1}{\sigma^2}$. Therefore, there is no need to project $\boldsymbol{b}$ to the space of the $(M - D)$ eigenvectors because the linear transformation in that space is simply a scaling by $\frac{1}{\sigma^2}$. All in all, we have that

$$(\mathbf{A}\mathbf{A}^T + \sigma^2\mathbf{I}_{M \times M})^{-1}\boldsymbol{b} = \mathbf{E}\boldsymbol{\Lambda}\mathbf{E}^T\boldsymbol{b} + \frac{1}{\sigma^2}(\boldsymbol{b} - \mathbf{E}\mathbf{E}^T\boldsymbol{b}). \text{ (S21)}$$

Complexity of computing the right-hand-side of Eq.S21 is way lower than the $\mathcal{O}(M^3)$ requirement of the standard matrix inversion. We borrowed more computational ideas from the work on fast spectral clustering [10]. To compute the first $D$ eigenvlaues and eigenvectors of $\mathbf{A}\mathbf{A}^T$, we worked with the $D$-by-$D$ matrix $\mathbf{A}^T\mathbf{A}$ rather than the $M$-by-$M$ matrix $\mathbf{A}\mathbf{A}^T$ (recall that $D << M$), because given the eigenvalues and eigenvectors of $\mathbf{A}^T\mathbf{A}$, those of $\mathbf{A}\mathbf{A}^T$ are easily computable [10]. The procedure is explained in Alg.S5. In Alg.S5, lines 1-3 compute the eigenvalues/vectors of the matrix $\mathbf{A}^T\mathbf{A}$. Afterwards, lines 4 and 5 compute the first $D$ eigenvalues/vectors of $\mathbf{A}\mathbf{A}^T$ using those of $\mathbf{A}^T\mathbf{A}$. Finally, line 8 computes $(\mathbf{A}\mathbf{A}^T + \sigma^2\mathbf{I})^{-1}\boldsymbol{b}$ according to the right-hand-side of Eq.S21. To make the computations faster, we made use of the following equation $\mathbf{A}\mathbf{A}^T = \sum_m \mathbf{A}[m, :]\mathbf{A}[m, :]^T$, where $\mathbf{A}[m, :]$ is the $m$-th row of the matrix $\mathbf{A}$. Thanks to this equation, we compute $\mathbf{A}\mathbf{A}^T$ only once at the beginning of the training. Afterwards, as each mini-batch alters only some rows of $\mathbf{A}$, we update the previously computed $\mathbf{A}\mathbf{A}^T$ by considering only the effect of the modified rows.

## S2.2 Computing Pixel Contributions to the Similarity

We first explain the idea of CAM [34], afterwards we modify it for the architectures of our kernel modules. Let the kernel mapping $f(.)$ be a convolutional neural network that produces a volumetric map of size $C \times H \times W$ followed by a spatial average pooling that produces the $C$-dimensional vector in the kernel-space. In this case, $\mathcal{K}(\boldsymbol{x}_1, \boldsymbol{x}_2)$ is as follows:

$$\mathcal{K}(\boldsymbol{x}_1, \boldsymbol{x}_2) = f(\boldsymbol{x}_1)^T f(\boldsymbol{x}_2)$$
$$= \left(\sum_{i=1}^{H}\sum_{j=1}^{W} \boldsymbol{z}_{ij}^{(1)}\right)^T \left(\sum_{k=1}^{H}\sum_{\ell=1}^{W} \boldsymbol{z}_{k\ell}^{(2)}\right)$$
$$= \sum_{i=1}^{H}\sum_{j=1}^{W}\sum_{k=1}^{H}\sum_{\ell=1}^{W} \left(\boldsymbol{z}_{ij}^{(1)}{}^T \boldsymbol{z}_{k\ell}^{(2)}\right),$$
$$\text{(S22)}$$

where $\boldsymbol{z}^{(1)}$ and $\boldsymbol{z}^{(2)}$ are the volumetric maps of size $C \times H \times W$ and the indices $(i, j)$ and $(k, \ell)$ index the spatial locations over the volumetric maps. The last term in Eq.S22 shows that the total similarity $\mathcal{K}(\boldsymbol{x}_1, \boldsymbol{x}_2)$ is the sum of the contributions from each pair of positions $(i, j)$ on $\boldsymbol{x}_1$ and $(k, \ell)$ on $\boldsymbol{x}_2$. To compute the contribution of a specific location like $(i, j)$ on $\boldsymbol{x}_1$, we sum up the contributions of $(i, j)$ on $\boldsymbol{x}_1$ and all possible locations $\{(k, \ell)\}_{k=1 \ell=1}^{H \quad W}$ on $\boldsymbol{x}_2$.

The kernel-mappings that we used have a slightly different architecture than a volumetric map followed by spatial average pooling. Our kernel mappings produce a volumetric map of size $C \times H \times W$ followed by a spatial average pooling that produces a $C$-dimensional vector. Afterwards, the resulting vector is divided by its $\ell_2$-norm to produce a vector of norm 1. Consequently, this vector of norm 1 is fed to a leaky ReLU layer that produces the final kernel-space representation $f(\boldsymbol{x})$. For this architecture the pixel contributions can be computed according to an equation similar to Eq.S22 as follows. Our kernel mappings produce the volumetric map $\boldsymbol{z}$ of size $C \times H \times W$ followed by a spatial average pooling that produces the $C$-dimensional vector $\boldsymbol{a}$:

$$\boldsymbol{a} = \sum_{i=1}^{H} \sum_{j=1}^{W} \boldsymbol{z}_{ij}. \tag{S23}$$

Afterwards, the resulting vector is divided by its $\ell_2$-norm to produce the vector $\boldsymbol{b}$ of norm 1:

$$\boldsymbol{b} = [\frac{a_1}{||\boldsymbol{a}||_2}, \ldots, \frac{a_C}{||\boldsymbol{a}||_2}]. \tag{S24}$$

Consequently, this vector of norm 1 is fed to a leaky ReLU layer that produces the final kernel-space representation $f(\boldsymbol{x})$:

$$f(\boldsymbol{x}) = leakyReLU(\boldsymbol{b}). \tag{S25}$$

We begin with simplifying Eq.S25. The leaky ReLU activation function multiplies the input by a constant and this constant depends on the sign of the input. Therefore, applying the leaky ReLU activation is equivalent to multiplication by a diagonal matrix $\boldsymbol{\Lambda}$. Therefore,

$$f(\boldsymbol{x}) = \boldsymbol{\Lambda} \boldsymbol{b}. \tag{S26}$$

Let $\boldsymbol{x}_1$ and $\boldsymbol{x}_2$ be two images, and $\boldsymbol{z}^{(1)}$ and $\boldsymbol{z}^{(2)}$ be the corresponding volumetric maps. We have that

$$\boldsymbol{a}^{(1)} = \sum_{i=1}^{H} \sum_{j=1}^{W} \boldsymbol{z}_{ij}^{(1)},$$
$$\boldsymbol{a}^{(2)} = \sum_{k=1}^{H} \sum_{\ell=1}^{W} \boldsymbol{z}_{k\ell}^{(2)}. \tag{S27}$$

And

$$\boldsymbol{b}^{(1)} = [\frac{a_1^{(1)}}{||\boldsymbol{a}^{(1)}||_2}, \ldots, \frac{a_C^{(1)}}{||\boldsymbol{a}^{(1)}||_2}],$$
$$\boldsymbol{b}^{(2)} = [\frac{a_1^{(2)}}{||\boldsymbol{a}^{(2)}||_2}, \ldots, \frac{a_C^{(2)}}{||\boldsymbol{a}^{(2)}||_2}]. \tag{S28}$$

And

$$f(\boldsymbol{x}^{(1)}) = \boldsymbol{\Lambda}^{(1)} \boldsymbol{b}^{(1)},$$
$$f(\boldsymbol{x}^{(2)}) = \boldsymbol{\Lambda}^{(2)} \boldsymbol{b}^{(2)}. \tag{S29}$$

Now we simplify the similarity $\mathcal{K}(\boldsymbol{x}_1, \boldsymbol{x}_2)$:

$$\begin{aligned} \mathcal{K}(\boldsymbol{x}_1, \boldsymbol{x}_2) &= (\boldsymbol{\Lambda}^{(1)} \boldsymbol{b}^{(1)})^T (\boldsymbol{\Lambda}^{(2)} \boldsymbol{b}^{(2)}) \\ &= (\boldsymbol{\Lambda}^{(1)^T} \boldsymbol{\Lambda}^{(2)}) (\boldsymbol{b}^{(1)^T} \boldsymbol{b}^{(2)}) \\ &= \frac{(\boldsymbol{\Lambda}^{(1)^T} \boldsymbol{\Lambda}^{(2)})}{||\boldsymbol{a}^{(1)}||_2 \; ||\boldsymbol{a}^{(2)}||_2} (\sum_{i=1}^{H} \sum_{j=1}^{W} \boldsymbol{z}_{ij}^{(1)})^T (\sum_{k=1}^{H} \sum_{\ell=1}^{W} \boldsymbol{z}_{k\ell}^{(2)}) \\ &= \frac{(\boldsymbol{\Lambda}^{(1)^T} \boldsymbol{\Lambda}^{(2)})}{||\boldsymbol{a}^{(1)}||_2 \; ||\boldsymbol{a}^{(2)}||_2} \sum_{i=1}^{H} \sum_{j=1}^{W} \sum_{k=1}^{H} \sum_{\ell=1}^{W} (\boldsymbol{z}_{ij}^{(1)^T} \boldsymbol{z}_{k\ell}^{(2)}). \end{aligned} \tag{S30}$$

Indeed, as the used architecture for kernel-mappings is slightly different than producing a volumetric map followed by spatial average pooling, instead of Eq.S22, we used Eq.S30 that we derived above.

## S3 EXAMINING FAITHFULNESS OF GPS TO ANNS

In Sec.4.1. of the main article, we examined the faithfulness of the found GPs to their corresponding ANNs. In this section we provide more information and insights about the analogy between the GPs found by our proposed GPEX and their corresponding ANNs. Figs.S2, S4, and S6 illustrate the scatter plots of ANN-GP outputs on Cifar10 [15], MNIST [6], and Kather [12], respectively. These scatter plots are obtained on the testing set which has been invisible to the proposed GPEX. Note that in Figs.S2, S4, and S6 each ANN's output head and its corresponding GP have a seprate scatter plot.

In the main article, we discussed that our proposed GPEX is applicable to any subcomponent of a pipeline. To verify this, in Sec.4.1. of the main article we applied the proposed GPEX to attention subcomponents of classifier pipelines. Here we provide more information about the faithfulness of the found GPs to the attention subcomponents. Figs.S3, S5, and S7 illustrate the scatter plots for attention submodules and their corresponding GPs.

For Cifar10 [15] in Fig.S3, each attention mask is 3 x 3 and we have 9 scatter plots. According to Fig.S3, in attention masks some output heads like head 1, head 2, and head 3 do not turn on for any instnace (the values change around -2, and sigmoid of -2 is a small number). Therefore, in Fig.3 of the main article we have excluded the attention heads which are always off. Similarly, for MNIST [6] and Kather [12] we see some attention heads are always off in Figs.S5 and S7, and we have excluded those heads in Fig.3 of the main article.

So far we reported corelation coeffients (Fig.3 of the main article) and scatter plots (Figs.S2, S3, S4, S5, S6, S7) to examine the faithfulness of GPs to their corresponding ANNs. To get more insights, we selected mini-batches of testing instances and fed each mini-batch to both ANN and corresponding GPs. The output from ANN (and simmilarly GPs) is a matrix of shape $batchsize \times D_v$, where $D_v$ is the number of output heads from the ANN. Ideally, we should get two identical $batchsize \times D_v$ matrices for each mini-batch, because the GPs are supposed to be faithfull to ANNs. Figs. S59, S60, S61, and S62 illustrate the heatmaps for four randomly fed mini-batches from Cifar10 [15], MNIST [6], Kather [12], and DogsWolves [30], respectively. According

to Figs. S59, S60, S61, and S62 the outputs from GPs almost match those from their corresponding ANNs. In Figs. S59, S60, S61, and S62 the red rectangles show the test instances for which the GP's decision (i.e. the class with the highest score) does not match the ANN's decision. According to Figs. S59, S60, S61, and S62 the disagreement between GPs prediction and ANN prediction mostly happens when either some output activations are very close to one another or all activations are close to zero. This is consistent with the scatter plots of Figs. S2, S4, and S6 in which the scatters are slightly dispersed for intermediate values. Tab. S1 reports the test accuracy of the ANNs and their corresponding GPs. We see that GPs' accuracies are slightly lower than those of the corresponding ANNs. Figs. S59, S60, S61, and S62 provide insights about how this small disagreement can be potentially solved in future research by, e.g., preventing the ANN from having near-zero activations or having output heads which are very close to one another. We repeated the experiment with 5 different random splits and reported the results in Fig. S67. According to Fig. S67 the correlation coefficients are high for different training/testing splits.

We repeated the experiment of Figs. S59, S60, S61, and S62 for the attention submodules and corresponding GPs. Reults are provided in Figs. S63, S64, and S65. According to Figs. S63, S64, and S65 our proposed GPEX has found GPs which are faithful to the attention submodules.

## S4  EXPLAINING ANNs' DECISIONS

In Sec.4.2 of the main article we applied our proposed method (i.e. Alg.S4) to some ANN classifiers. Afterwards, we explained the decisions made by the ANNs via the GPs and the kernel-spaces that our proposed GPEX has found. Here we are going to provide more explanations for ANNs' decisions on more testing instances.

We explain the decision made for a test instance like $x_{test}$ as follows. We consider the GP and the kernel-space that correspond to the ANN's head with maximum value (i.e. the ANN's head that relates to the predicted label). Consequently, among the instances in the inducing dataset, we find the 10 closest instances to $x_{test}$, like $\{x_{i1}, x_{i2}, ..., x_{i10}\}$. Intuitively the ANN has labeled $x_{test}$ in that way because it has found $x_{test}$ to be similar to $\{x_{i1}, x_{i2}, ..., x_{i10}\}$. Besides finding the nearest neighbours, we provide explanation as to why $x_{test}$ and an instance like $x_{ij}, 1 \leq j \leq 10$ are considered similar by the model. The procedure is explained in Sec.S2.2.

For MNIST digit classification, some test instances and nearest neighbours in training set are shown in Figs.S8, S9, S10, and S11. In these figures each row corresponds to a test instance. The first column depicts the test instance itself and columns 2 to 11 depict the 10 nearest neighbours. According to rows 2 and 3 of Fig.S8, the classifier has labeled the two images as digit 1 because it has found 1 digits with similar inclinations in the training set. We see the model has also taken the inclination into account for the test instances of rows 8 and 9 of Fig.S8 and rows 1, 2, and 3 of Fig.S11. In Fig.S8, according to rows 4, 5, and 6 the test instances are classified as digit 2 because 2 digits with similar styles are found in the training set. We see the model has also taken the style into account for the test instances of rows 7, 8, 9,

10, 11 of Fig.S8 and rows 1, 2, 3, 4, 5, 6, 7, and 8 of Fig.S9. For instance, the test instance in row 1 of Fig.S9 is a 4 digit with a short tail and the two nearest neighbours are alike. Or for the test instances in rows 5, 6, 7, and 8 of Fig.S10 the test instances have incomplete circles in the same way as their nearest neighbours.

Figs.S12, S13, S14, S15, S16, S17, S18, and S19 illustrate sample explanations for similarities. For instance row 1 of Fig.S12 illustrates a test instance as well as the 10 nearest neighbours. The second row of Fig.S12 highlights to what degree each region of each nearest neighbour contributes to its similarity to the test instance. The third row of Fig.S12 illustrates to what degree each region of the test instance contributes to its similarity to each of the nearest neighbours. For example, according to rows 1, 2, and 3 of Fig.S17 the cross pattern of the 8 digits have had a significant contribution to their similarities. For MNIST [6], more similarity explanations are provided in Figs.S12, S13, S14, S15, S16, S17, S18, and S19.

Figs.S36, S37, S38, S39, S40, S41, S42, S43, S44, S45, S46, S47, S48, and S49 illustrate some sample explanations for Cifar10 [15]. Like before, each row corresponds to a test instance, the first column depicts the test instance itself and columns 2 to 11 depict the 10 nearest neighbours. In rows 8, 9, 10, and 11 of Fig.S44 and rows 1 and 2 of Fig.S45, the test instances are captured from horses' heads from closeby, and the nearest neighbours are alike. However, in rows 3, 4, 5, 6, 7, 8, and 9 of Fig.S45 the test images are taken from faraway and the found similar training images are also taken from faraway. Intuitively, as the classifier is not aware of 3D geometry, it finds training images which are captured from the same distance. We constantly observe this pattern in more explanations: row 6, 7, 8, 9, 10, and 11 in Fig.S39, all rows of Fig.S40, rows 1, 2, 6, 7, 8, 9, 10 and 11 of Fig.S41, rows 1, 7, 8, 9, 10, and 11 of Fig.S42, rows 1, 2, 3, 4, 5, 6, 7, and 8 of Fig.S43, all rows of Fig.S45 and rows 1-10 of Fig.S46.

Animal faces tend to be recognized by similar faces. We see this pattern in rows 2, 3, 4, 5 and 6 of Fig.S40, rows 6, 7, 8, and 9 of Fig.S41, rows 7 and 8 of Fig.S43, rows 8, 9, 10, and 11 of Fig.S44 and rows 1, 2, 10, and 11 of Fig.S45. To classify airplanes, the model has taken into account the inclination. For instance, in Fig.S36 the model has taken into account whether the airplane is taking off (rows 1, 8, 9, 10, and 11 of Fig.S36), flying straight (rows 2 and 4 of Fig.S36) or is inclined downwards (rows 3, 5, 6 and 7 of Fig.S36). Furthermore, the bat-like airplanes are recognized by the model because similar bat-like airplanes are found in the training set, as we see in rows 1, 2, 3, 4, 5, 6 and 7 of Fig.S37. Cessnas are often classified by finding cessnas in the training set, as we see in rows 8, 9 and 10 of Fig.S37 and row 1 of Fig.S38.

Since the classifier has no knowledge about 3D geometry, it tends to find training instances which are captured from the same angle as the test instance, as we see in rows 6, 7, 8, 9, 10 and 11 of Fig.S39, rows 7, 8, 9, 10 and 11 of Fig.S42, rows 9, 10 and 11 of Fig.S43, rows 1, 2, 3, 4, 5, 6 and 7 of Fig.S44, row 11 of Fig.S46, all rows of Fig.S47, and rows 1, 2, 3, 4, 5, 6, 7, and 8 of Fig.S48. In rows 3, 4, and 5 of Fig.S41 it seems the model takes into account the ostrich-like shape of the animal. In rows 2, 3, 4, and 6 of Fig.S42 the horns seem to have an effect. In rows 6, 7, 8, and 9 of Fig.S45, we see

the model have made use of the riders to classify the test instances as horse. According to rows 1, 2, 3, 4, 5, 6, 7, 8, 9, and 10 of Fig.S46, the model distinguishes between medium sized ships and huge cargo ships. To classify firefighter trucks, model tends to find similar firefighter trucks in the training set, as we see in rows 10 and 11 of Fig.S47, and rows 1, 2, 3, and 4 of Fig.S48. For some testing instances, the model finds training instances which are almost identical to the test instance, as we see in rows 2 and 5 of Fig.S40, row 7 of Fig.S42, row 8 of Fig.S43, and row 8 of Fig.S48.

In rows 2, 4, 5, 6, 7, 8, 9, 10, and 11 of Fig.S38 it seems the classifier has taken into account the blue background. We used the proposed GPEX to explain as to why some testing instances get missclassified. Rows 9, 10, and 11 of Fig.S48 and all rows of Fig.S49 illustrate some instances which are misclassified. For instance in row 10 of Fig.S48 the test image shows an airplane, but the model has classified it as a cat, because it is similar to the cat faces shown in columns 2 to 11 (can you find the cat face in the airplane image?). In row 11 of Fig.S48, the car is classified as truck partially because it very similar to the truck at column 2. In row 2 of Fig.S49, the deer is classified as horse partially because it is very similar to the training image shown in column 2. In row 3 of Fig.S49, we hypothesize the dog is classified as cat because the model has taken into account the cyan and red colors in the background. In this case, adding dog images with cyan and red background may make the model classify this test instance correctly. In rows 5 and 6 of Fig.S49, the model correctly understands the test images are similar to some faces from other animals, but it fails to find similar frog faces in the training set. In this case, adding more images from frog faces may solve this issue. In row 7 of Fig.S49 the horse is classified as airplane, because the model thinks the horse image is similar to some airplane training images which are taking off. Interestingly, the jumping frog in column 4 has been considered similar to the horse image. It seems having inclined edges (due to taking off, jupming) has contributed to the similarities, and therefore the model has incorrectly classified the horse as airplane.

For the DogsWolves dataset [30] the explanations are provided in Figs.S25-S35. According to rows 10, 11, and 12 of Fig.S29, the red ball in the dog's mouth (as highlighted in row 12 of Fig.S29) has the most contribution to the similarities. According to row 2 of Fig.S29, patterns like human hand in column 4 or woody or pink background in columns 8, 10, and 11 are highlighted in nearest neighbours while in the test insntace (row 3 of Fig.S29) the red ball at the bottom right is highlighted. Our explanations consistently show that the model detects dogs by any pattern that rarely appear in a wolf image. For instance in rows 4-6 of Fig.S29, according to row 4 humans in columns 3, 9, and 11, and dog collars or costumes in columns 4, 5, 6, and 10, and the brick wall in the test instance (row 6 of Fig.S29) are used by the model. According to rows 9, 12, and 15 of Fig.S29, the flowers, the red ball in the dogs mouth, and the children are used by the model, respectively. According to rows 3, 6, 9, 12, and 15 of Fig.S30, the red rope, the dog's color, red patterns, brown background and brown background are used by the model, respectively. According to rows 3, 6, 9, 12, and 15 of Fig.S31, brown background, human, brown background, the red wallet, and the pink ball are

used by the model, respectively. According to rows 3, 6, 9, 12, and 15 of Fig.S32, the child, pink pillow, brown color, orange background, and red blood are used by the model, respectively. Note that in Fig.S32 the last two instances (rows 10-15) are misclassified. In Fig.S33 all test instances get misclassified. According to rows 3, 6, 9, 12, and 15 of Fig.S33, colorful background, the red object attached to the wolf, background, white background, and dark-green background are used by the model, respectively. Figs.S34 and S35 illustrate more explanations. For instance, according to row 6 of Fig.S34 and row 12 of Fig.S35, the test instances are misclassified due to their dark background. Moreover, according to rows 3, 6, and 15 of Fig.S35, the test instances are misclassified due to their background. All in all, our explanations reveal that for the DogsWolves dataset [30] the model makes use of potentially incorrect clues to label instances. This is not surprising because the dataset has only 2000 images.

For Kather dataset [12], some explanations are shown in Figs.S20, S21, S22, S23, and S24. Like before, in Figs.S20 and S21 each row corresponds to a test instance, the first column depicts the test instance itself and columns 2 to 11 depict the 10 nearest neighbours. In row 1 of Fig.S20, the test image is classified as fat tissue. According to rows 1, 2, and 3 of Fig.S22, the similarity is due to the wire mesh formed by cellular membranes described by our expert pathologist. Row 13 of Fig.S22 shows cancer-associated stroma which is classified correctly. All 10 nearest neighbours are also cancer-associated stroma. Distinguishing between cancer-associated stroma and normal smooth muscle is a challenging task even for expert pathologists, and they often look similar. According to rows 13, 14, and 15 of Fig.S22, the model cares about both the stroma and nuclei. In row 7 of Fig.S22, the test image is correctly classified as lymphocytes. For a pathologist they represent scattered well defined round structures. According to rows 7, 8, and 9 of Fig.S22, the model considers all regions which matches the way pathologists recognize lymphocytes. In rows 1, 2 and 3 of Fig.S23 and rows 1, 2, and 3 of Fig.S24, for the two test instances the model takes into account nuclei which is not the same way that a pathologists would classify the images. We hypothesize that for the model it is easier to extract features from nuclei than to consider the context information. Because even small changes in nuclei is easily measurable by the model while it is not easily noticeable by human eyes. The test image in row 7 of Fig.S24 gets missclassified. According to rows 7, 8, and 9 of Fig.S24 the artificial white holes are considered as glandular lumens by the model and that explains why the test instance gets misclassified. The test image in row 10 of Fig.S23 gets misclassified. According to rows 10, 11, and 12 of Fig.S23, the test image is smooth muscle. But it contains artifactual white spaces (retractions) like the found similar training instances. This make the model think the test image is similar to debris images that contain artifactual white spaces. For Kather dataset [12], more sample explanations are provide in Figs.S22, S23, and S24.

## S5 Practical Details and Parameter Settings

In this section we discuss some practical details which we have not yet discussed in this paper. Moreover, we provide the exact parameter settings that we used throughout our experiments. As explained in Sec.3 of the main article, there are $L$ kernel mappings that we denoted by $[f_1(.), f_2(.), ..., f_L(.)]$. One can implement this kernel mappings by, e.g., considering $L$ independent CNNs. However, doing so dramatically increases the computation cost. Therefore, we modeled the $L$ mappings by a common ResNet-50 [9] backbone. After the common backbone, we placed $L$ branches. Each branch has two convolutional layers followed by global spatial average pooling that produce a vector. Each branch ends with an L2 normalizer layer (that sets the L2-norm of the vector to 1) followed by a leaky-ReLU layer. During our experiments, we noticed that the L2-normalization layer and the final leaky-ReLU layer are essential. Without the L2 normalization layer, the vectors in the kernel-space can have arbitrarily-small or arbitrarily-big elements, and this makes the training unstable. We included the last leaky-ReLU layer, because according to GP posterior mean formula, vectors in the kernel-space go through a linear transformation. Therefore, without the last leaky-ReLU layer, the pipeline would have two consequtive linear layers. Throughout our experiments, we set the output of each branch (i.e. vectors in the kernel-space of each GP) to be 20-dimensional.

As illustrated by Fig.S66 (to be discussed in Sec.S7), we need to make the inducing dataset as large as possible. Therefore, throughout our experiments we selected the whole training dataset as the inducing dataset. Unlike training instances, we didn't apply data-augmentation on inducing instances. By doing so, the training dataset and the inducing dataset will have very similar instances. This causes a difficulty that we are going to discuss in this part. The kernel-mappings $[f_1(.), ..., f_L(.)]$ are trained according to Alg.S4. After selecting an instance like $x$ from the training dataset, $x$ is actually the augmented version of an inducing instance like $\tilde{x}_m$. Indeed, we have that $x = DataAug(\tilde{x}_m)$. Because $x$ and $\tilde{x}_m$ are very similar, they will be very close to one another in the kernel-spaces regardless of what parameters $[f_1(.), ..., f_L(.)]$ have. Therefore, regardless of the kernel-mappings, the GP-mean will match the ANN value at $x$, and there will be no training signal for the kernel-mappings $[f_1(.), ..., f_L(.)]$. Note that in this case the GPs match the ANNs only on training instances, and the analogy does not generalize to testing instances. To avoid this issue, we optimized the GP-ANN analogy (i.e. the objective in Eq.5 of the main article) on instances like $\lambda x_i + (1 - \lambda) x_j$, where $x_i$ and $x_j$ are two instances randomly selected from the training set and $\lambda$ is a scalar uniformly selected from $[-1, 2]$.

When applying our proposed GPEX we used Adam optimizer [14]. Although the AMSGrad version of this optimizer is often recommended, for our proposed GPEX we noticed the Adam optimizer [14] without AMSGrad works the best. For explaining classifier ANNs, we used a learning-rate of 0.0001 while for explaining the attention submodules we used a learning rate of 0.00001. On a RTX3090 GPU, the experiments took around 3 days for the 4 image datasets

and around 2 hours for the biological dataset. We ran Alg. S4 for 200 epochs. Afterwards, we ran line 8 of Alg. S4 for the inducing dataset. Afterwards, we continued Alg.S4 for 200 more epochs and repeating line 8 of Alg.S4 10 times instead of once.

## S6 Qualitative Comparision of GPEX and Representer Point Selection

We qualitatively compared the explanations of our proposed GPEX to those of representer point selection [33]. The results are provided in Figs.S50, S51, S52, S53, S54, S55, S56, and S57. In each triple, the first row shows the test instance and the 10 nearest neighbours found by our proposed GPEX. The second row shows the 10 nearest neighbours selected by representer point selection [33]. The third row shows the 10 nearest neighbours according to the kernel-space of representer point selection [33]. Representer point selection [33] assigns an importance weight to each training instance. Therefore, some training instances tend to appear as nearest neighbours regardless of what the testing instance is. We see this behaviour in rows 2, 5, 8, 11, and 14 of Figs.S50-S57. However, for our proposed GPEX the nearest neighbours can freely change for different test instances. We see this behaviour in rows 1, 4, 7, 10, and 13 of Figs.S50-S57. If we ignore the importance weights in representer point selection [33], the aforementioned issue in that method happens less frequently, as we see in rows 3, 6, 9, 12, and 15 of Figs.S50-S57. However, the issue is that without the importance weights, the explainer model in representer point selection [33] will not be faithful to the ANN itself.

## S7 Parameter Analysis

To analyze the effect of the number of inducing points (i.e. the variable $M$ in Sec. S2) we applied the proposed GPEX to the classifier CNN that we trained on Cifar10 dataset [15] in Sec. 4.1 of the main manuscript. This time, instead of considering all training instances as the inducing dataset, we randomly selected some training instances. In Fig. S66, the horizontal axis shows the size of the inducing dataset. For each size, we repeated the experiment 5 times (i.e. split 1-5 in Fig. S66). According to Fig. S66, to obtain GPs which are faithful to ANNs one needs to have a lot of inducing points. This highlights the importance of the scalability techniques that we used (the computational techniques are elaborated upon in Sec. S2.1 of supplementary material). Another intriguing point in Fig. S66 is that if we are to select a few training images as inducing points, the correlation coefficients highly depend on which instances are selected. More precisely, Fig. S66 suggests that one may be able to reach high correlation coefficients by selecting a few inducing points from the training set in a subtle way.

So far we analyzed the effect of the size of inducing dataset. Here we analyze two other important factors: the width of the second last layer and number of epochs for which the ANN has been trained. On Cifar10 [15] we trained ANNs with different number of neurons in the second last layer and we analyzed the ANN at different checkpoints during training (10, 50, 100, 150, and 200 epochs). The result is shown in Fig.S58. According to Fig.S58, increasing the

width of the second last layer increases the correlation coefficients. However, as illustrated by Fig.S58, the proposed GPEX can achieve almost perfect match even when the second last layer of the ANN is not wide. Moreover, according to Fig.S58, our proposed GPEX can reach high correlation coefficients even when the ANN's parameters are not a local minimum of the classification loss. This empirical results show that most theoretical results like requiring all layers of the ANN to be wide [5], or requiring the ANN to be optimized on a loss [20] may not be necessary.

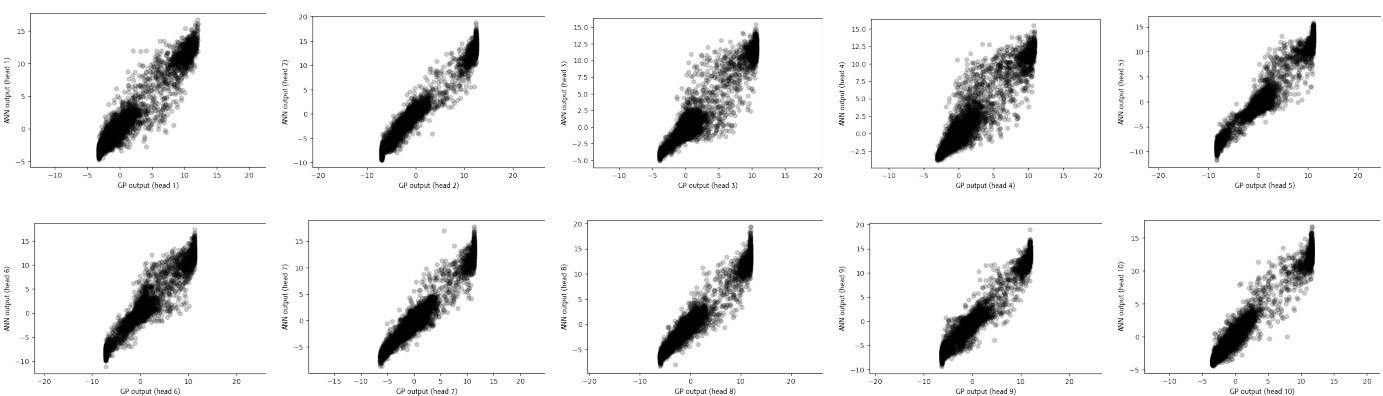

Fig. S2: Scatters for Cifar10 (classifier).

Fig. S3: Scatters for Cifar10 (attention).

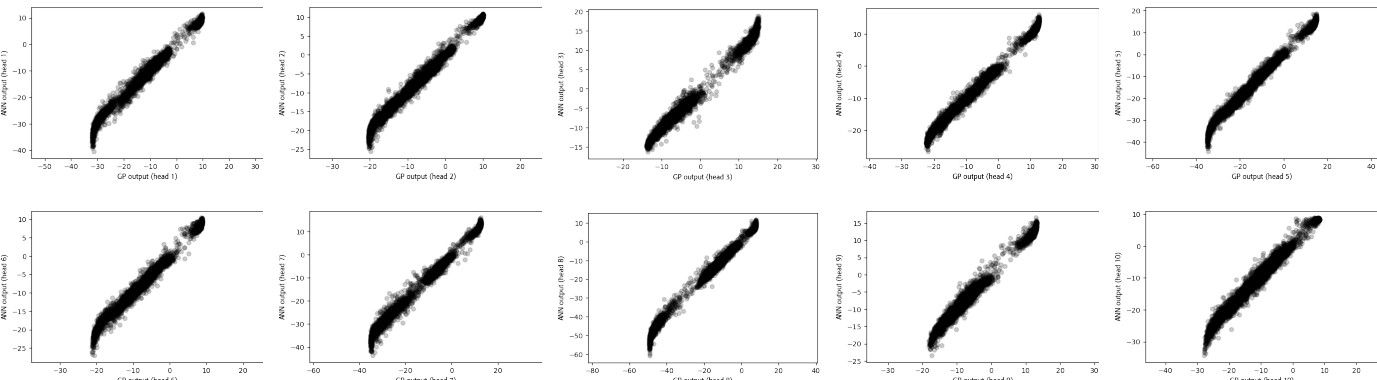

Fig. S4: Scatters for MNIST (classifier).

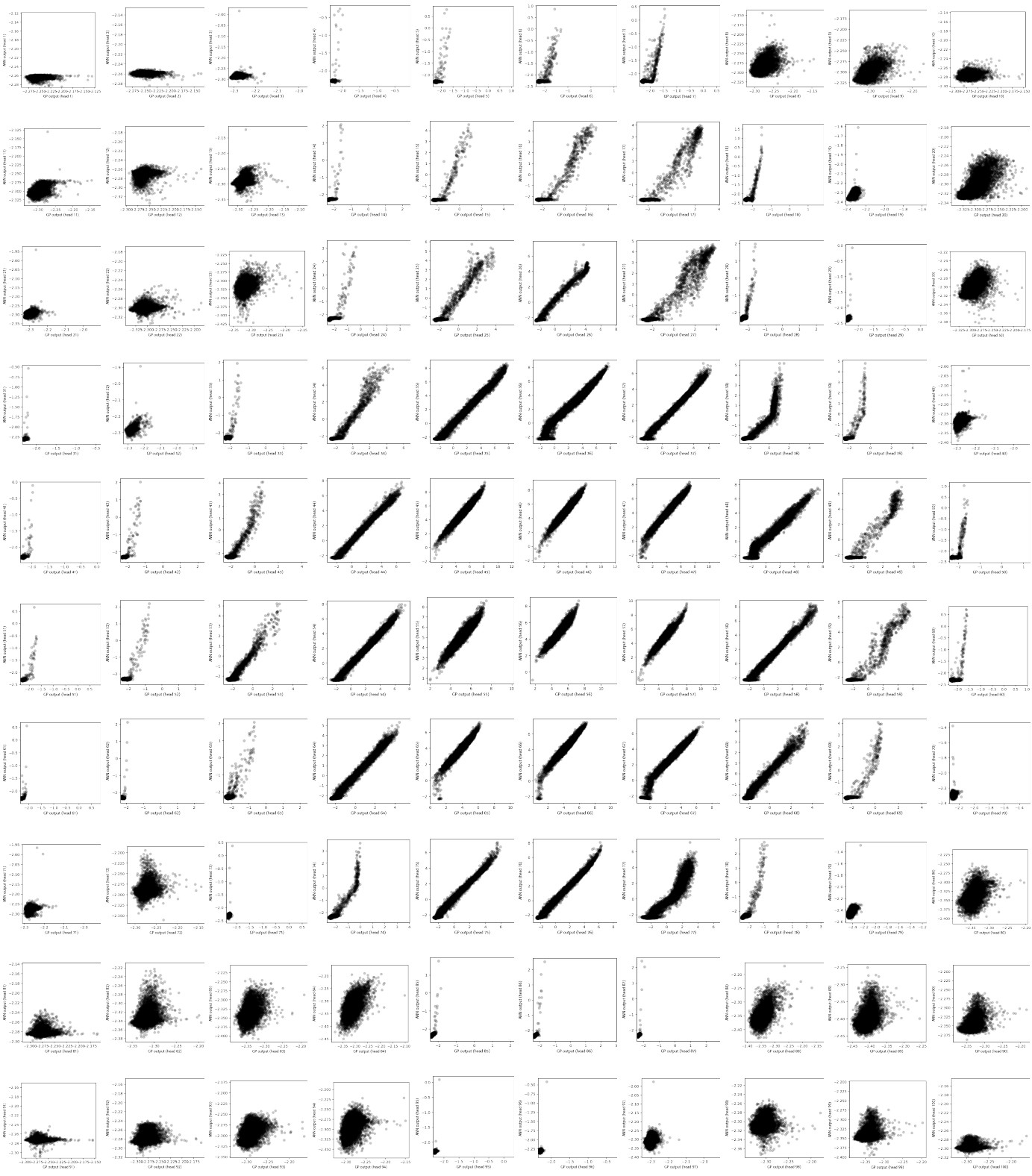

Fig. S5: Scatters for MNIST (attention).

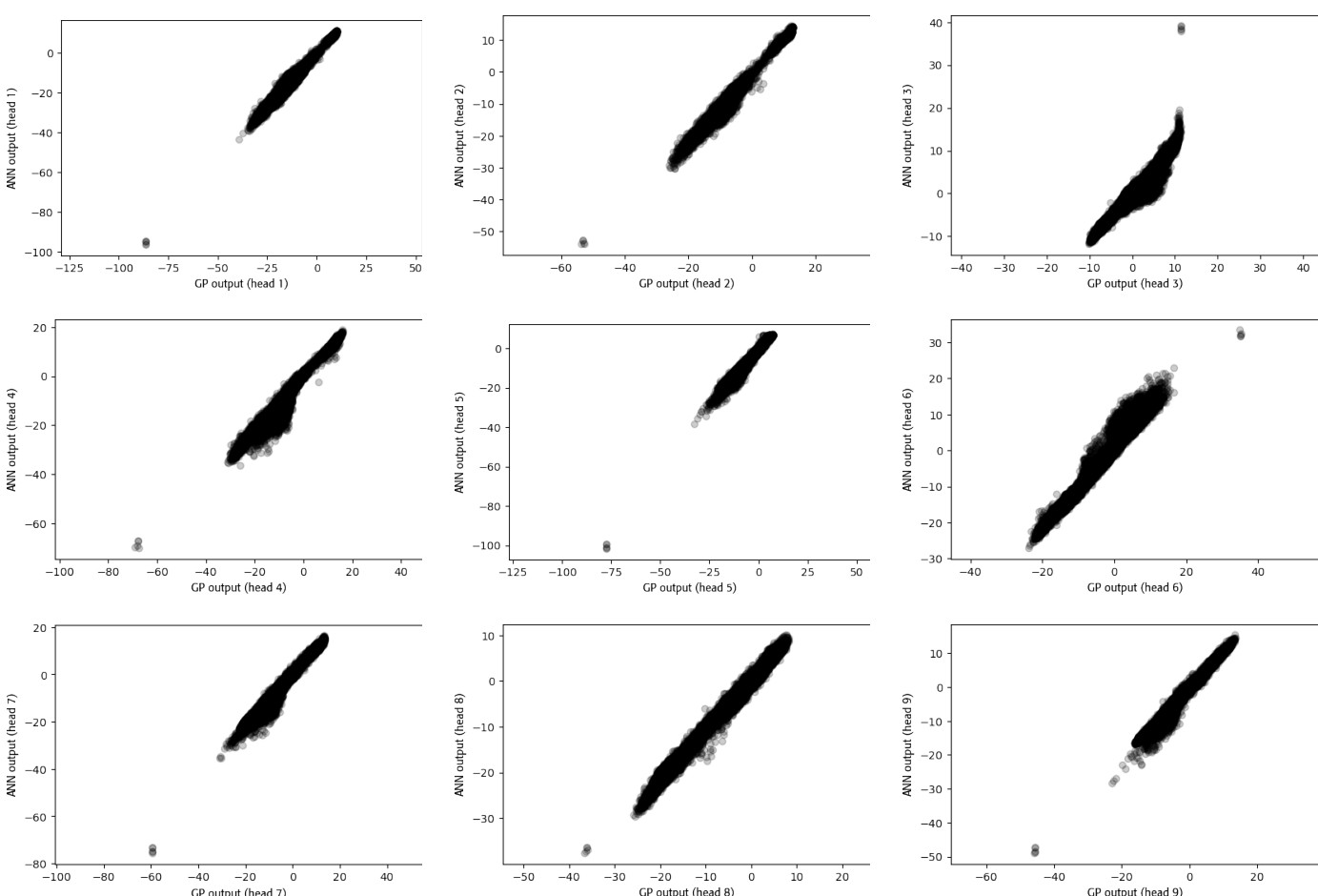

Fig. S6: Scatters for Kather dataset (classifier).

 

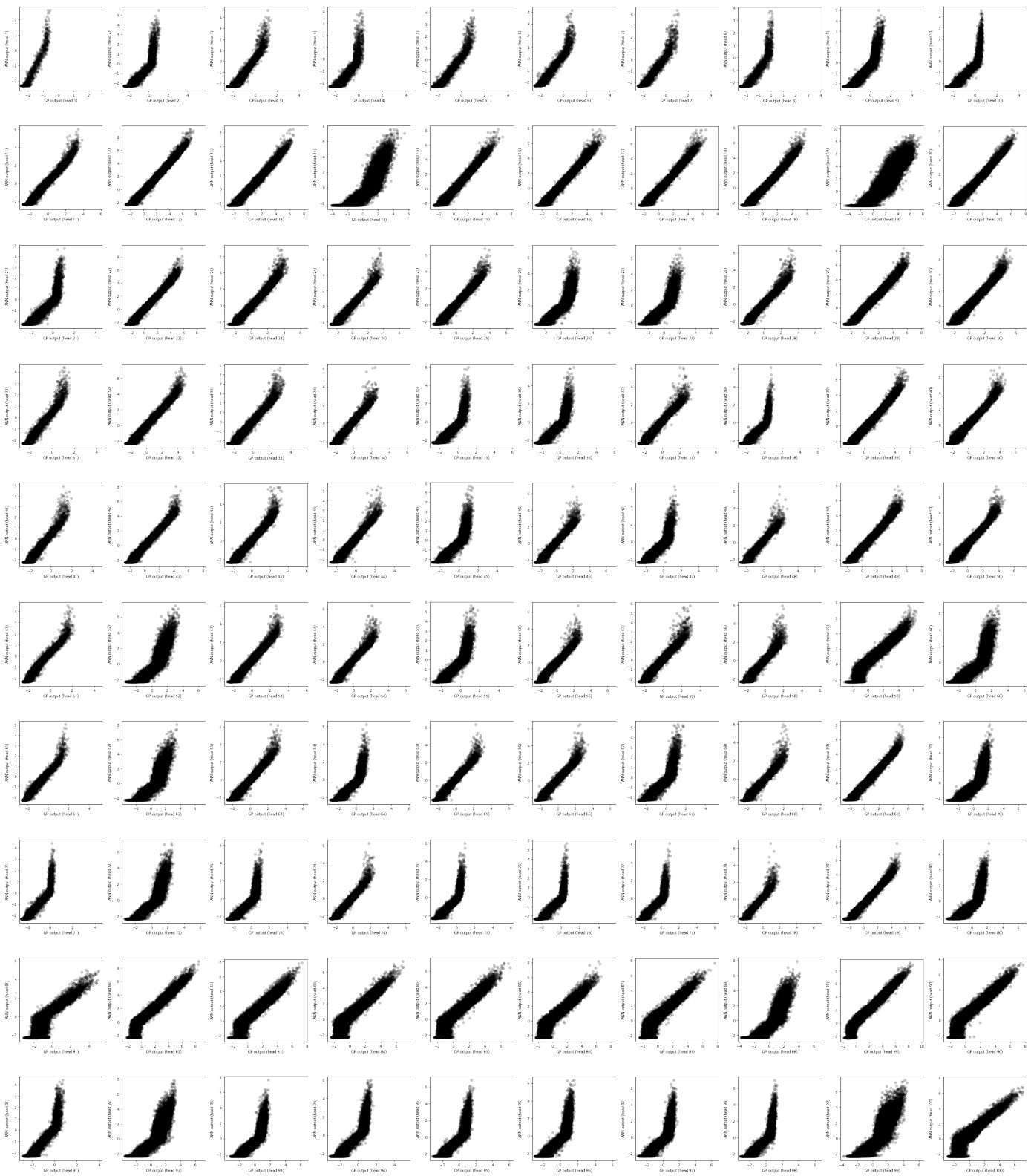

Fig. S7: Scatters for Kather dataset (attention).

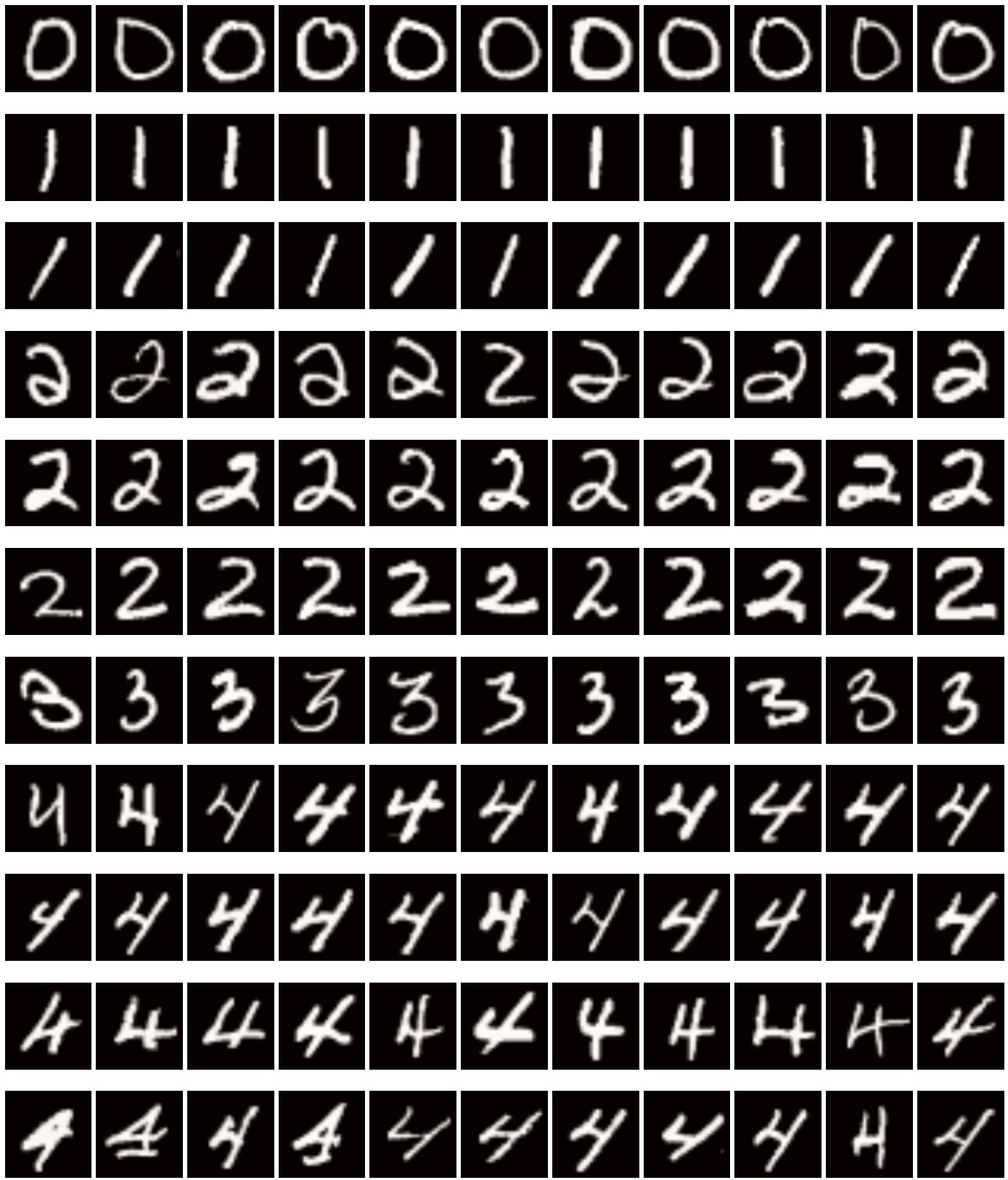

Fig. S8: Explanations for MNIST (set 1).

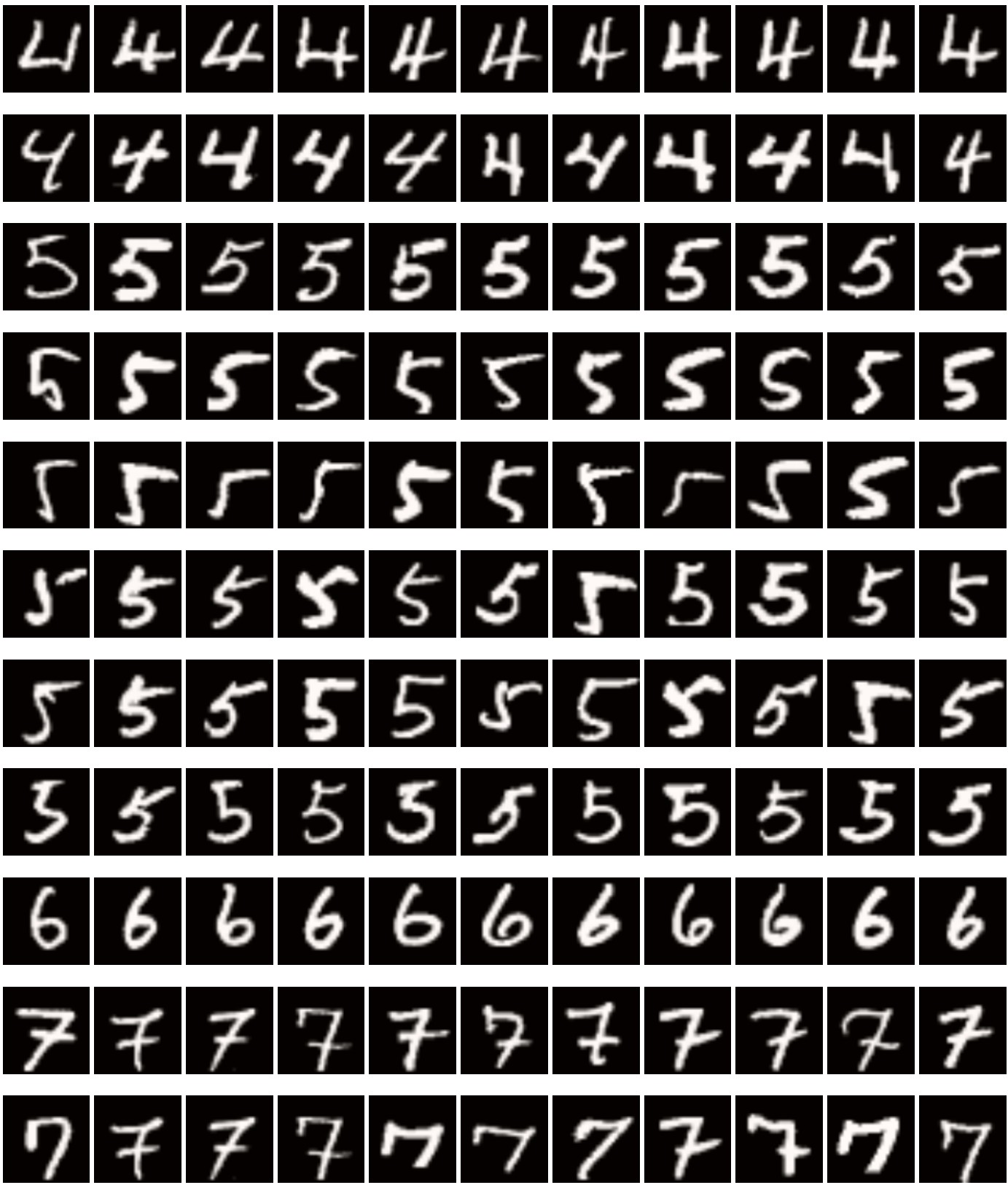

Fig. S9: Explanations for MNIST (set 2).

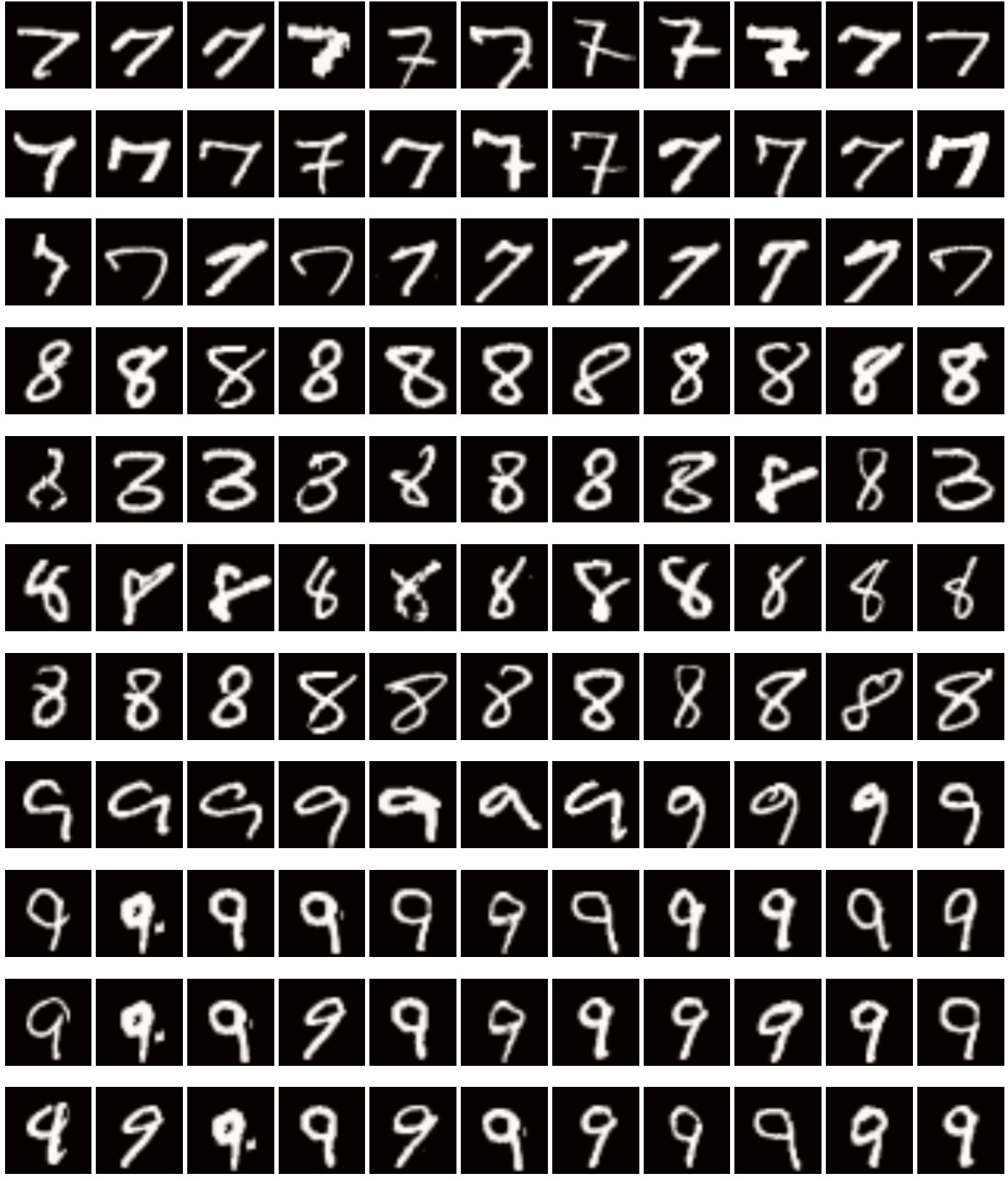

Fig. S10: Explanations for MNIST (set 3).

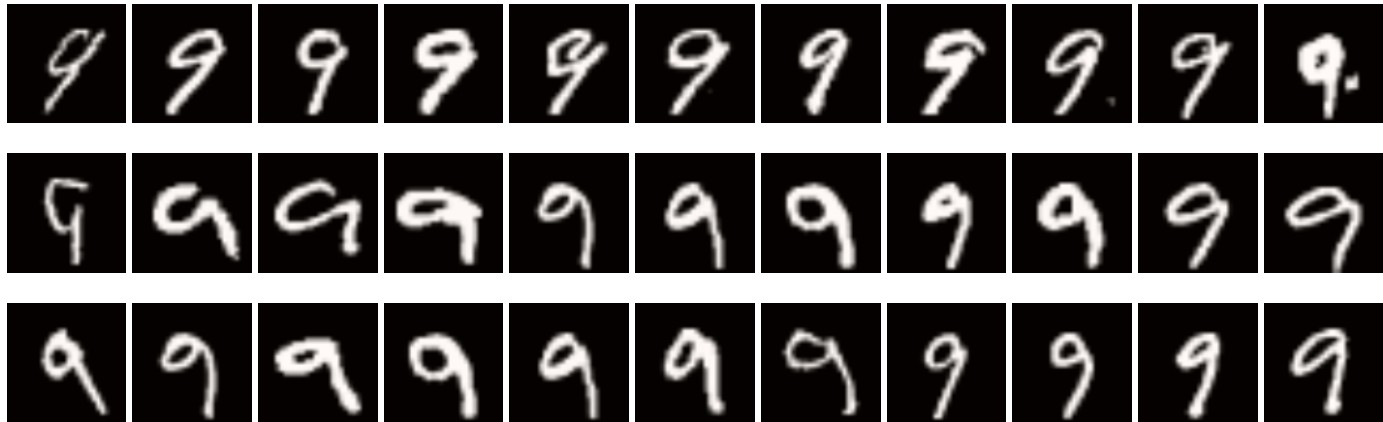

Fig. S11: Explanations for MNIST (set 4).

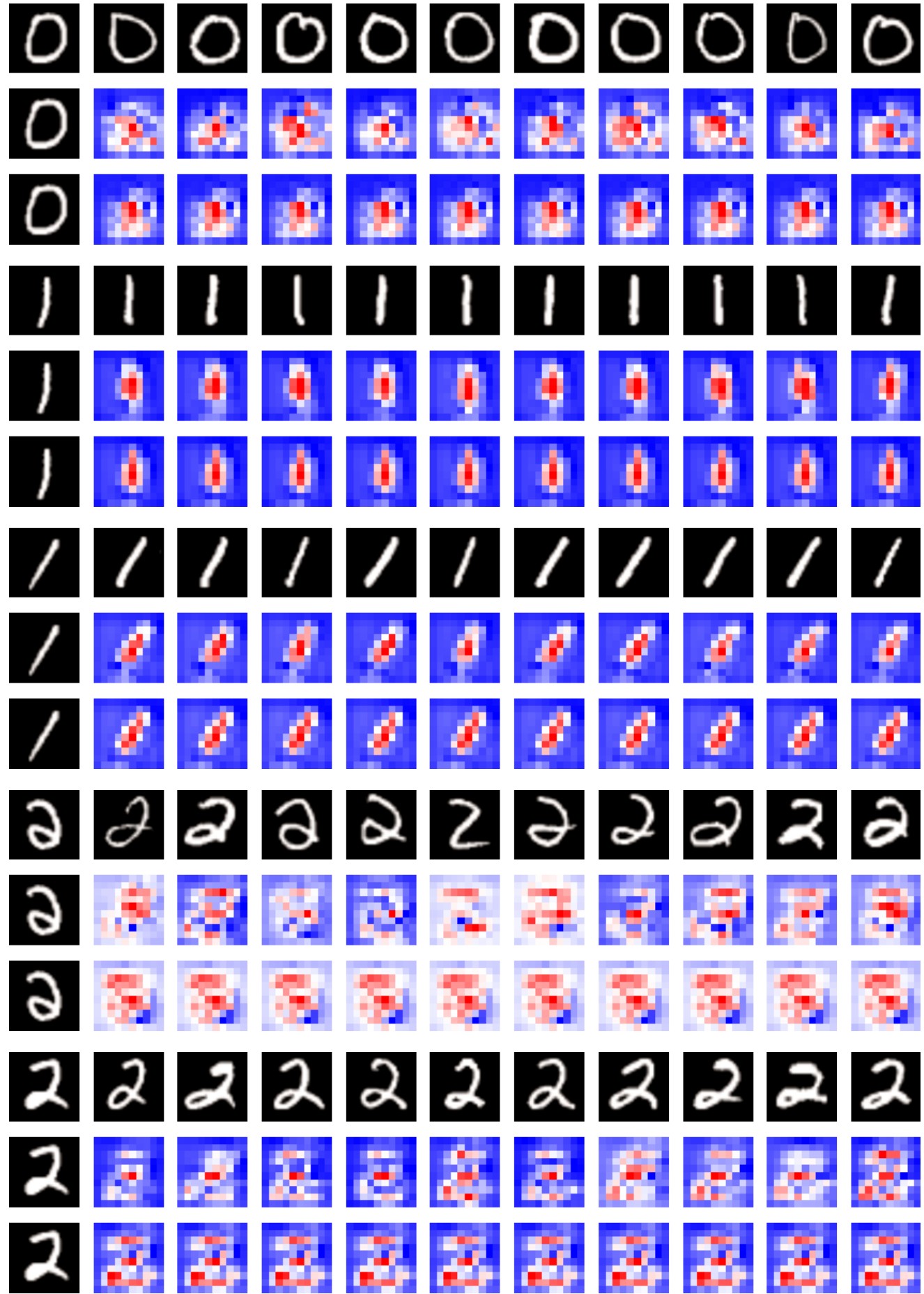

Fig. S12: Explanations for MNIST (set 5).

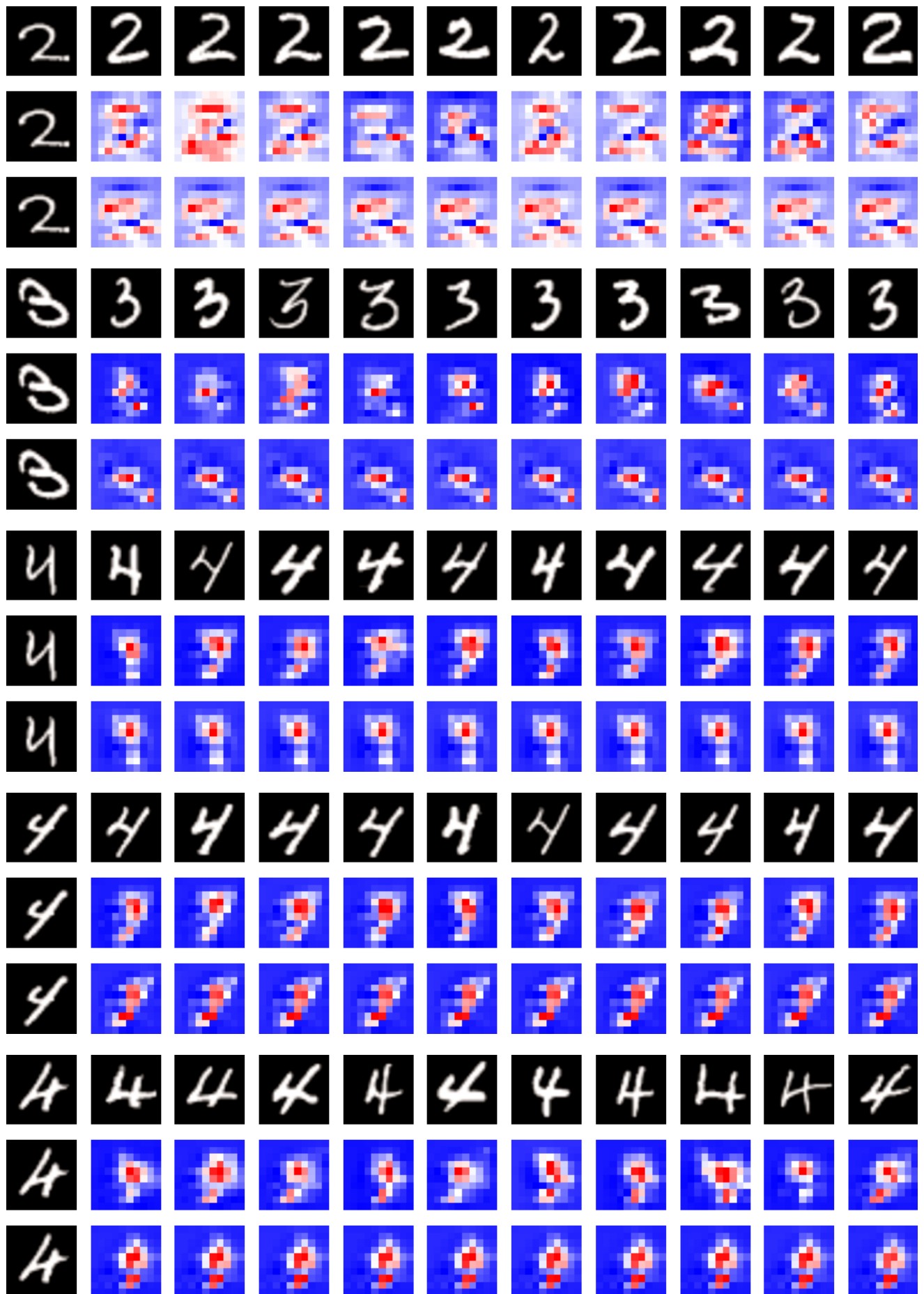

Fig. S13: Explanations for MNIST (set 6).

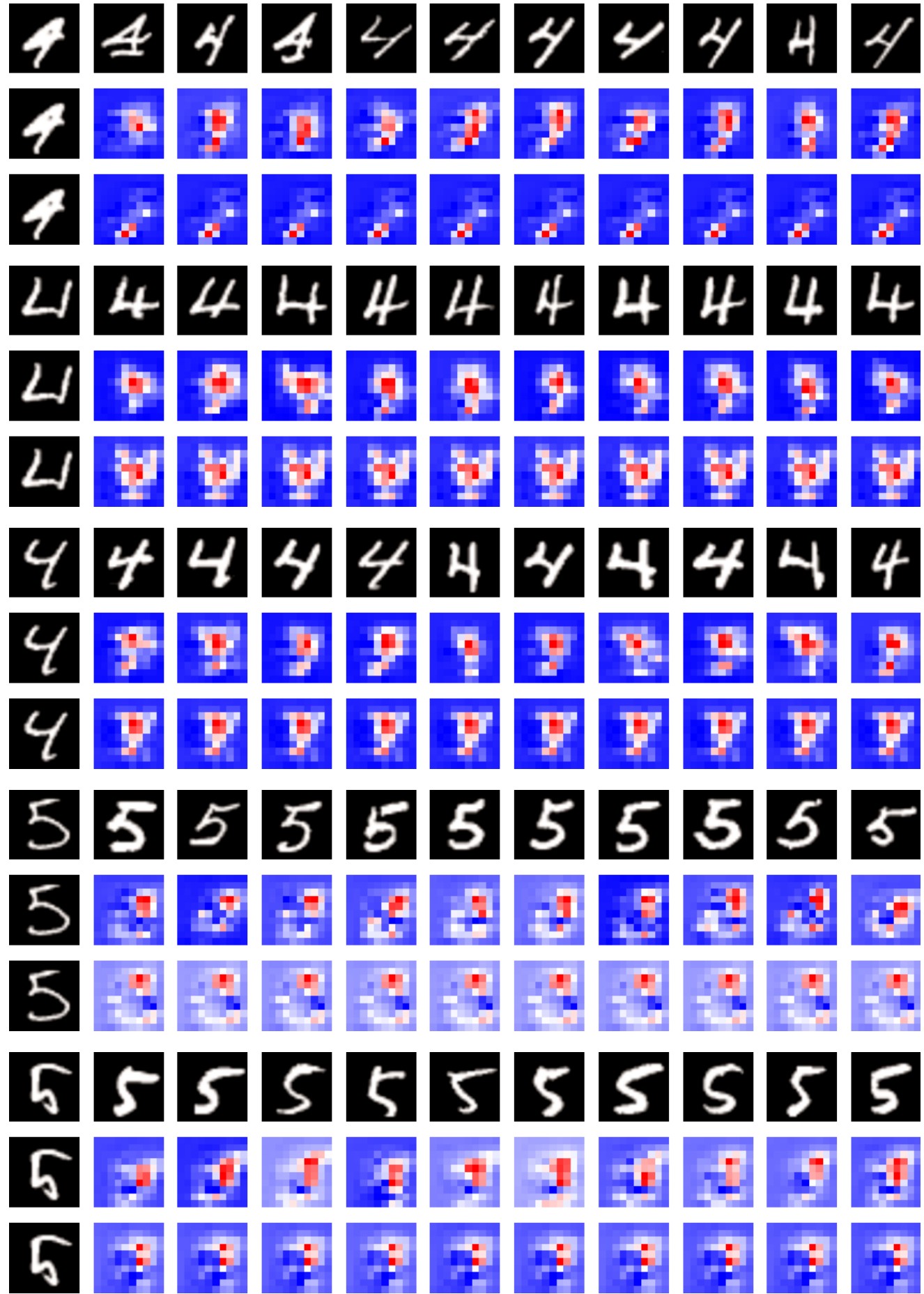

Fig. S14: Explanations for MNIST (set 7).

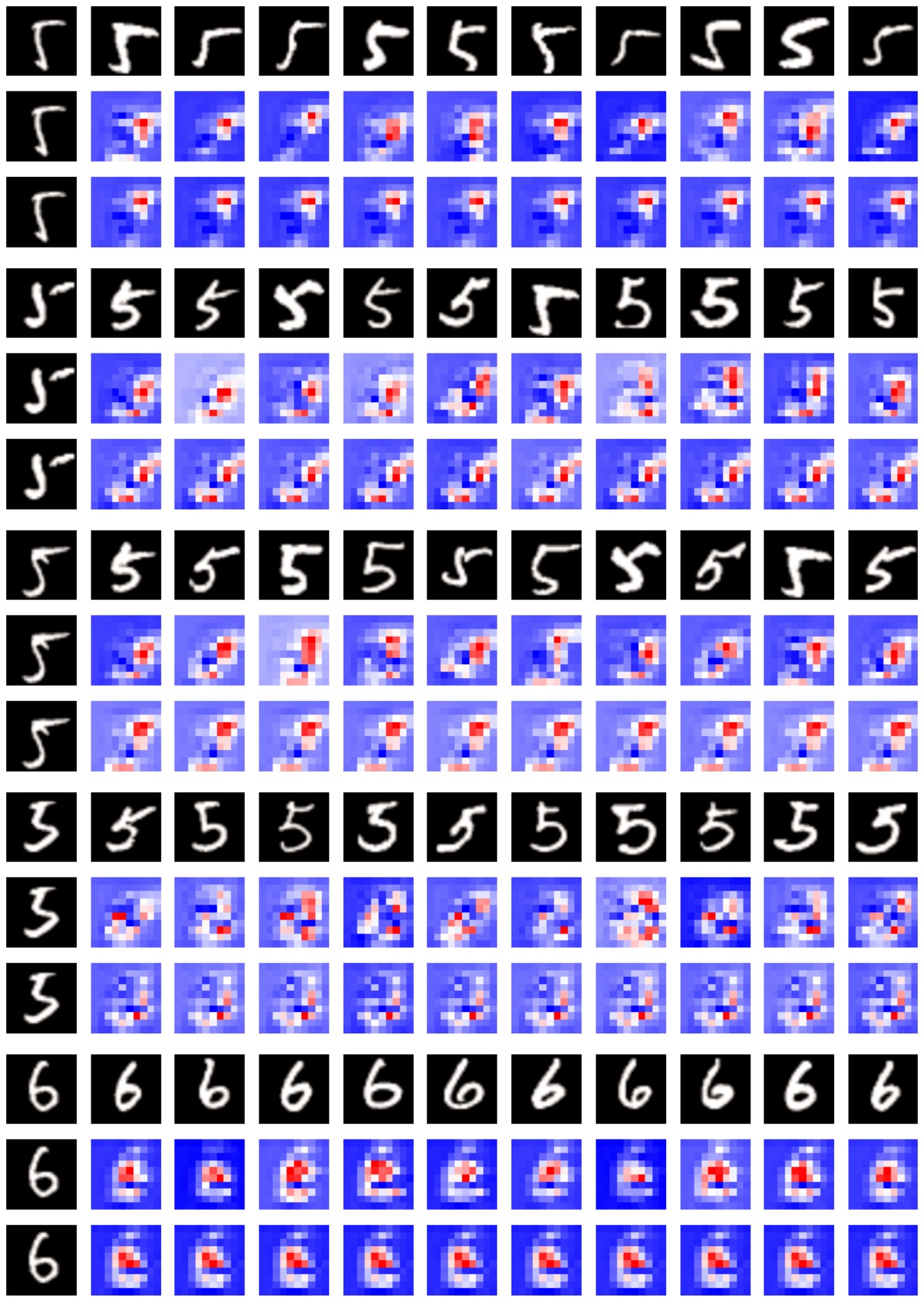

Fig. S15: Explanations for MNIST (set 8).

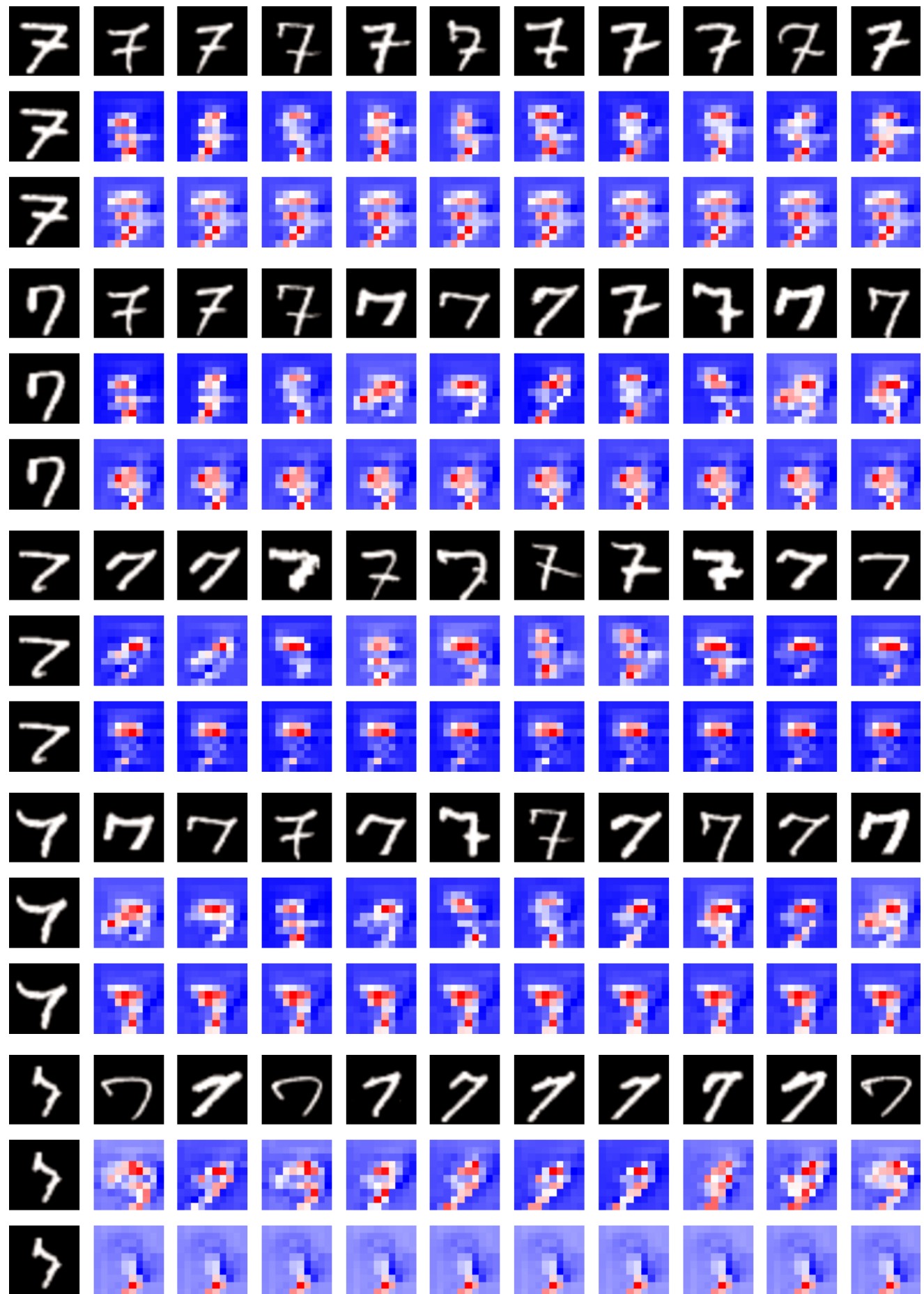

Fig. S16: Explanations for MNIST (set 9).

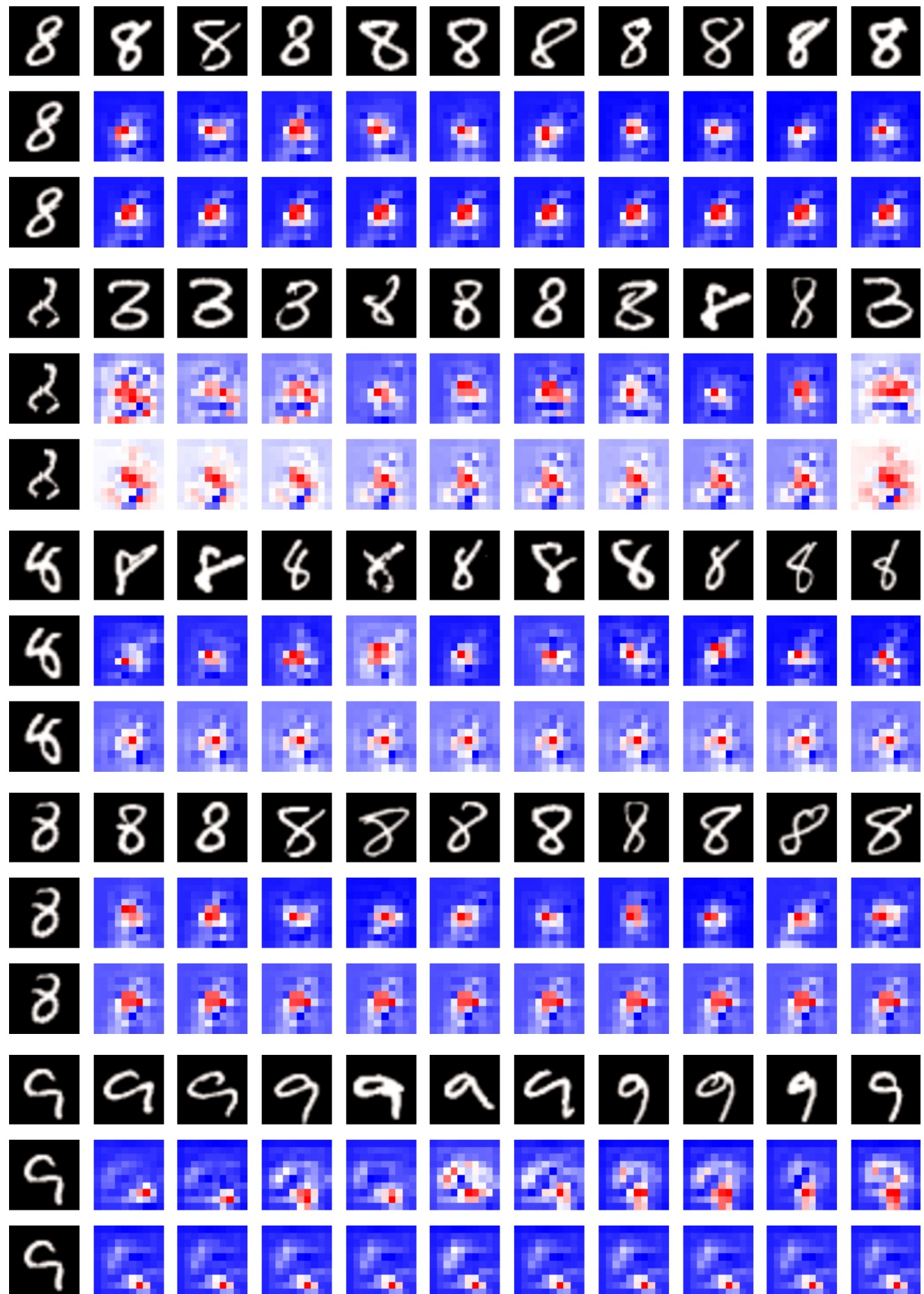

Fig. S17: Explanations for MNIST (set 10).

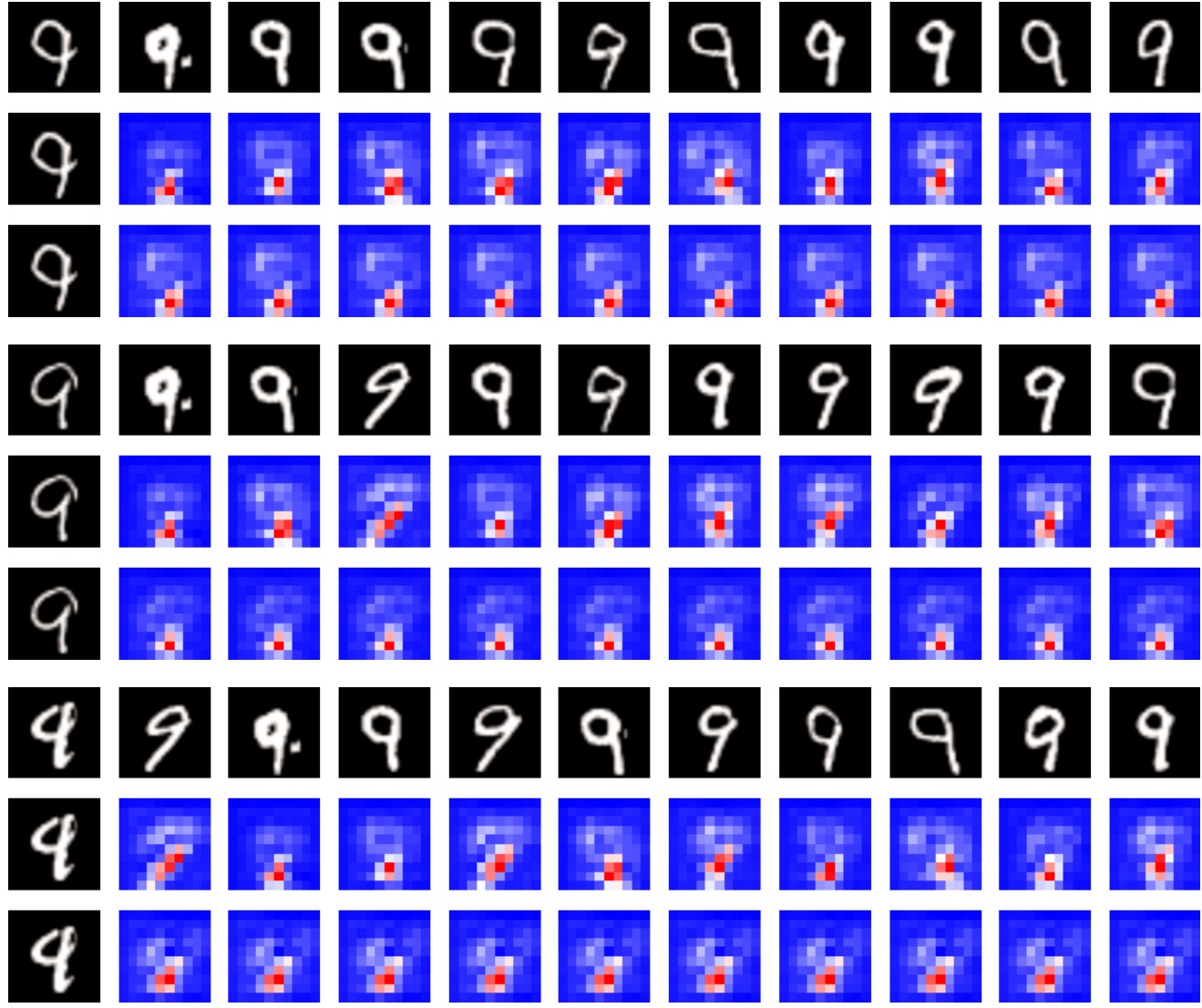

Fig. S18: Explanations for MNIST (set 11).

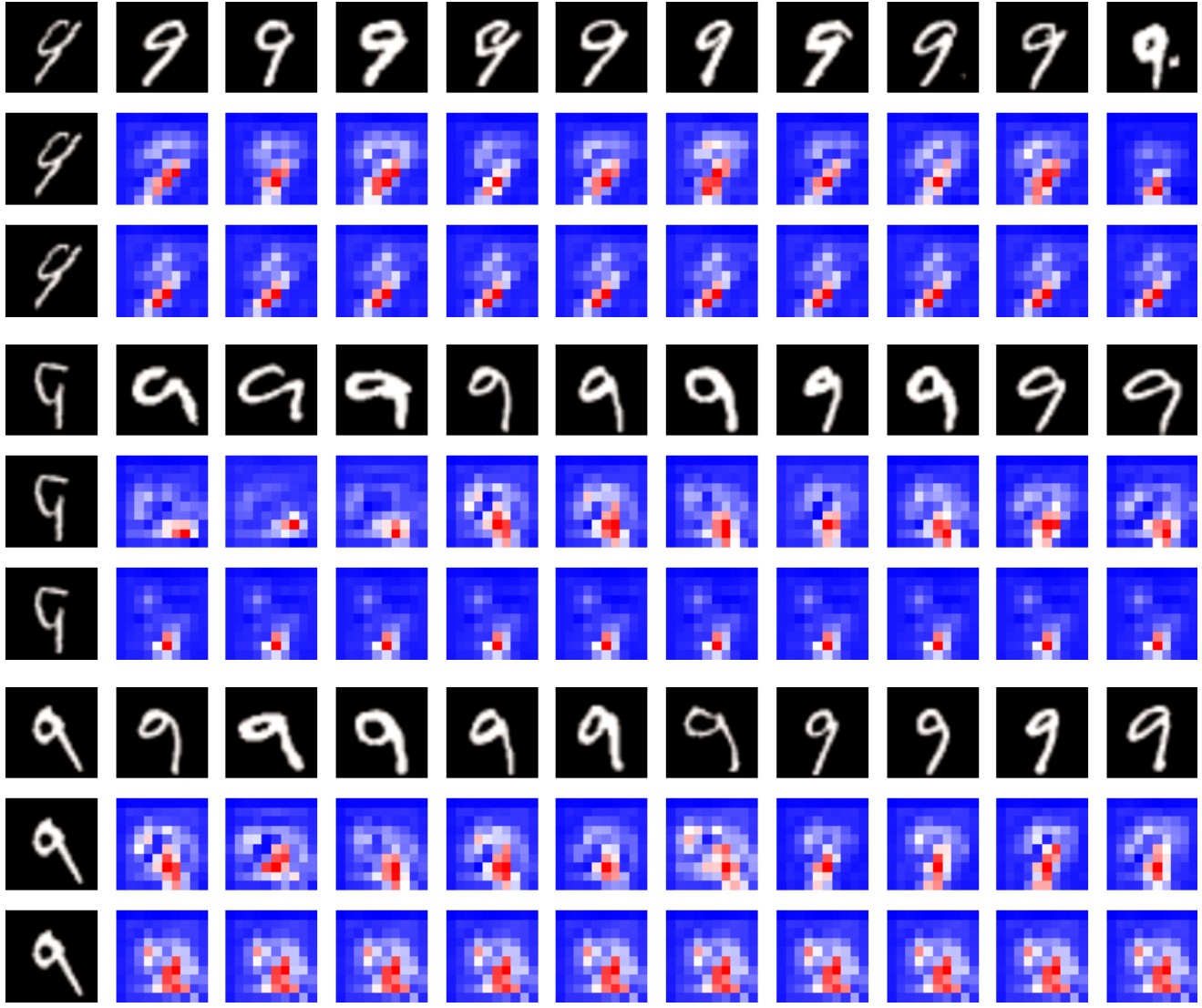

Fig. S19: Explanations for MNIST (set 12).

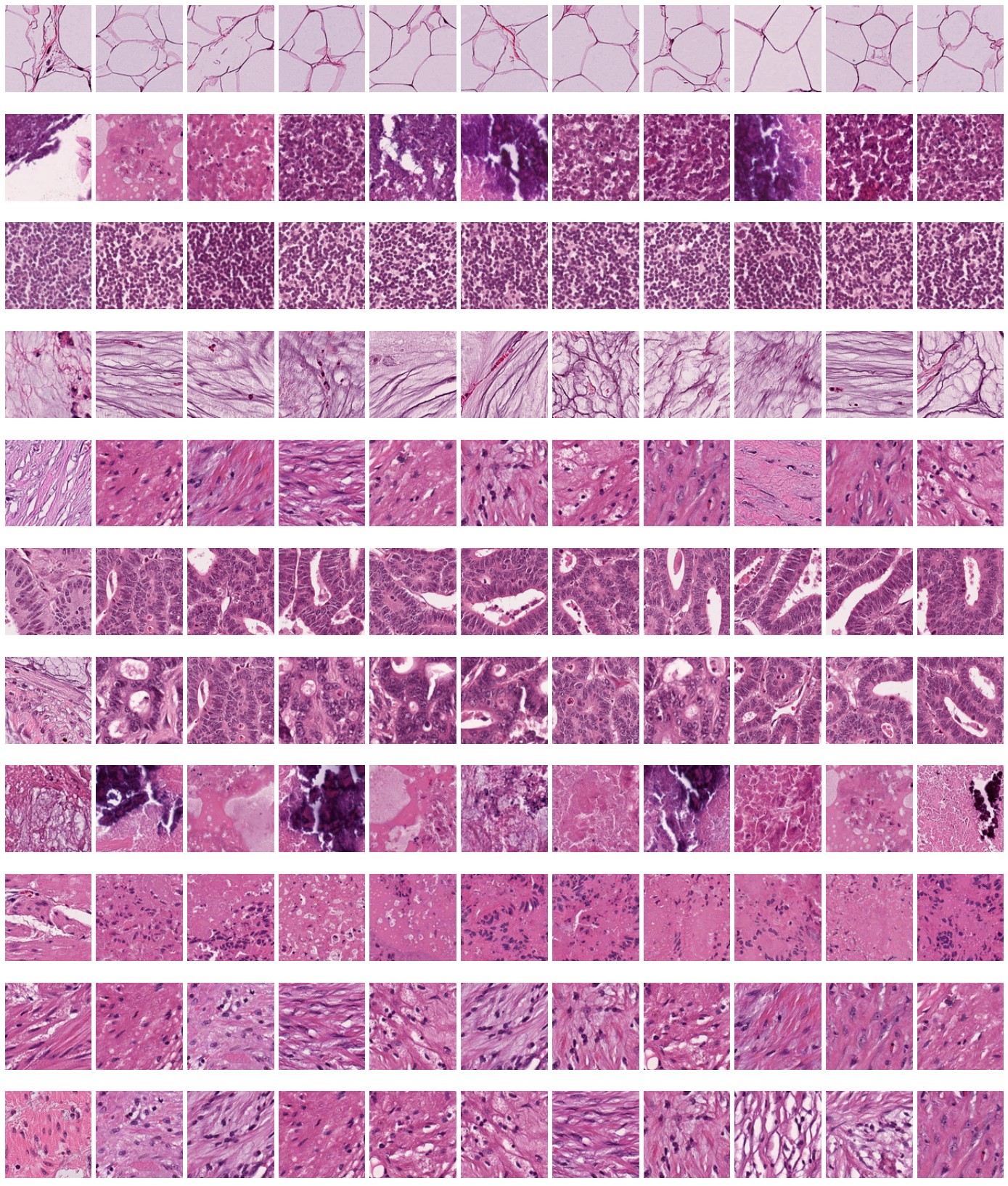

Fig. S20: Explanations for Kather dataset (set 1).

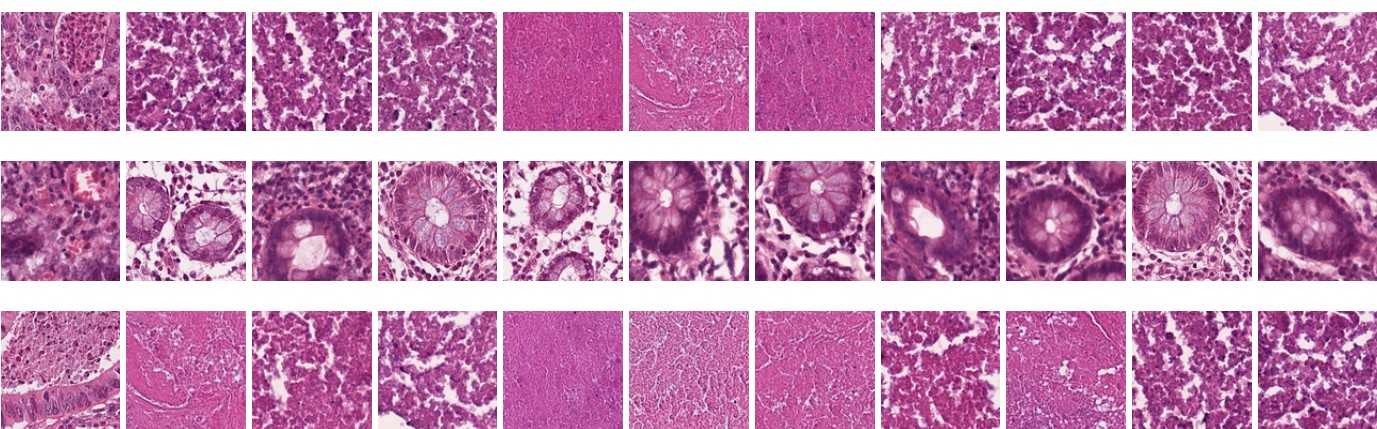

Fig. S21: Explanations for Kather dataset (set 2).

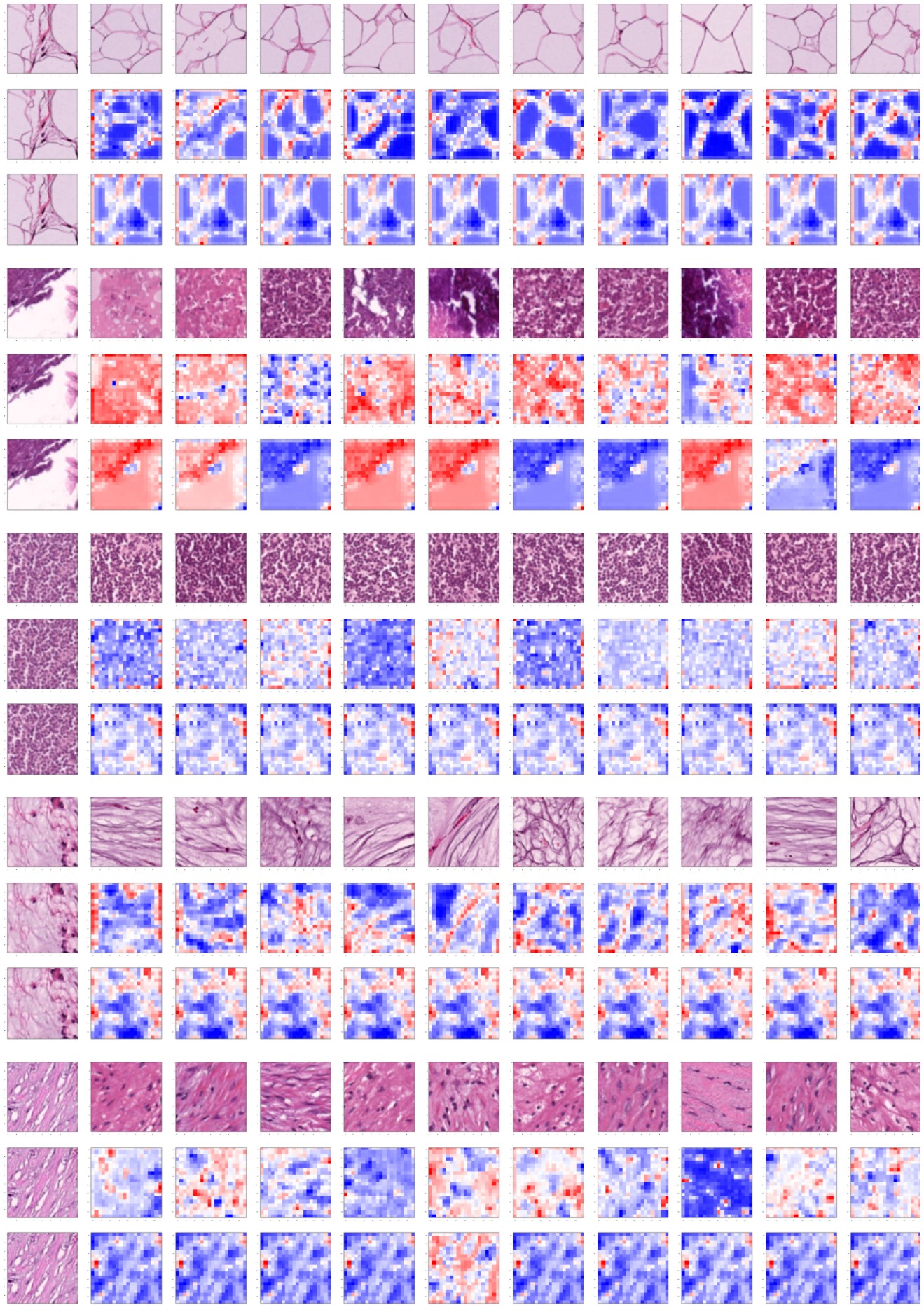

Fig. S22: Explanations for Kather dataset (set 3).

 

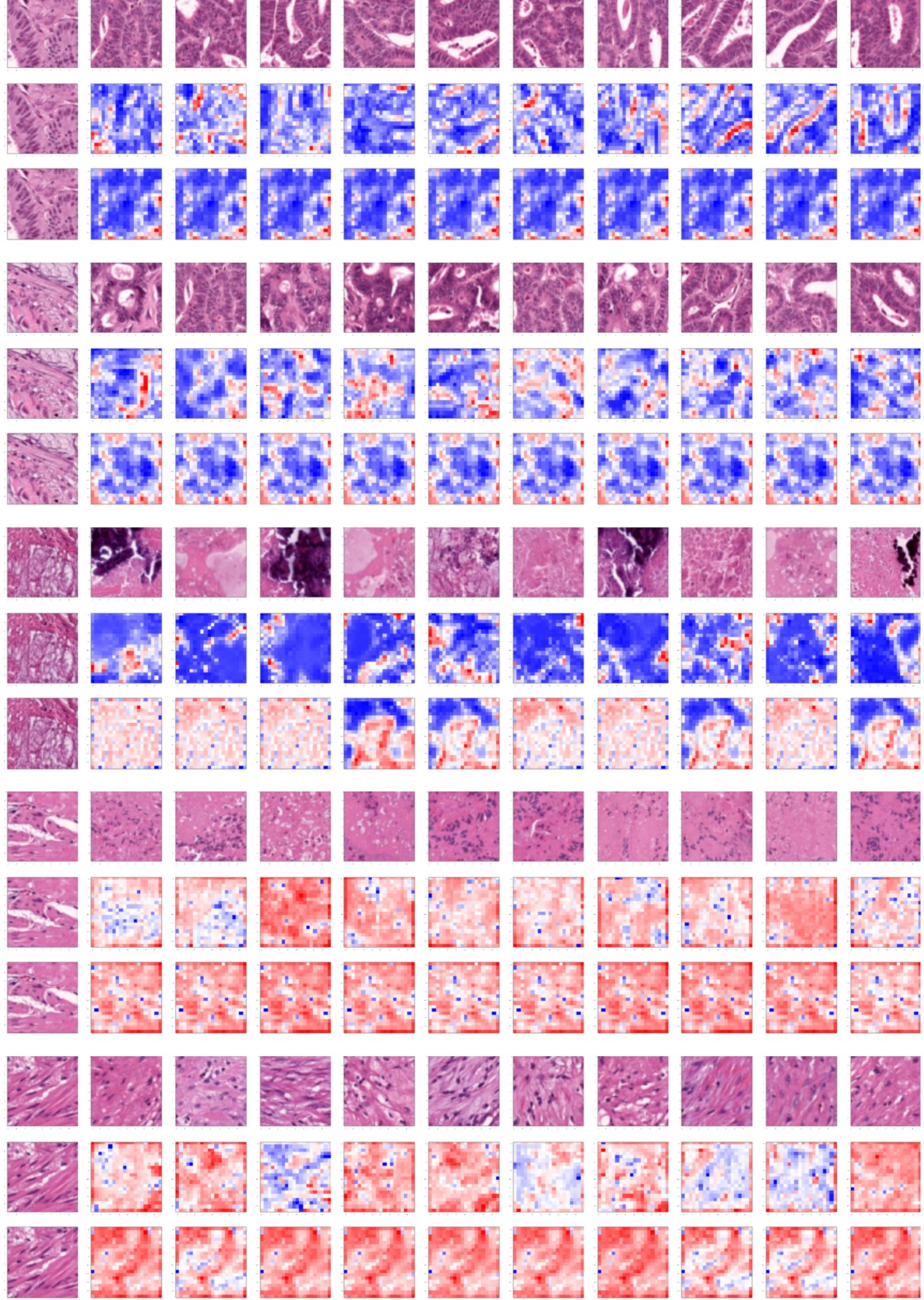

Fig. S23: Explanations for Kather dataset (set 4).

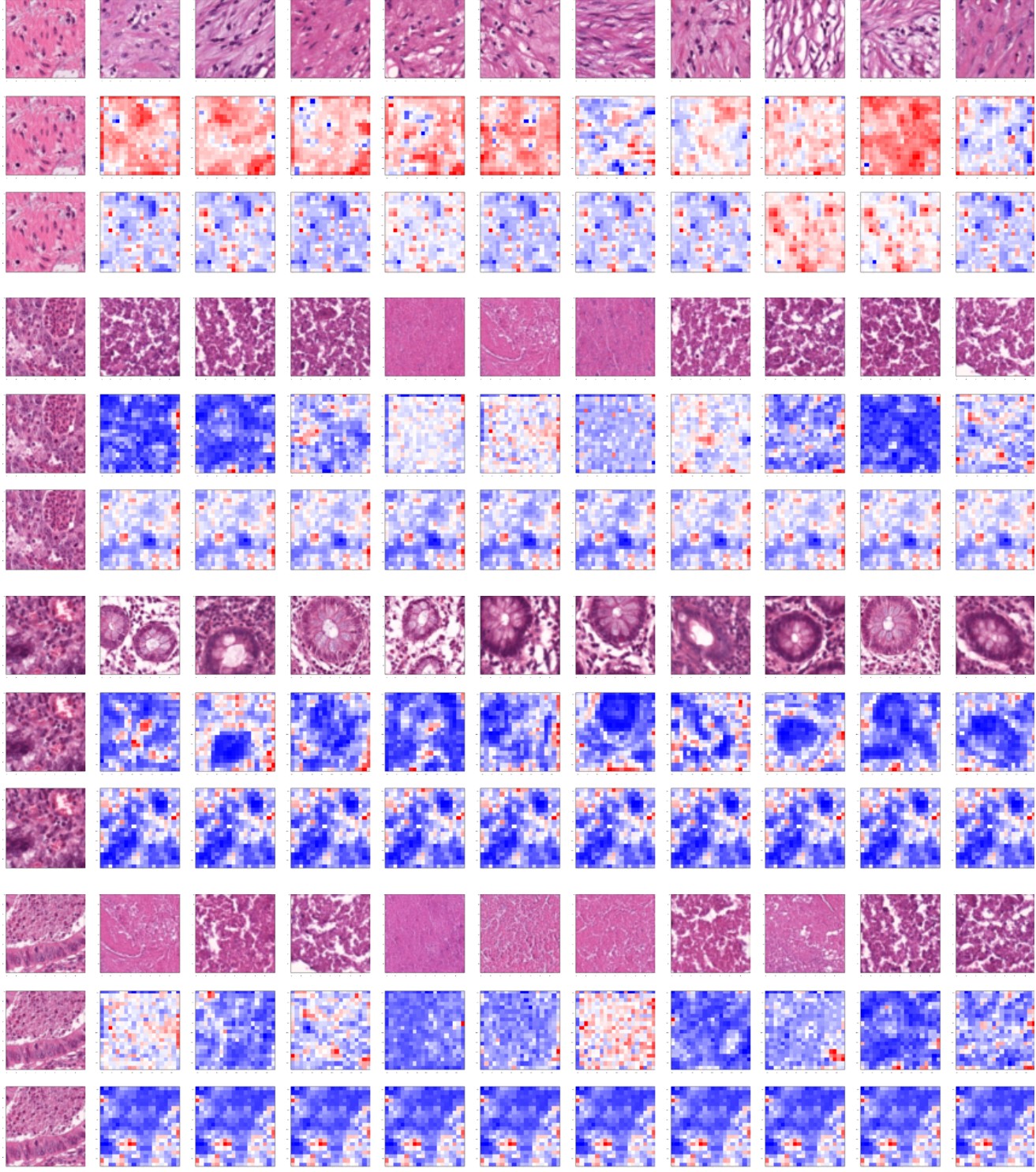

Fig. S24: Explanations for Kather dataset (set 5).

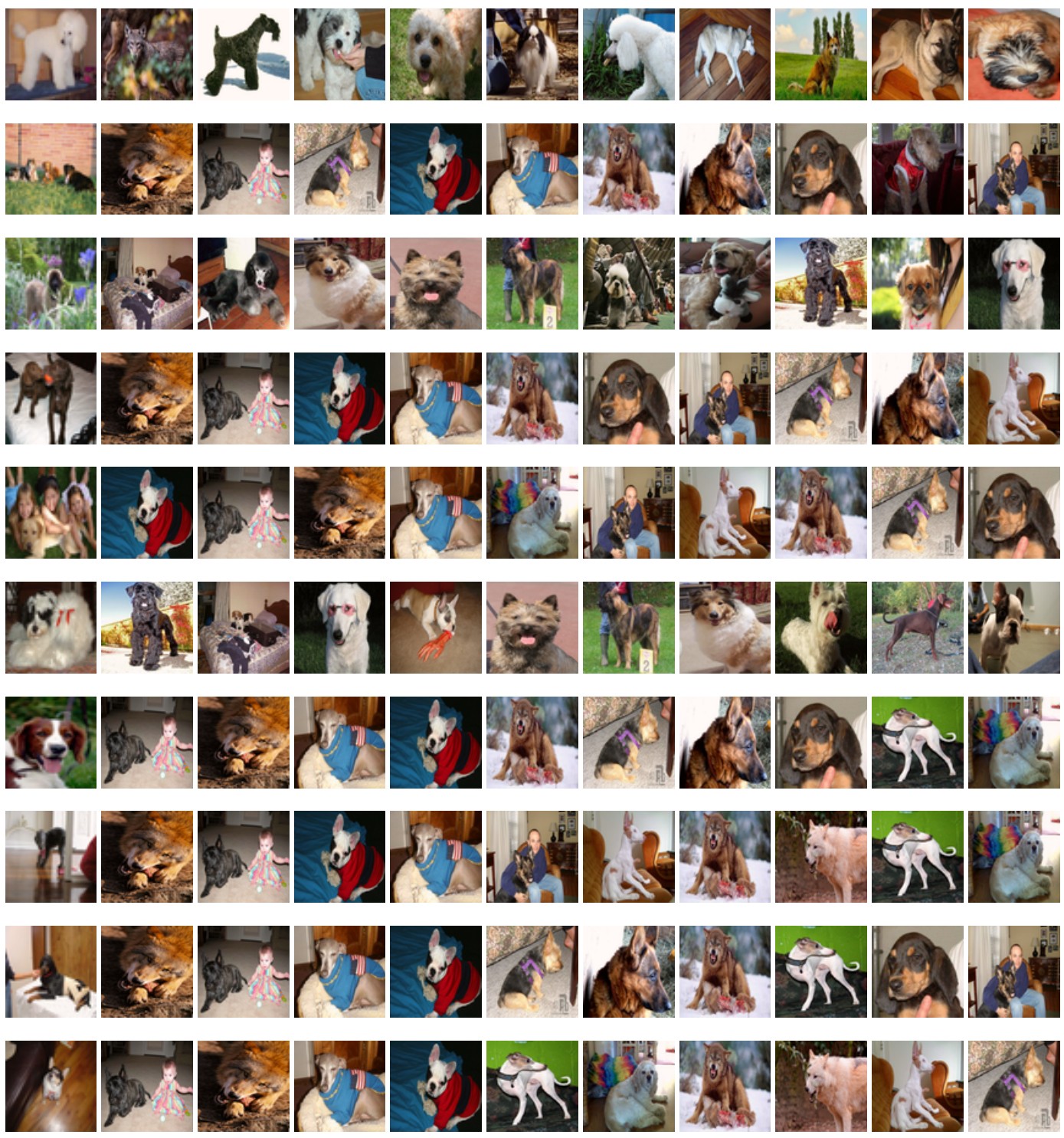

Fig. S25: Explanations for DogsWolves (set 1).

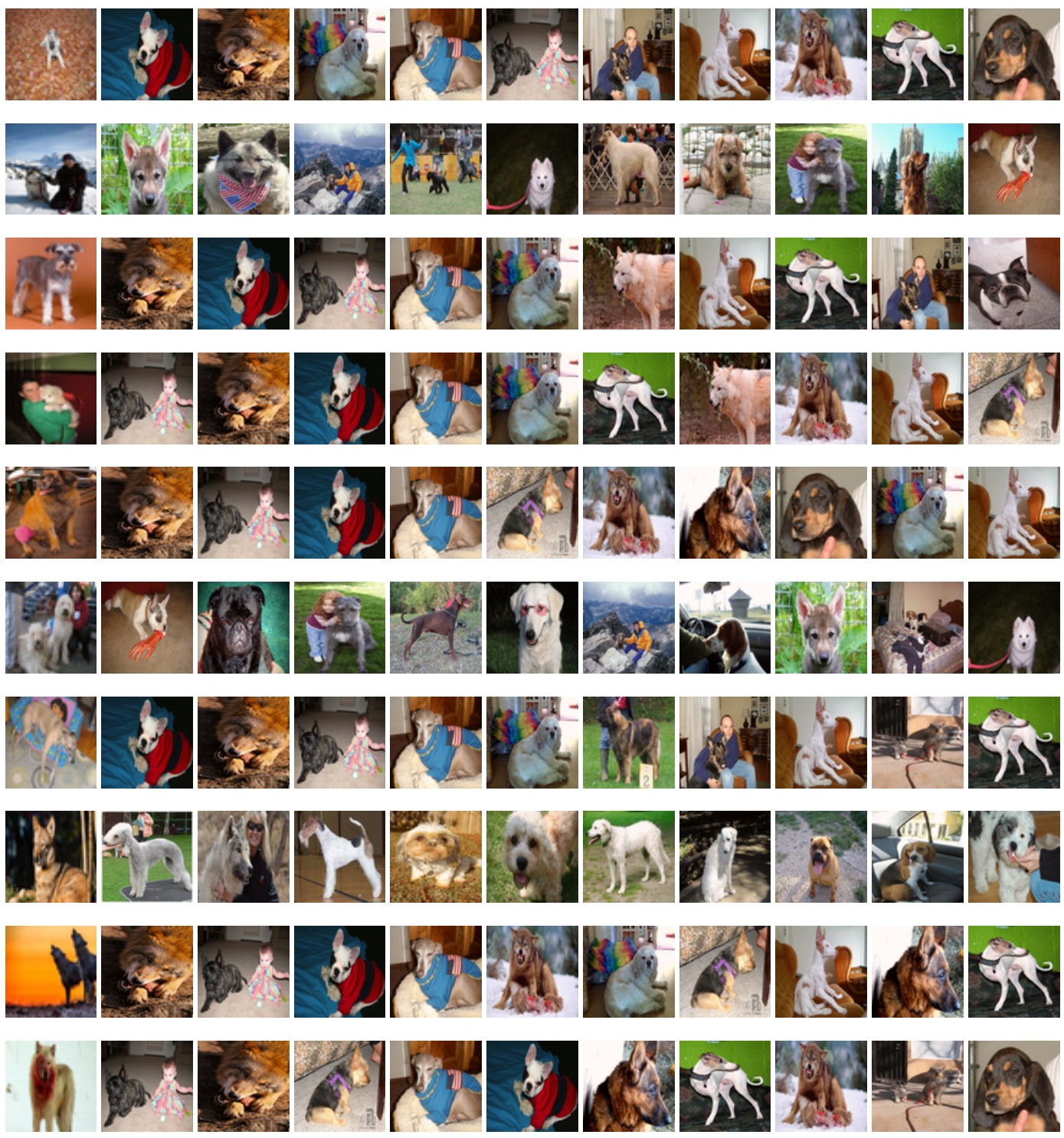

Fig. S26: Explanations for DogsWolves (set 2).

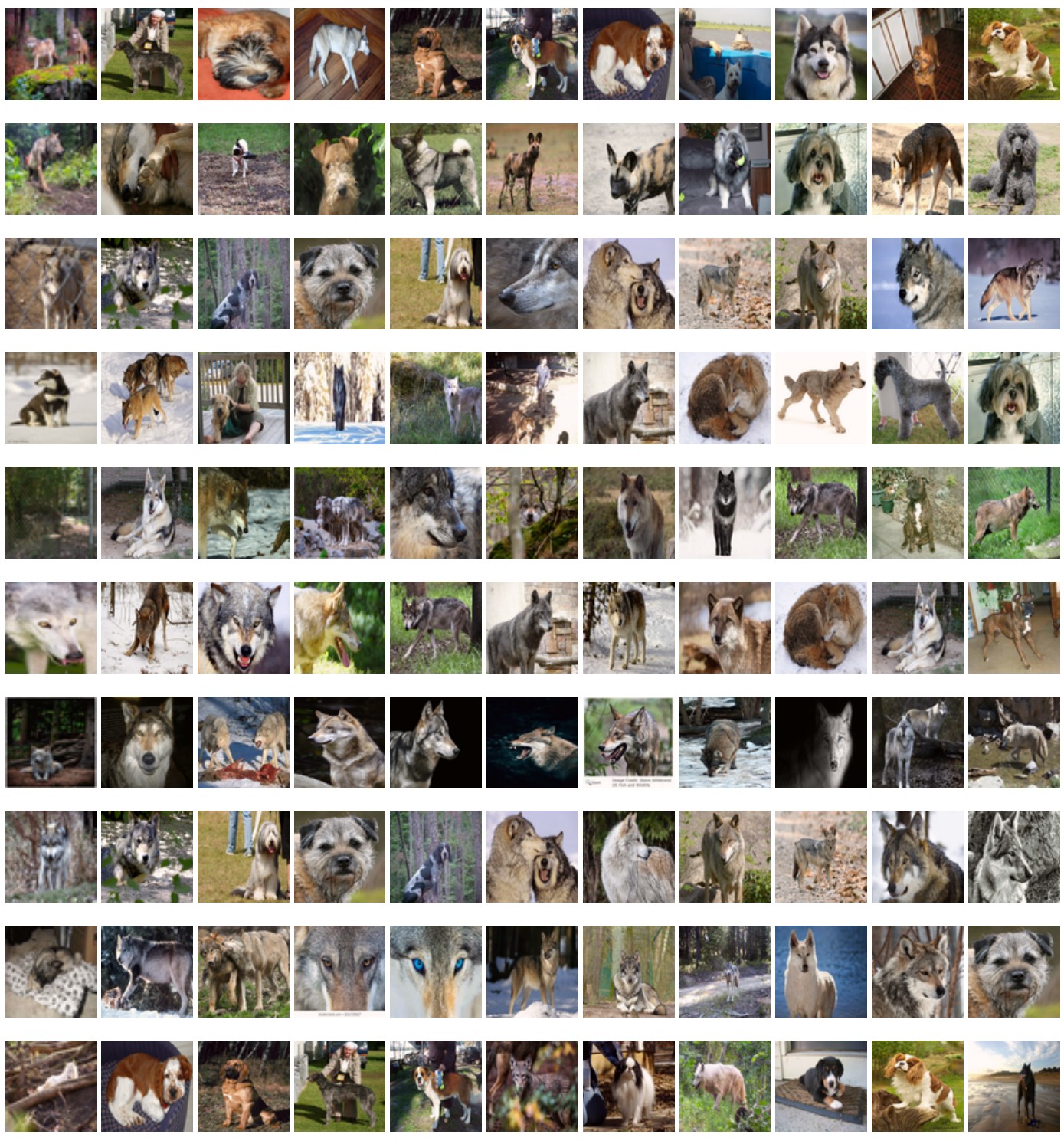

Fig. S27: Explanations for DogsWolves (set 3).

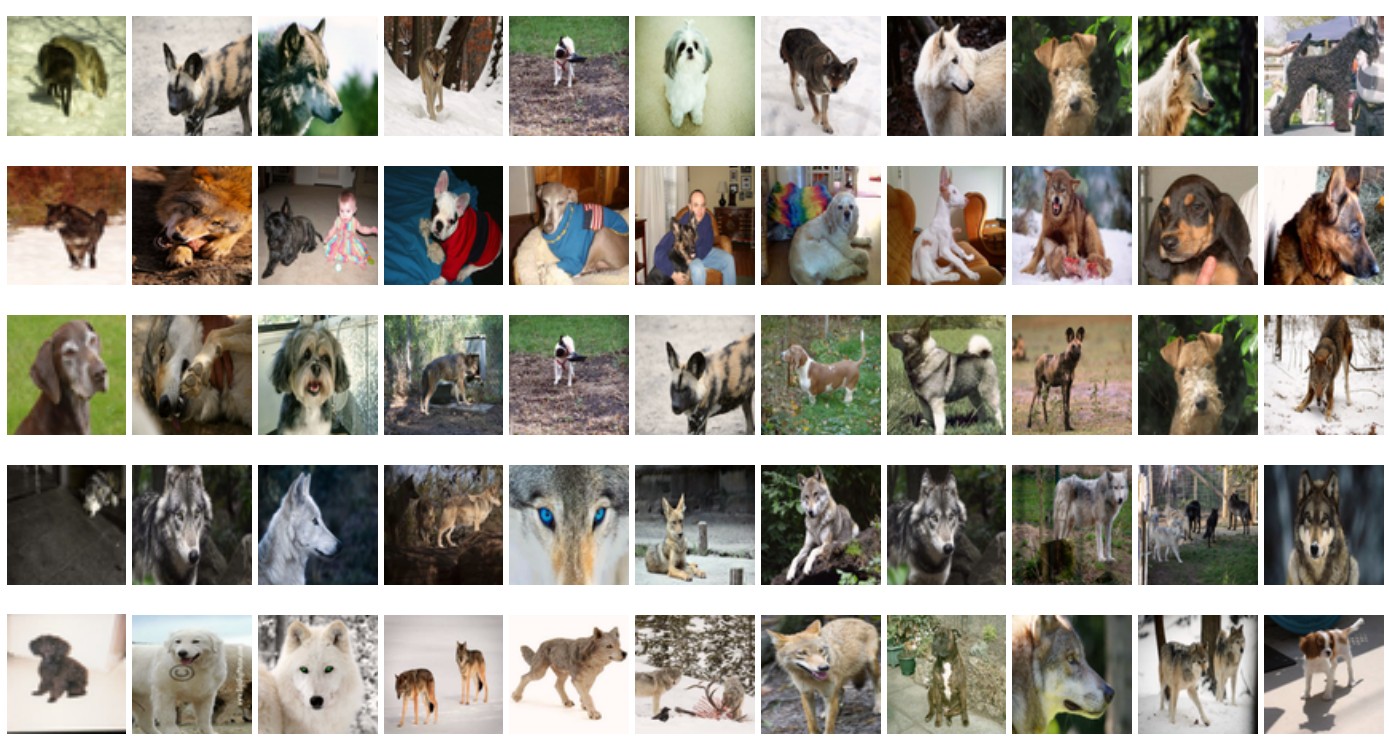

Fig. S28: Explanations for DogsWolves (set 4).

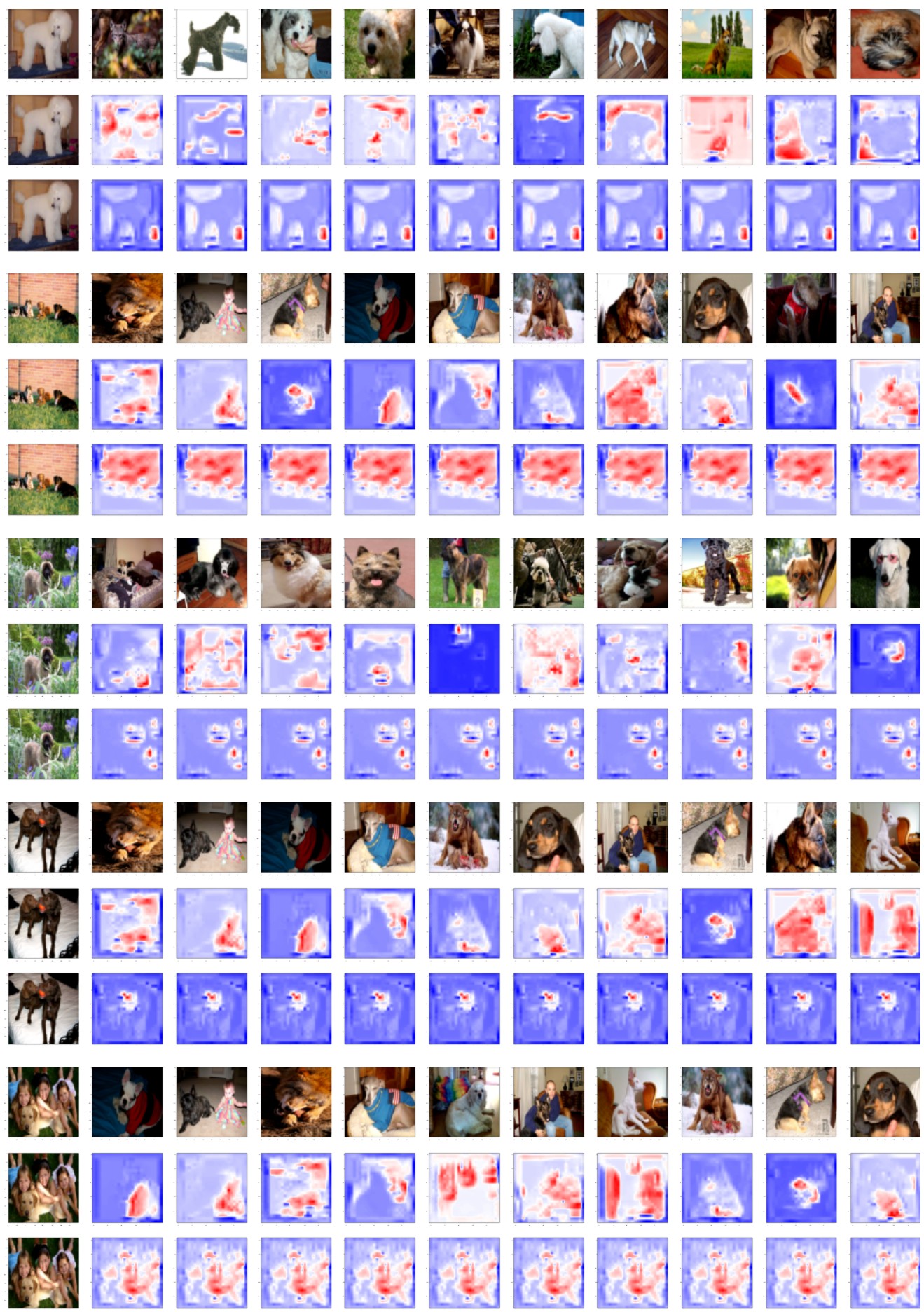

Fig. S29: Explanations for DogsWolves (set 5).

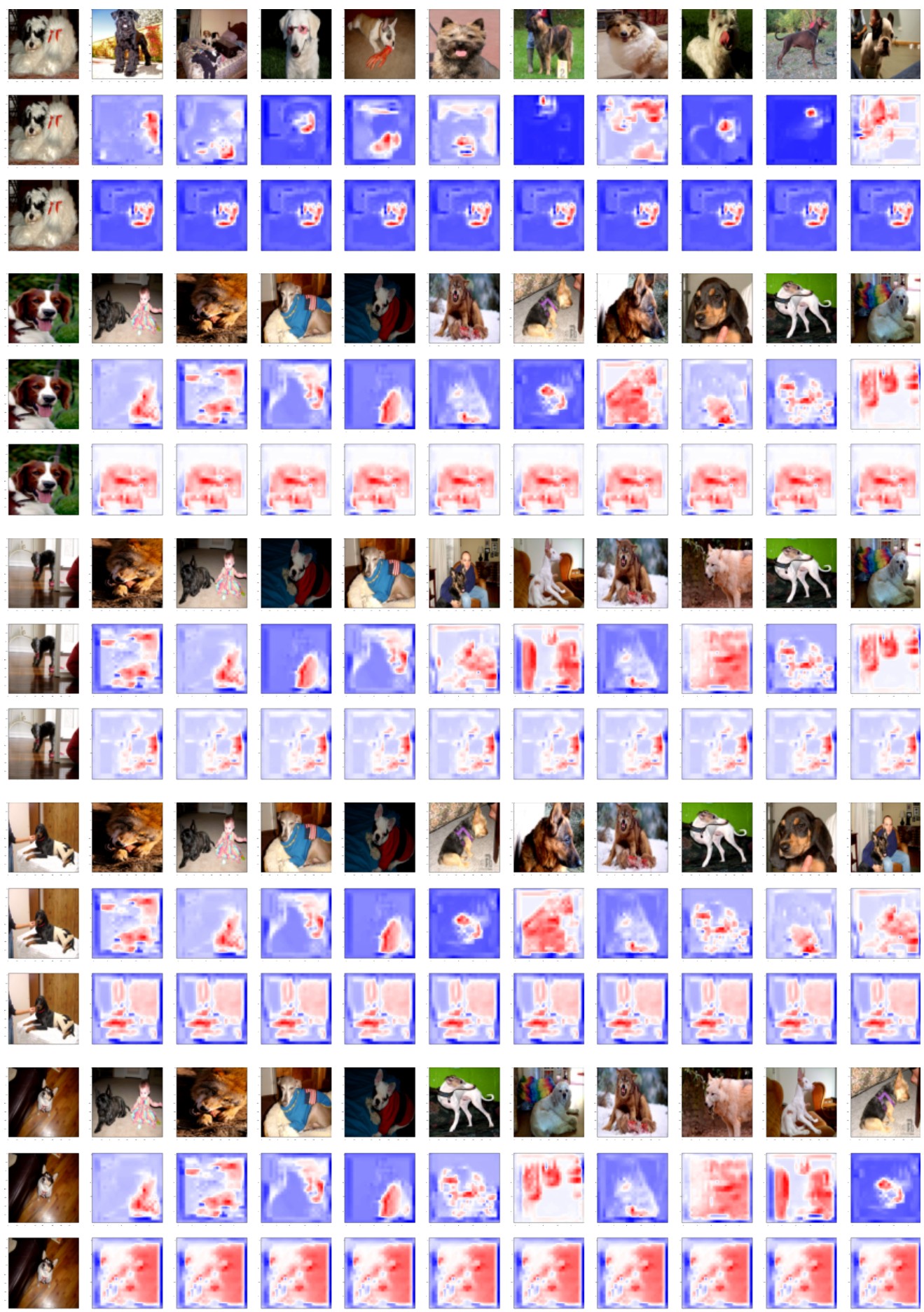

Fig. S30: Explanations for DogsWolves (set 6).

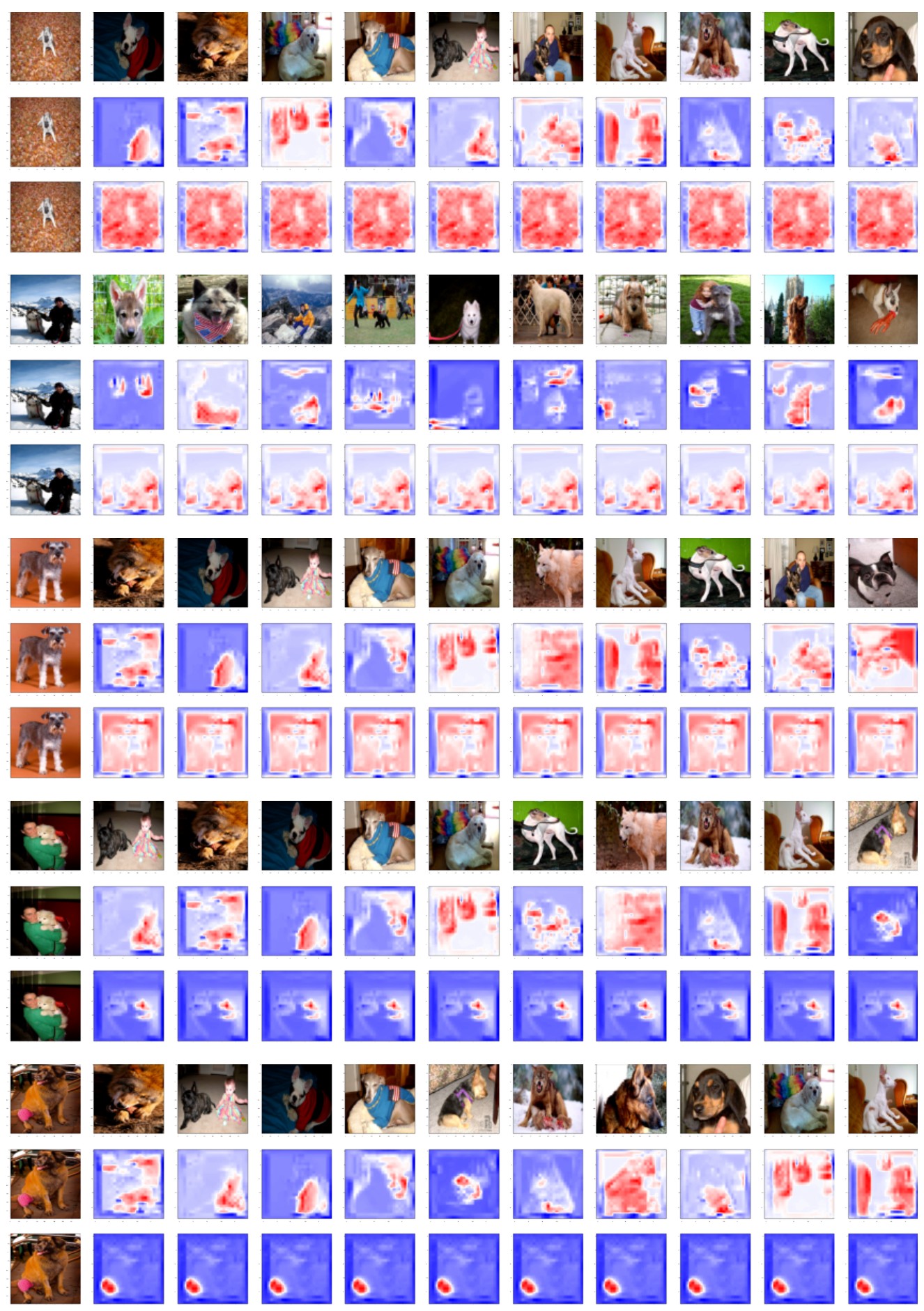

Fig. S31: Explanations for DogsWolves (set 7).

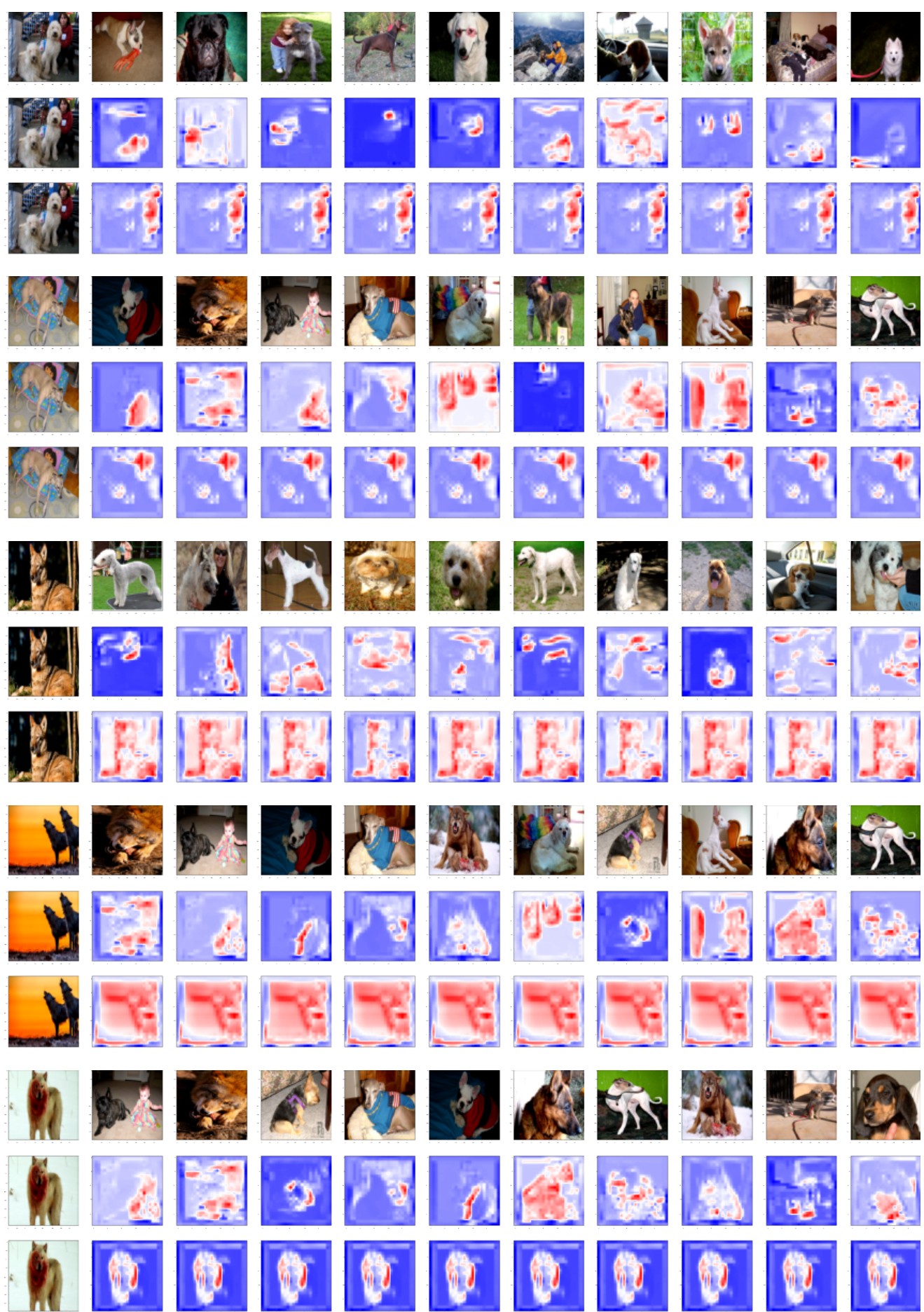

Fig. S32: Explanations for DogsWolves (set 8).

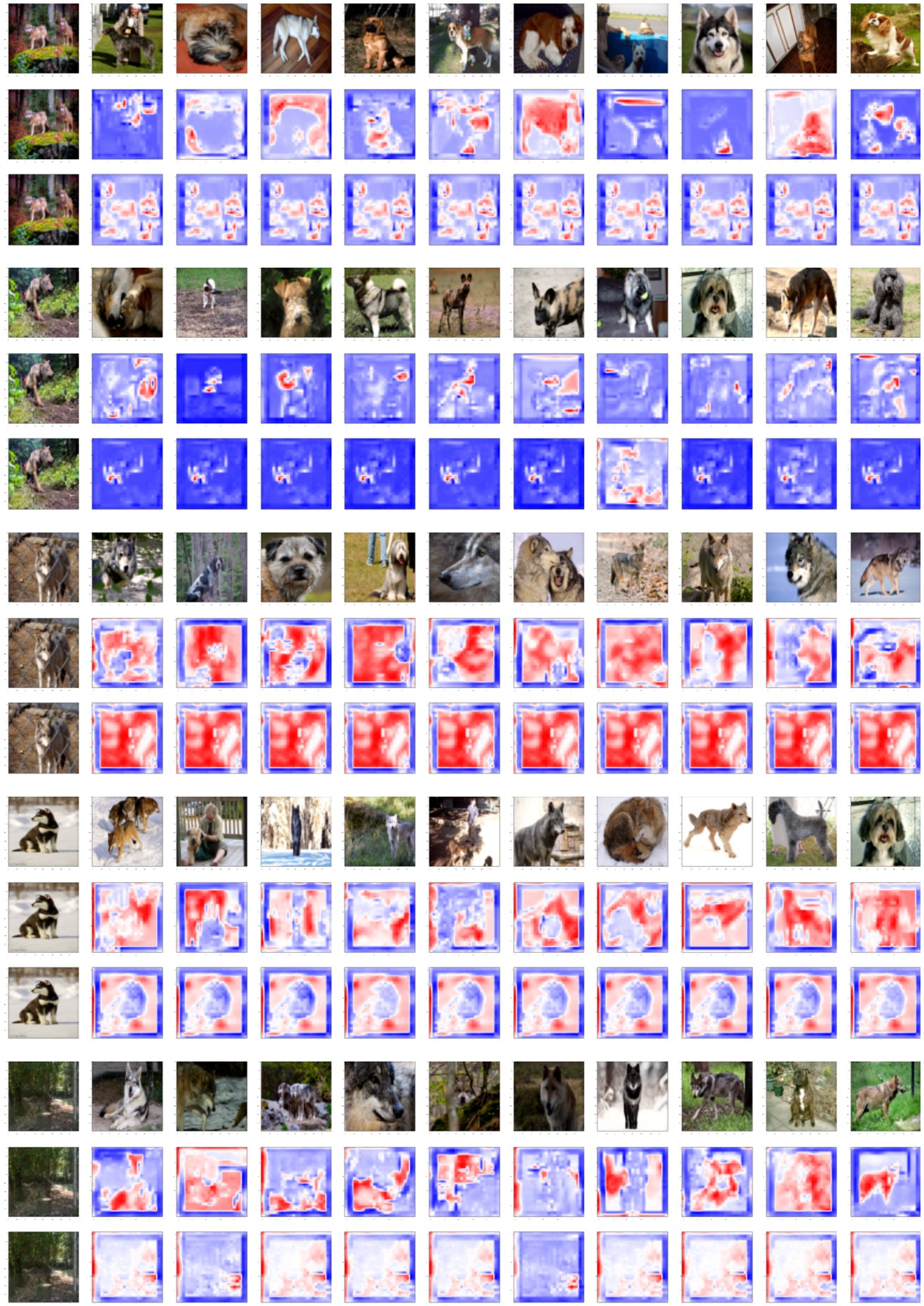

Fig. S33: Explanations for DogsWolves (set 9).

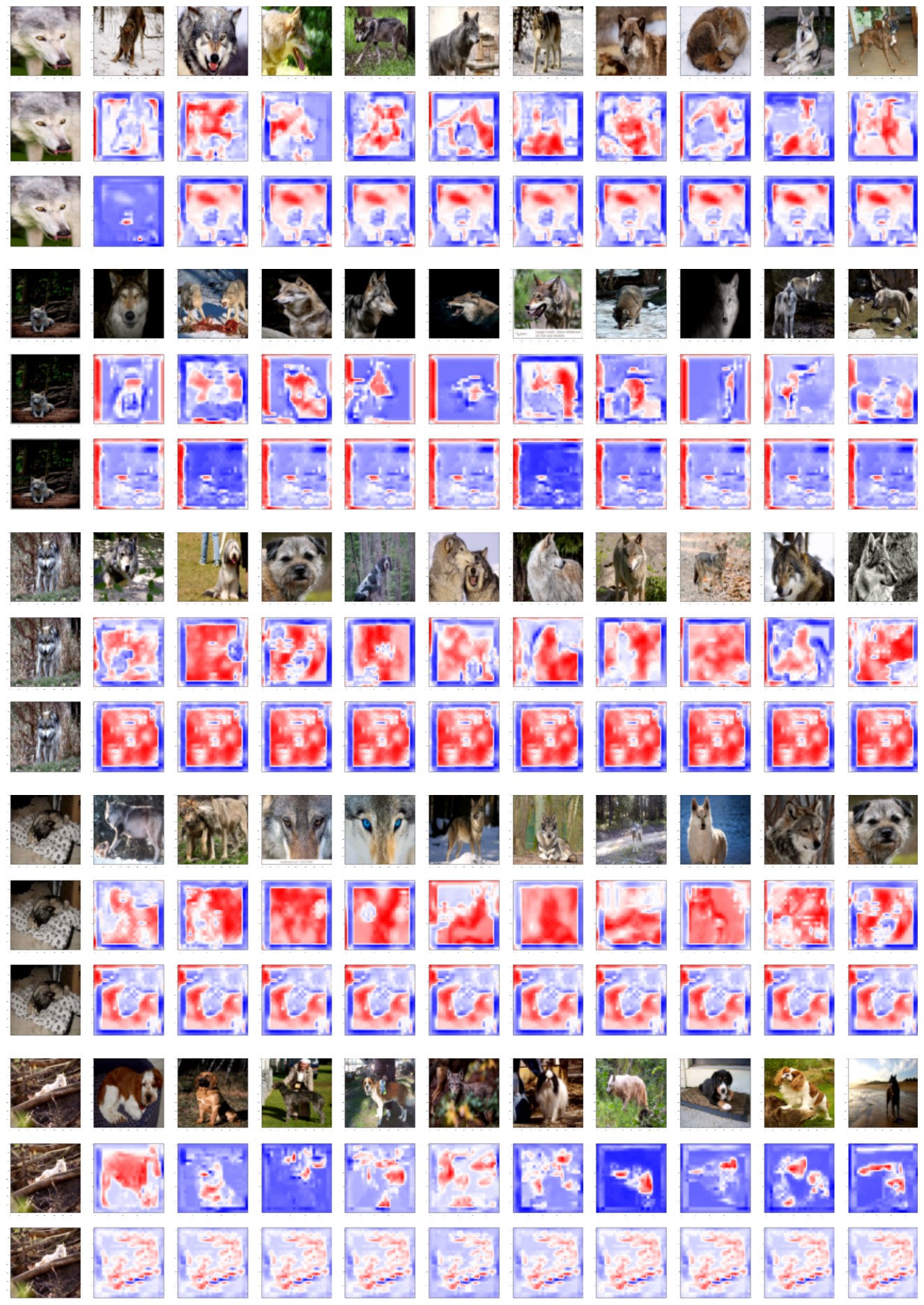

Fig. S34: Explanations for DogsWolves (set 10).

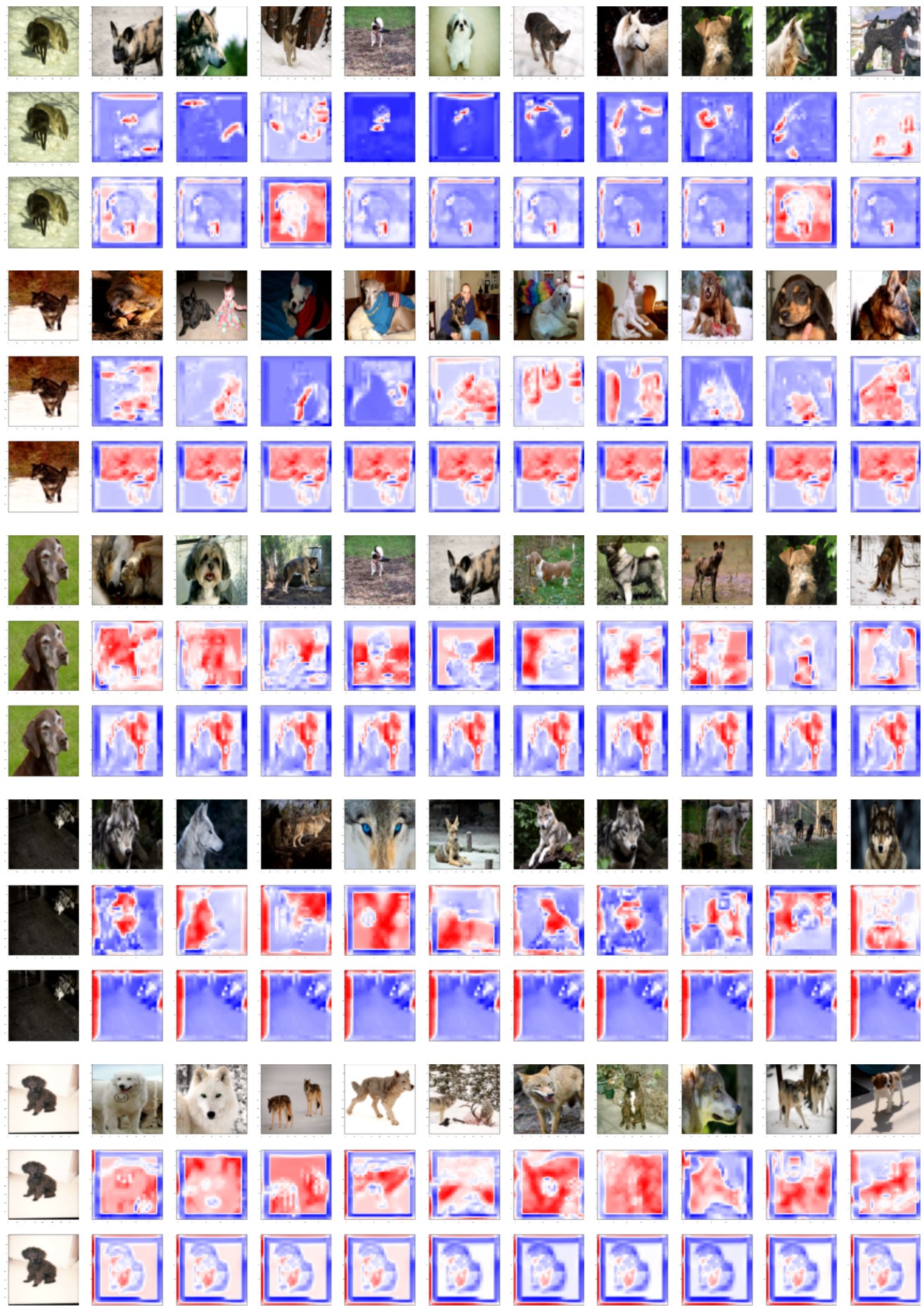

Fig. S35: Explanations for DogsWolves (set 11).

 

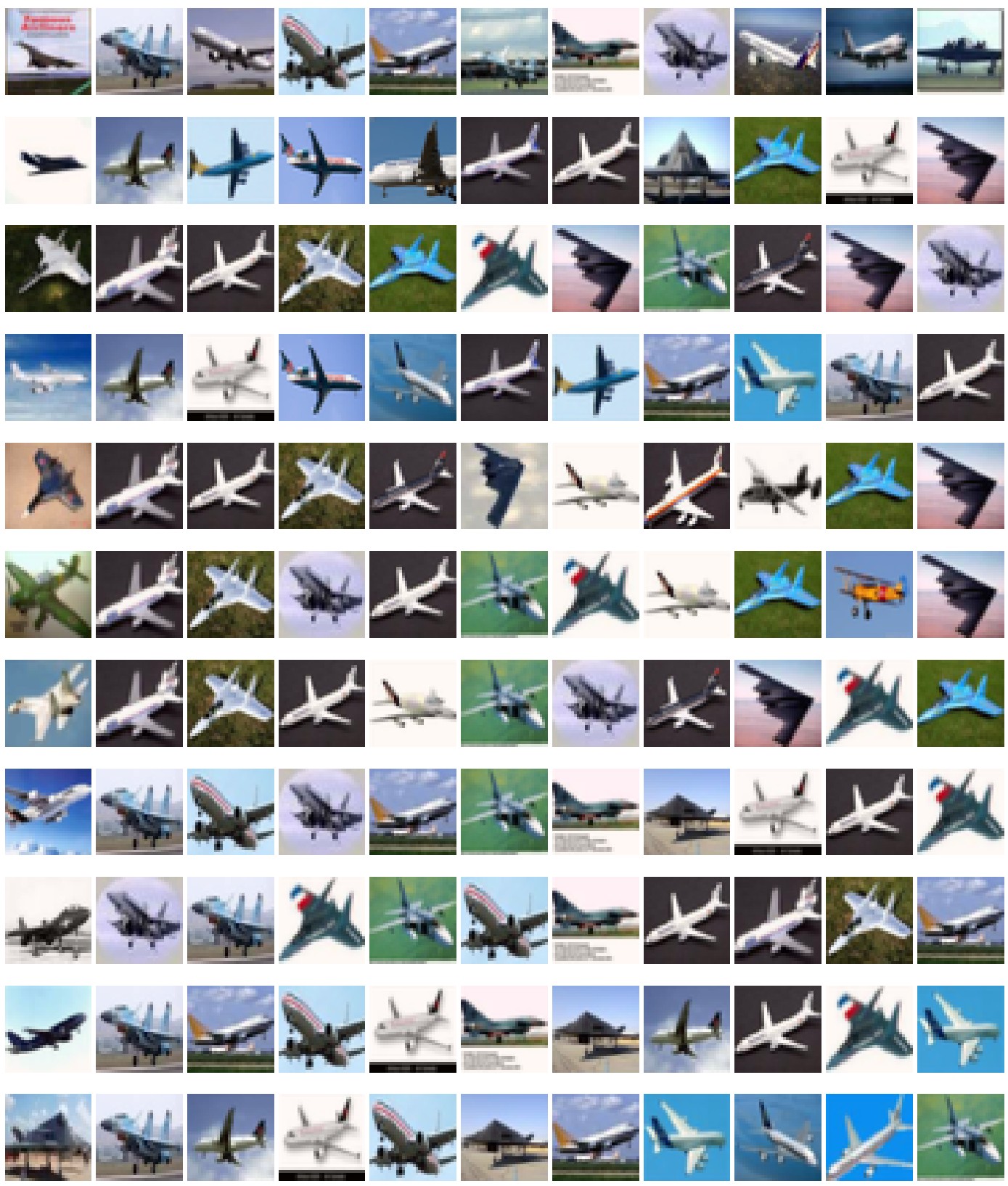

Fig. S36: Explanations for Cifar10 (set 1).

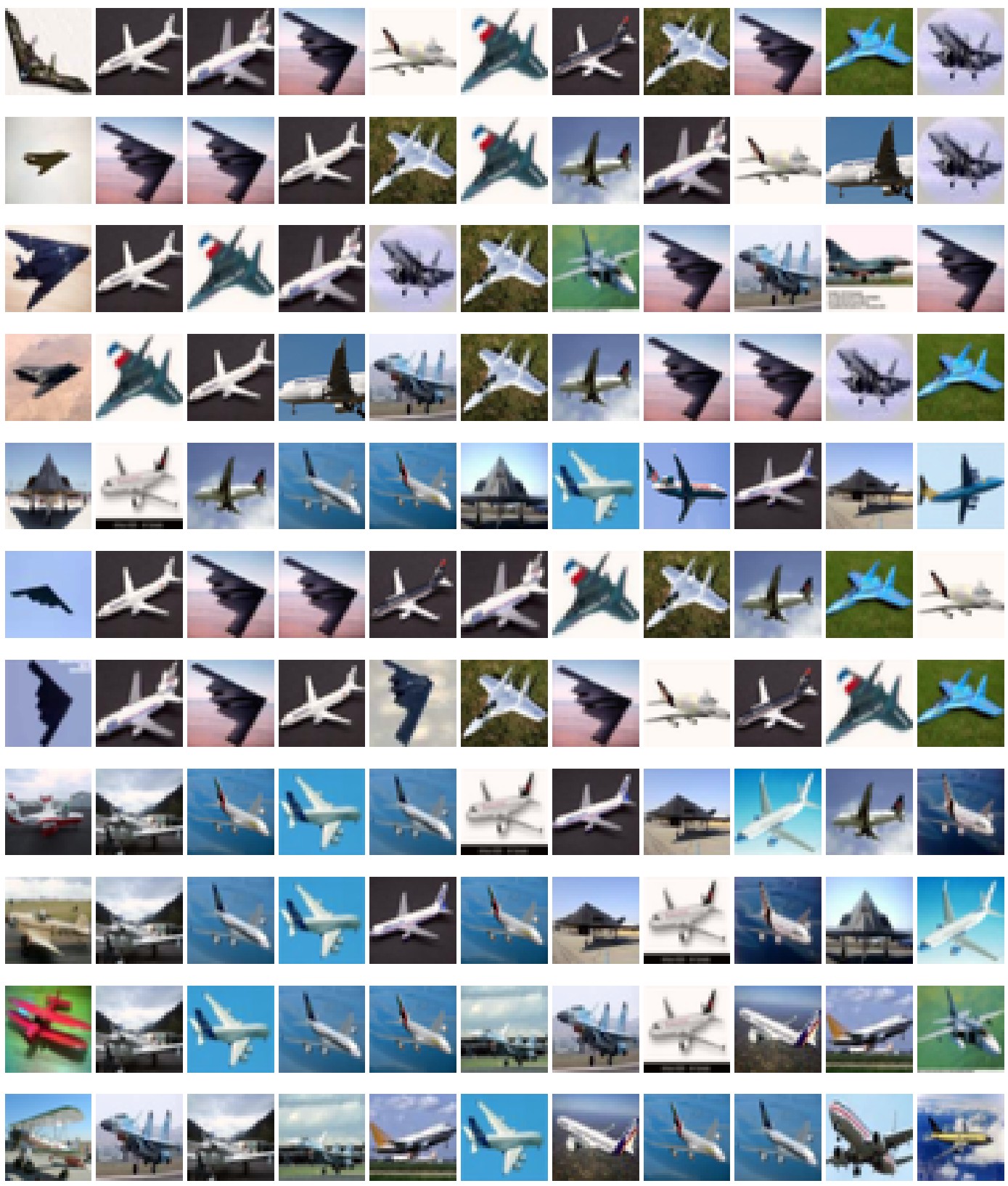

Fig. S37: Explanations for Cifar10 (set 2).

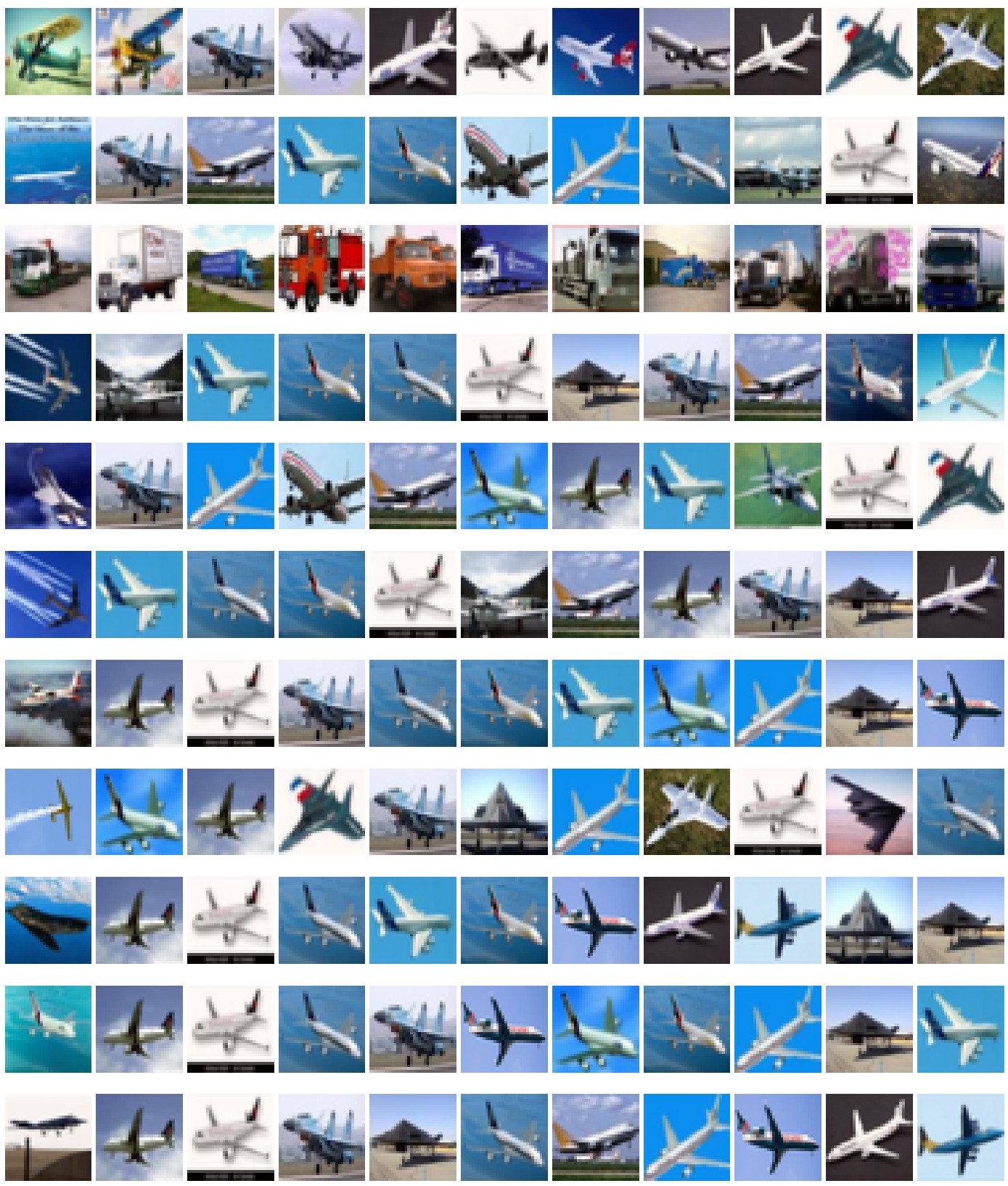

Fig. S38: Explanations for Cifar10 (set 3).

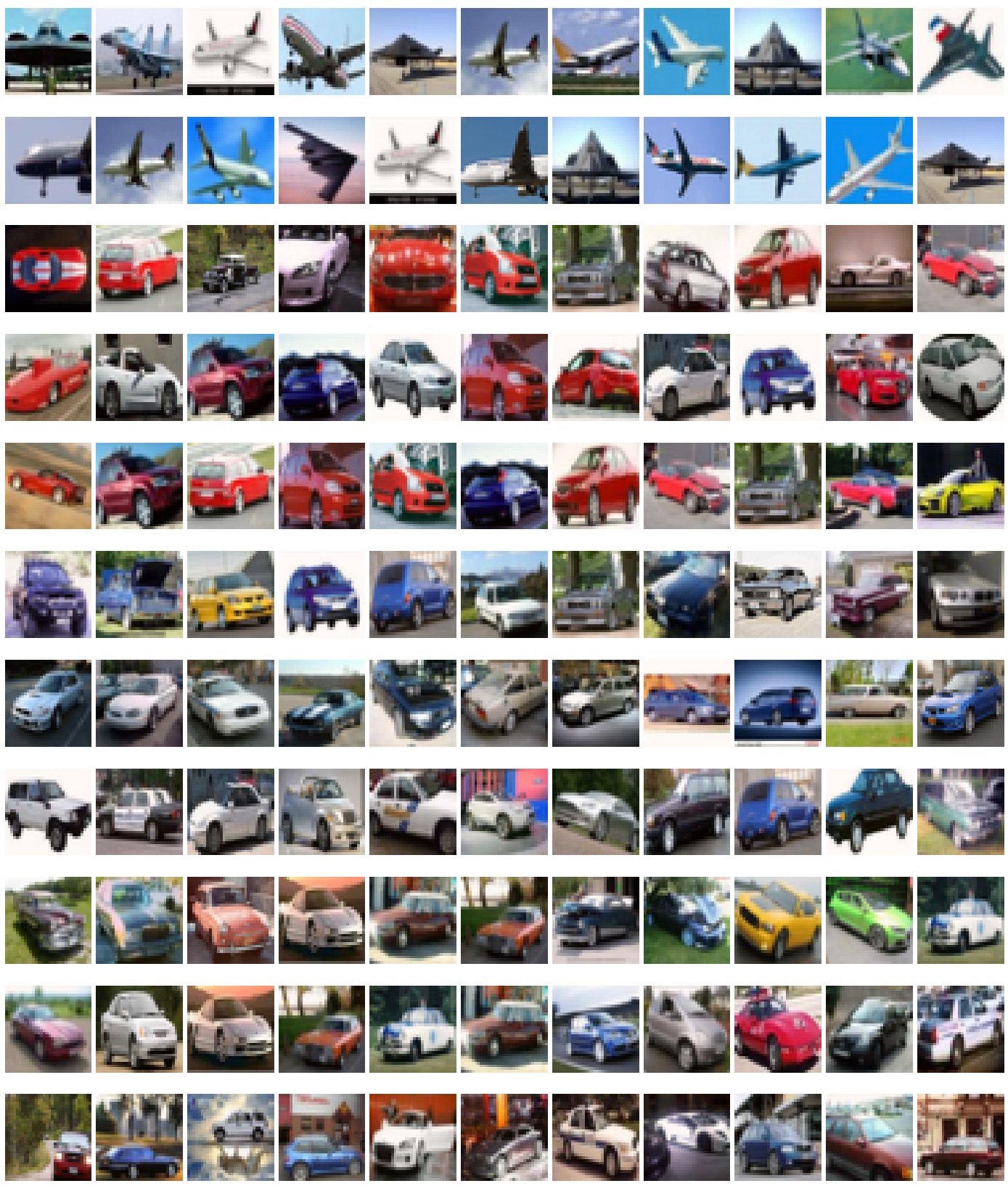

Fig. S39: Explanations for Cifar10 (set 4).

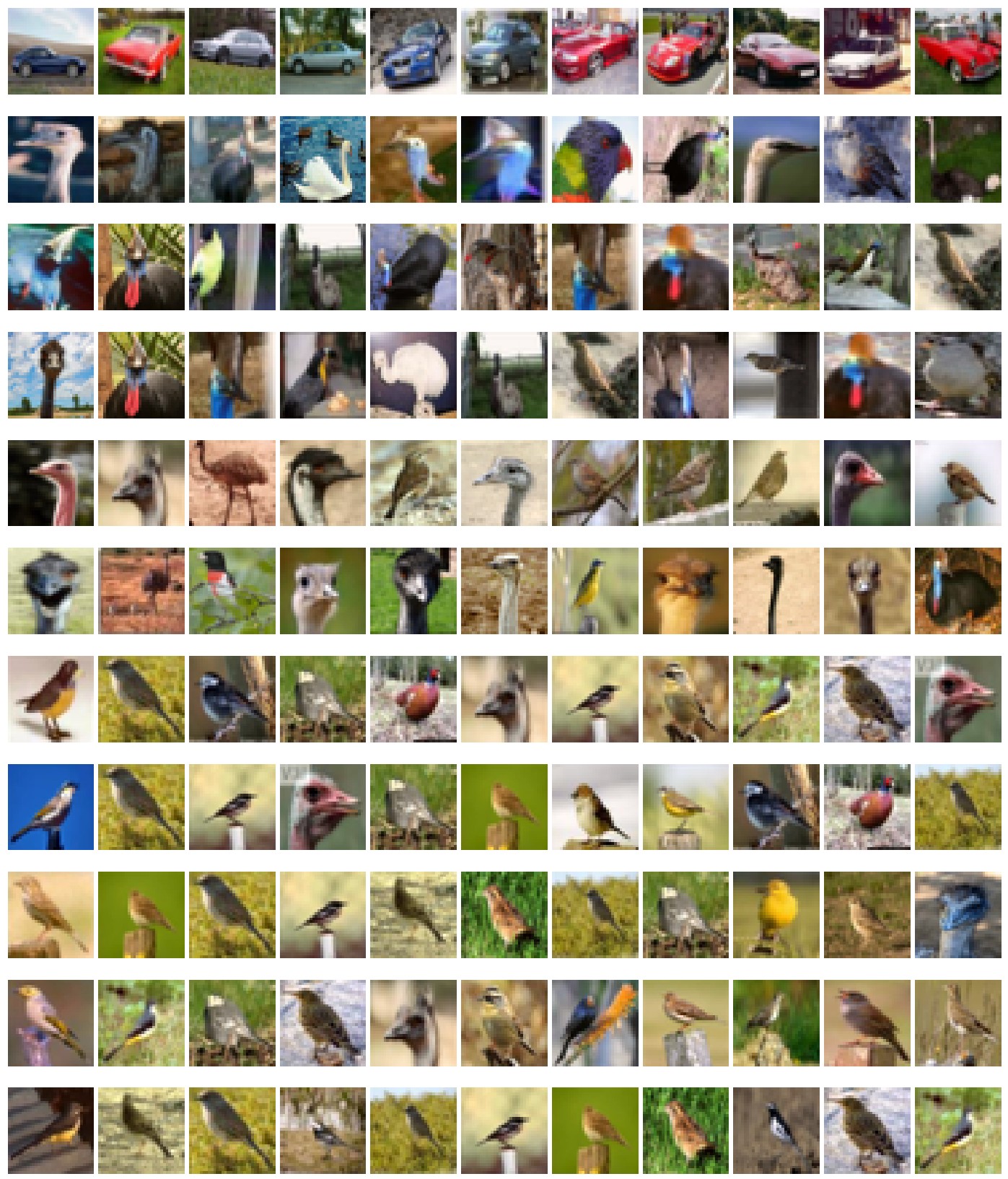

Fig. S40: Explanations for Cifar10 (set 5).

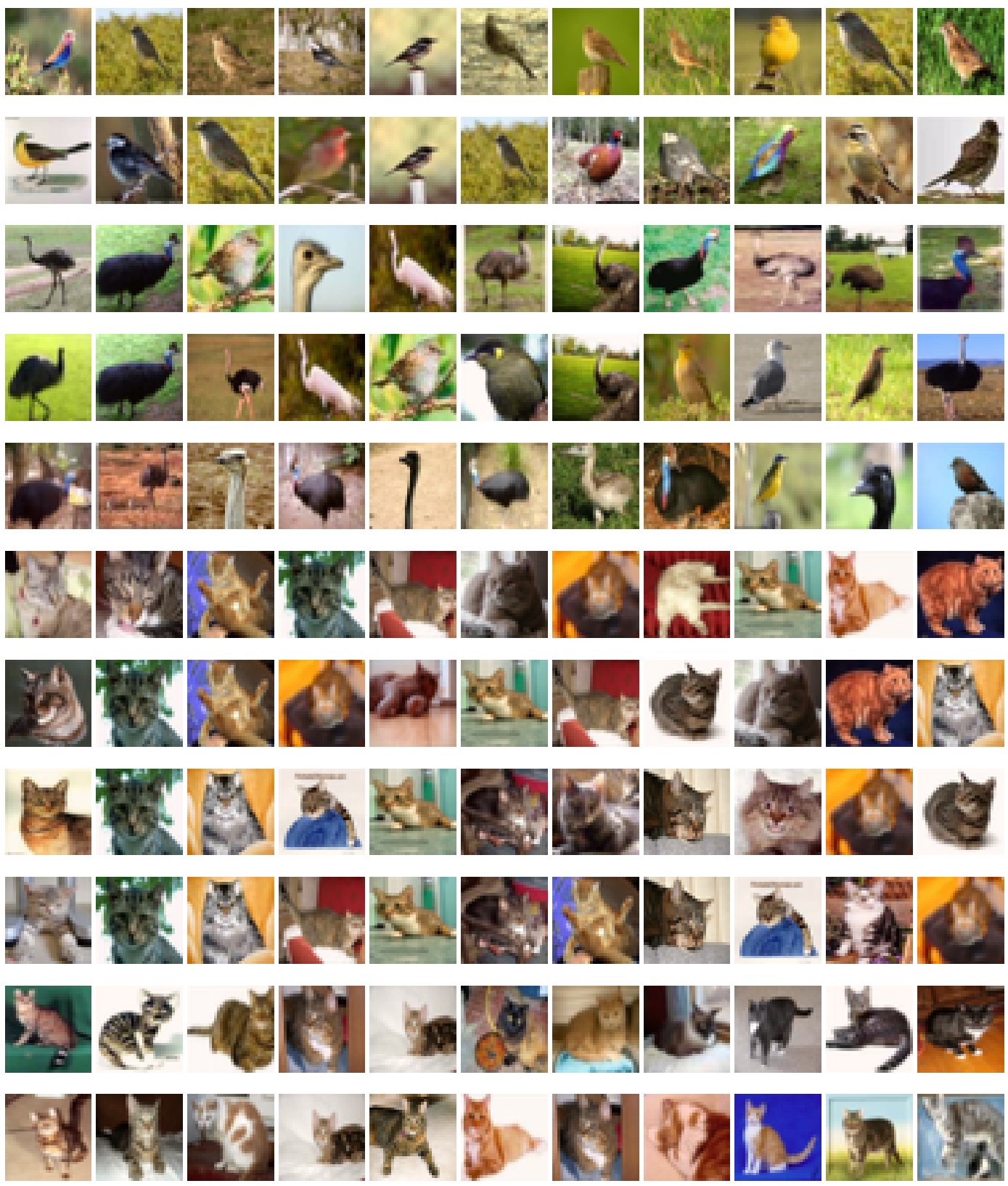

Fig. S41: Explanations for Cifar10 (set 6).

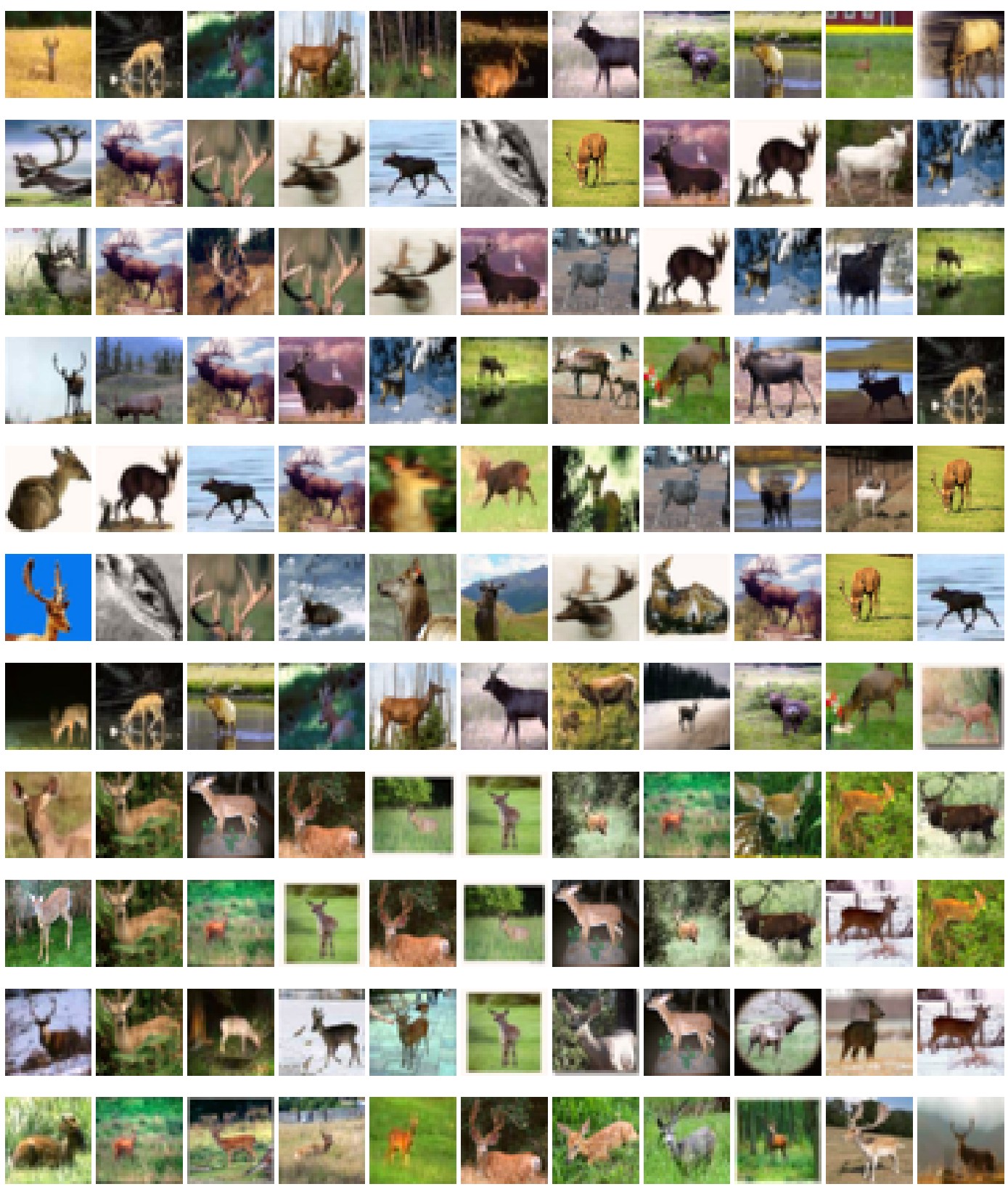

Fig. S42: Explanations for Cifar10 (set 7).

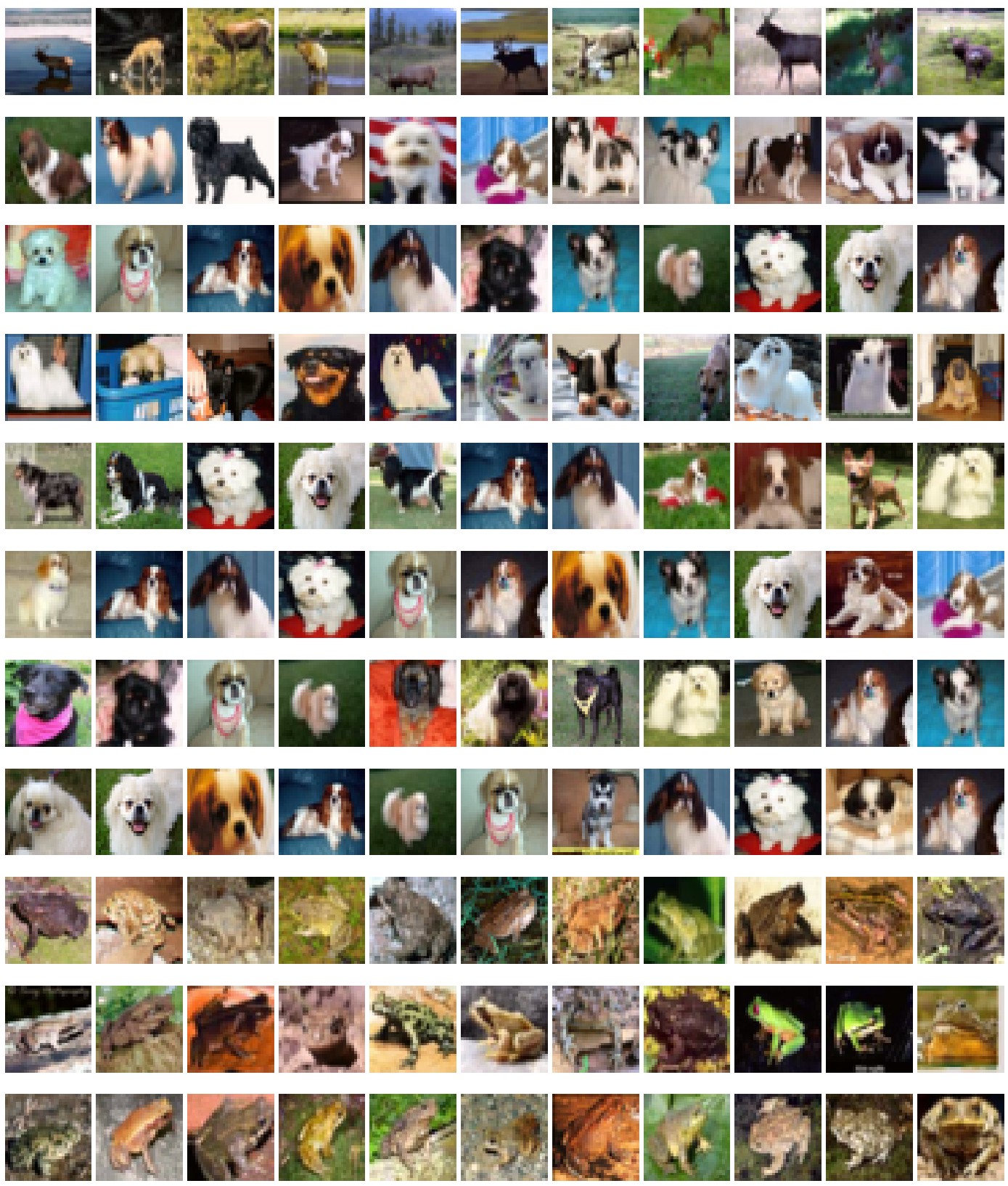

Fig. S43: Explanations for Cifar10 (set 8).

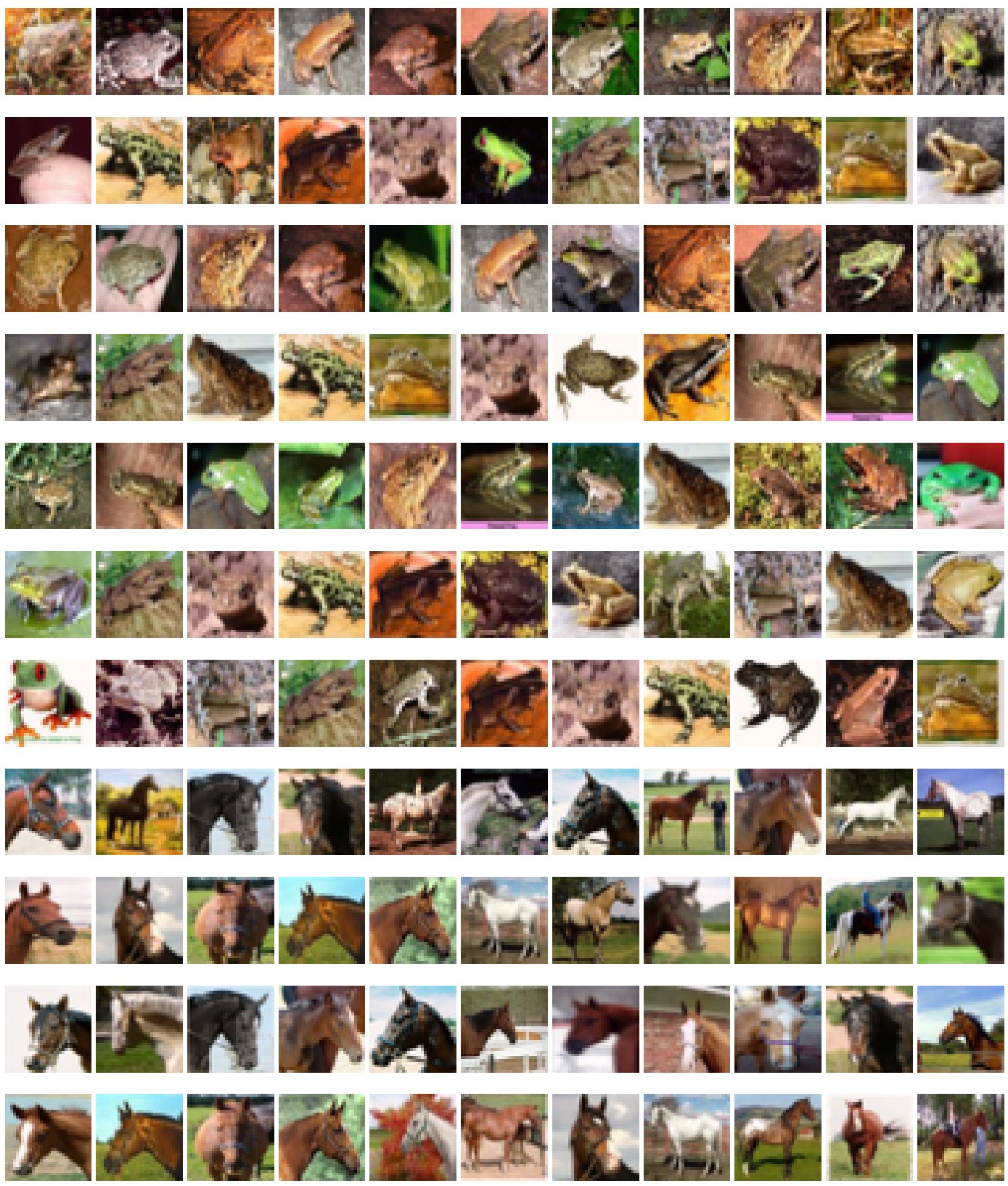

Fig. S44: Explanations for Cifar10 (set 9).

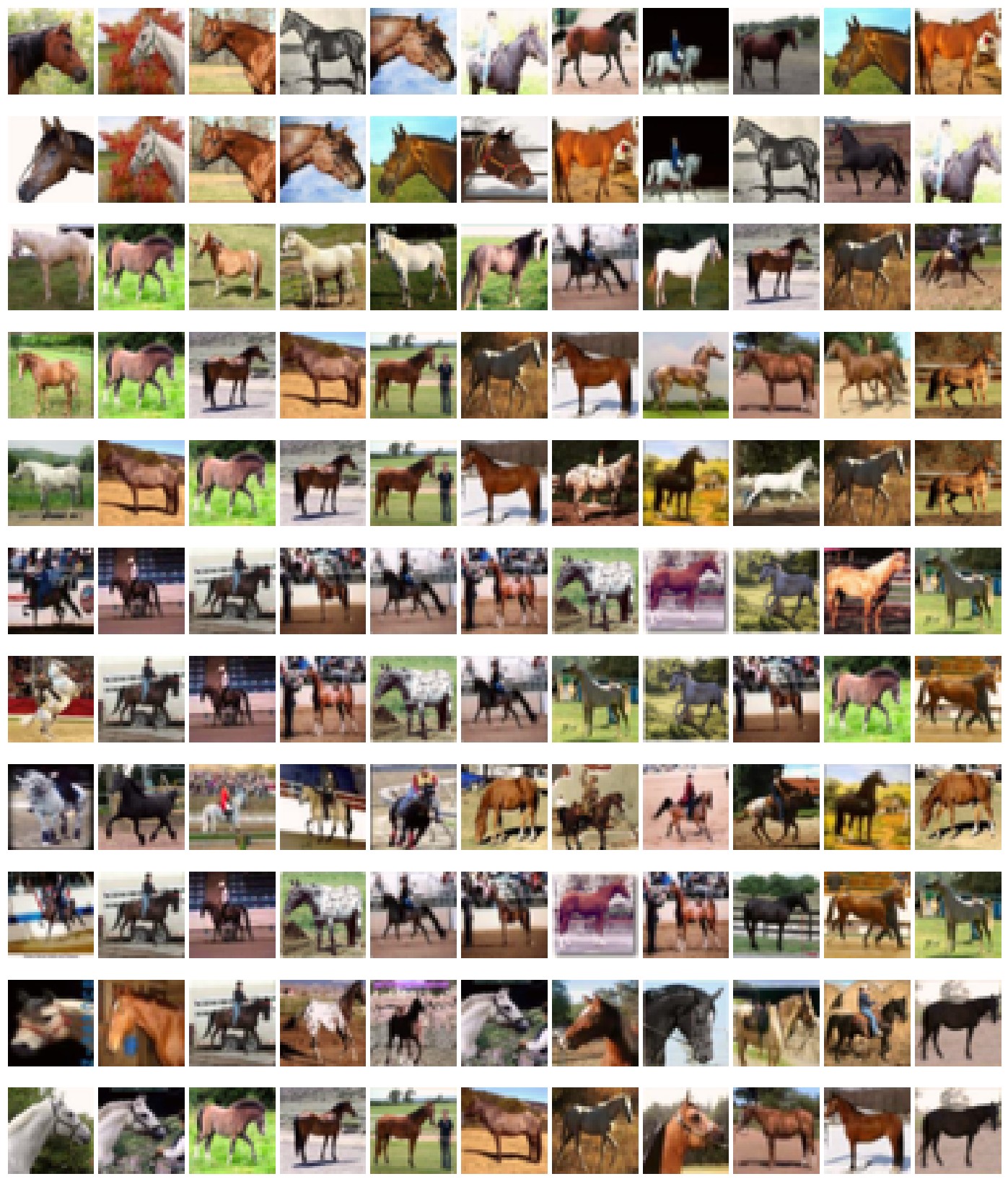

Fig. S45: Explanations for Cifar10 (set 10).

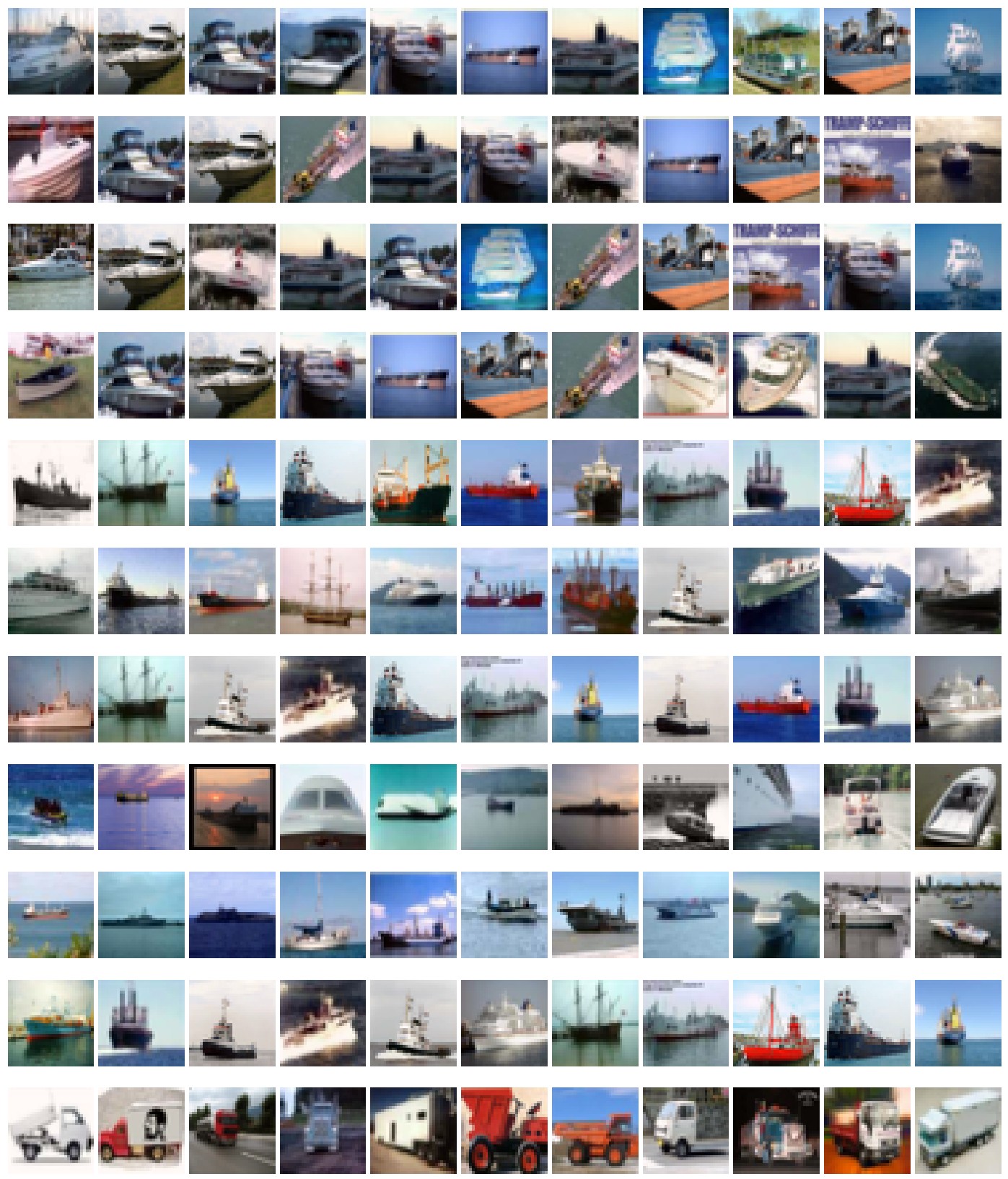

Fig. S46: Explanations for Cifar10 (set 11).

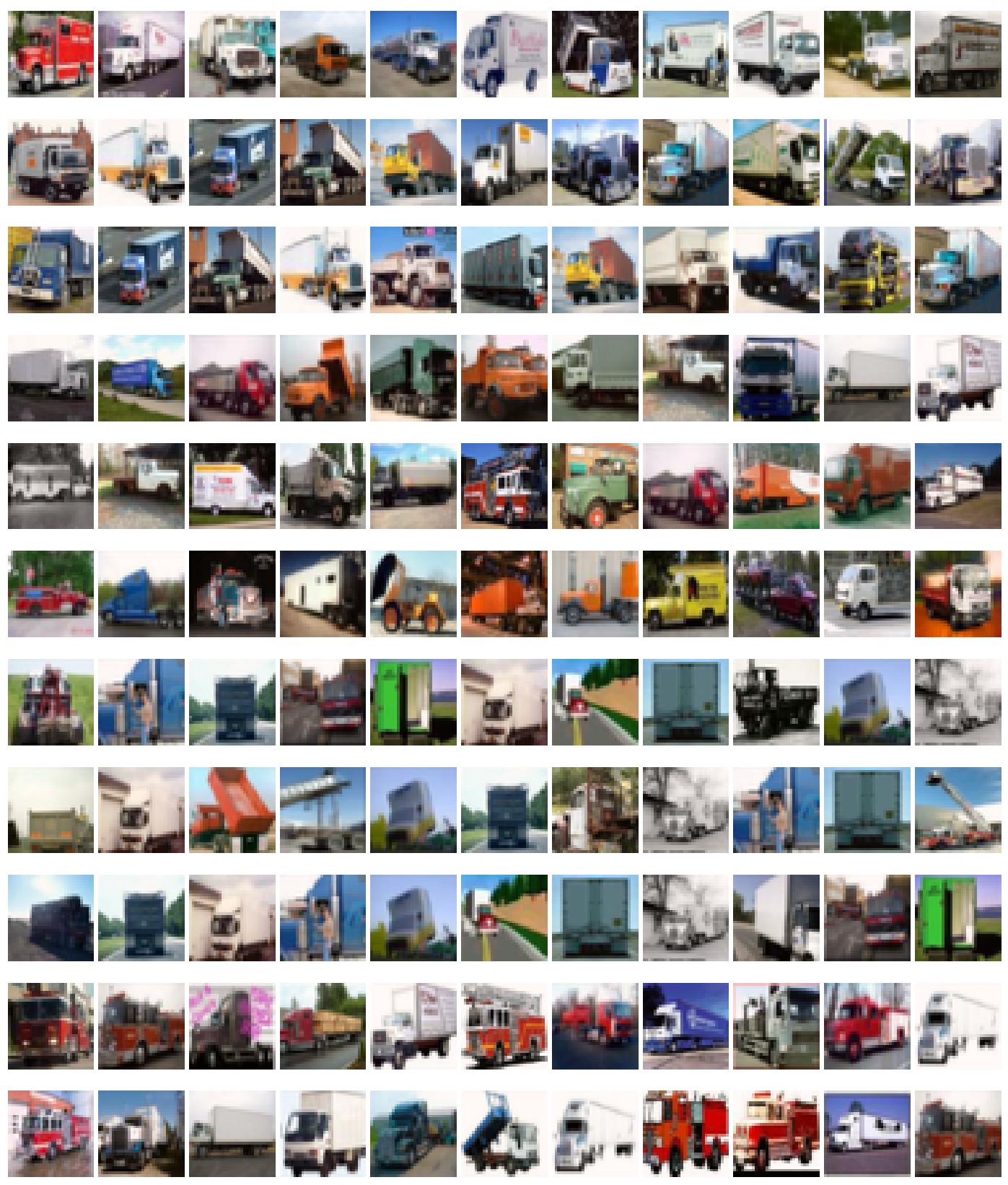

Fig. S47: Explanations for Cifar10 (set 12).

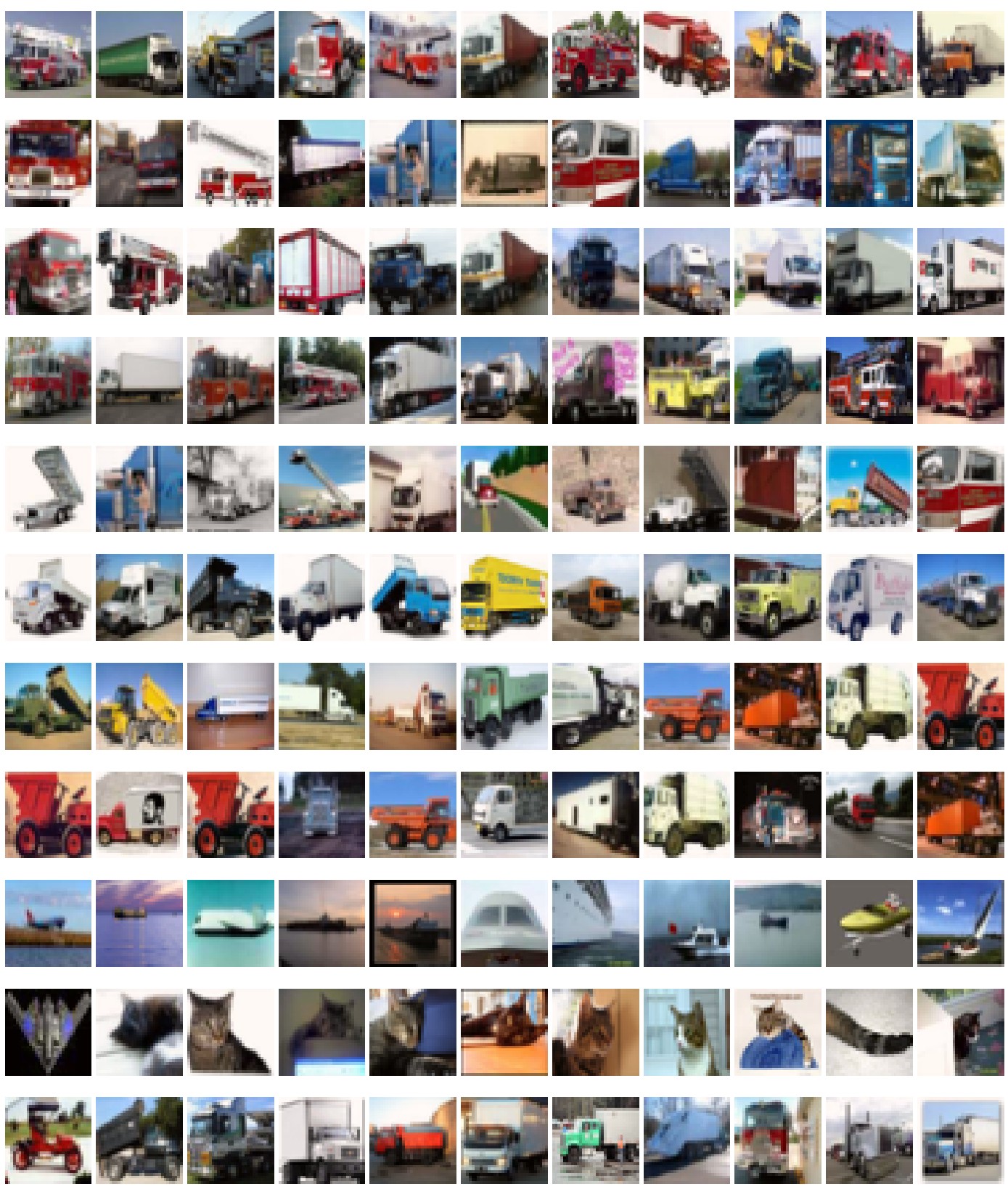

Fig. S48: Explanations for Cifar10 (set 13).

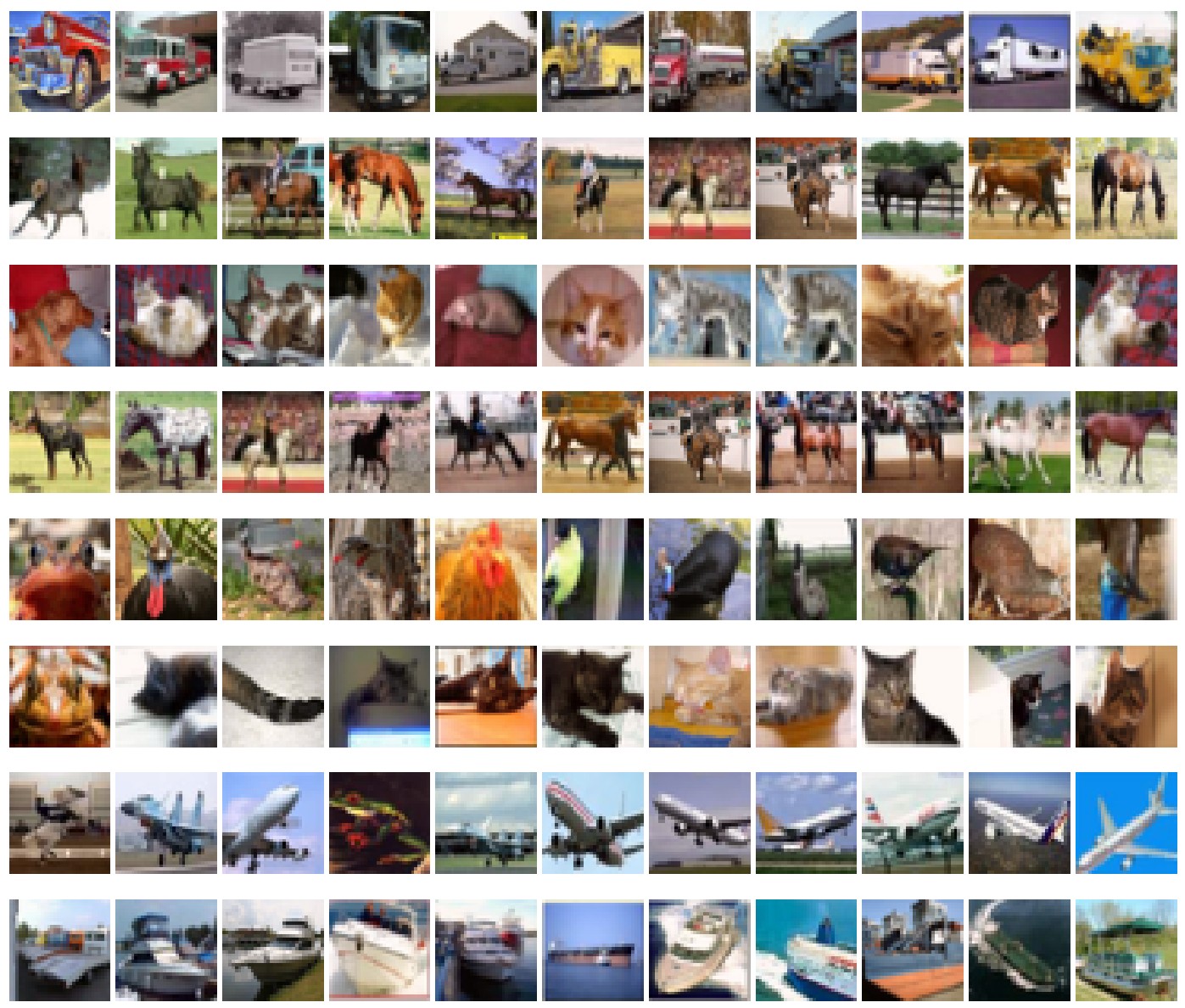

Fig. S49: Explanations for Cifar10 (set 14).

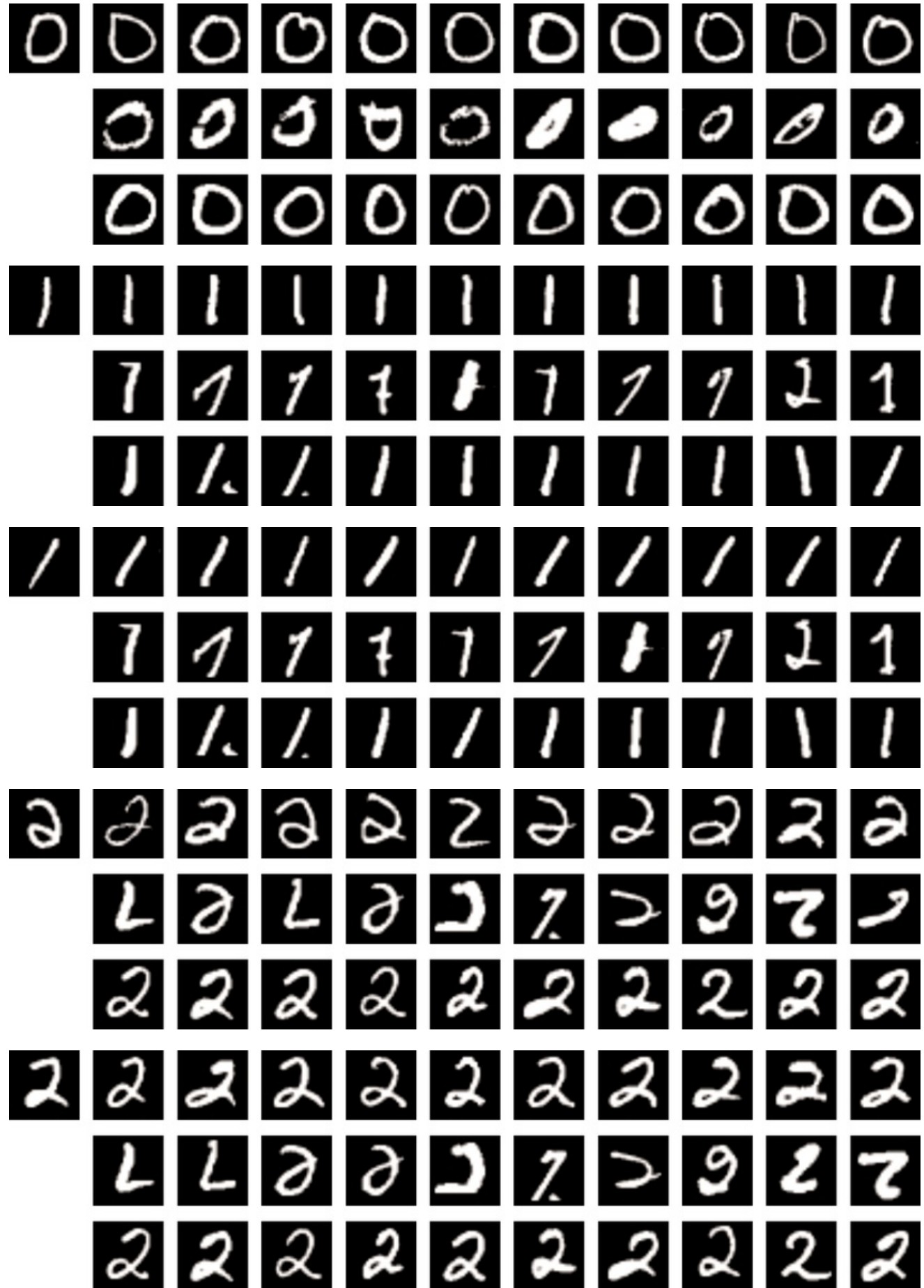

Fig. S50: Comparing the proposed GPEX with representer point selection [33] (set 1).

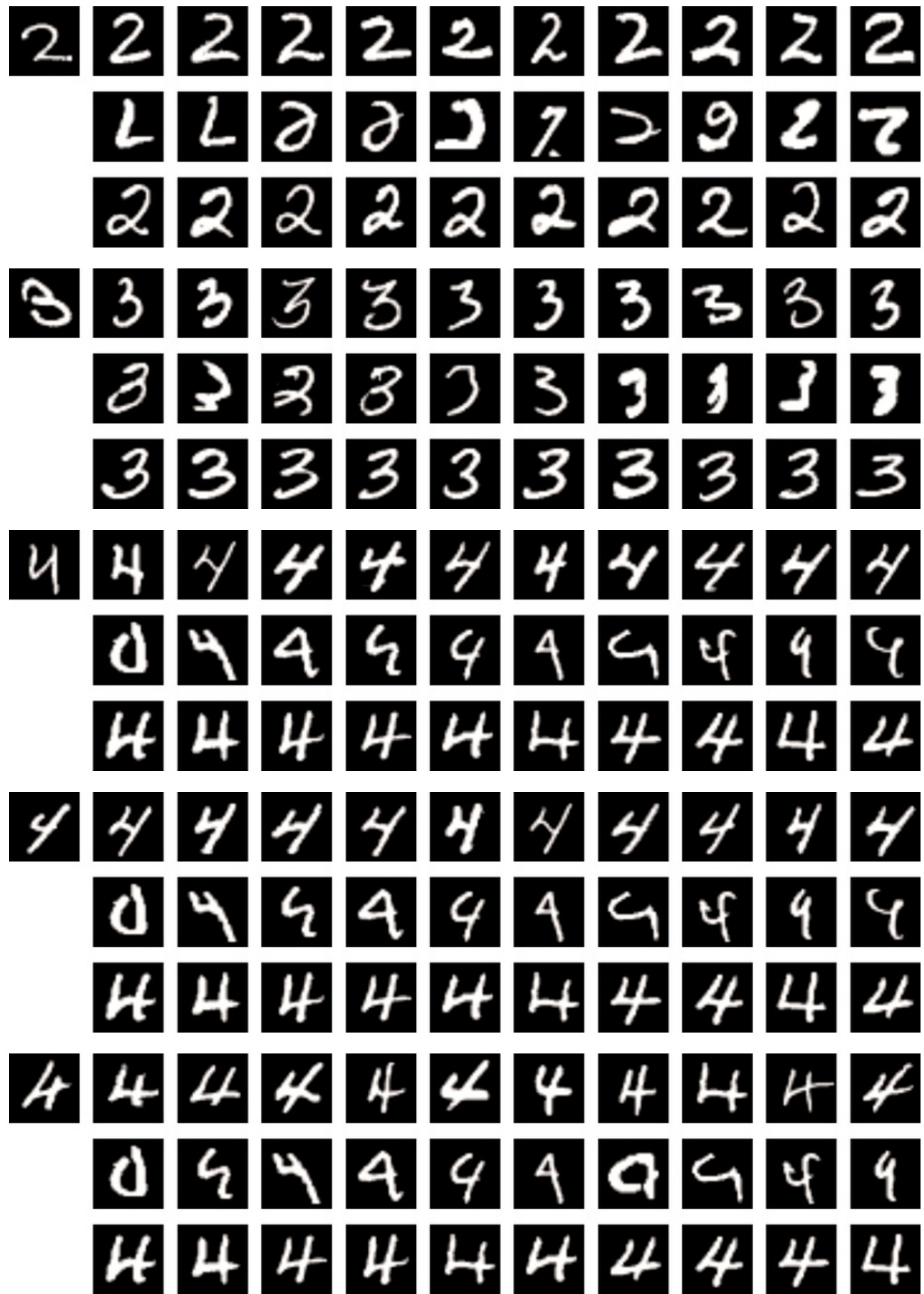

Fig. S51: Comparing the proposed GPEX with representer point selection [33] (set 2).

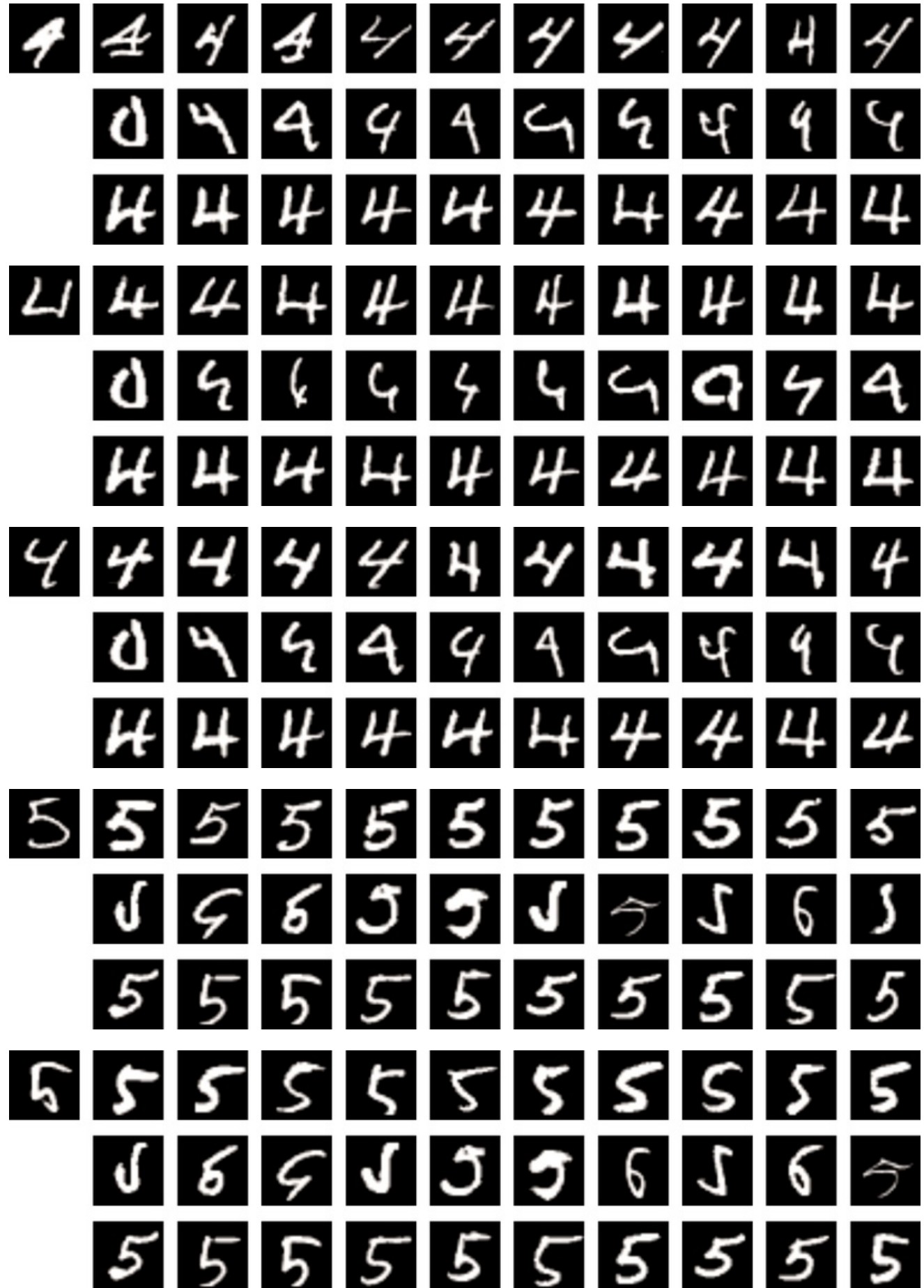

Fig. S52: Comparing the proposed GPEX with representer point selection [33] (set 3).

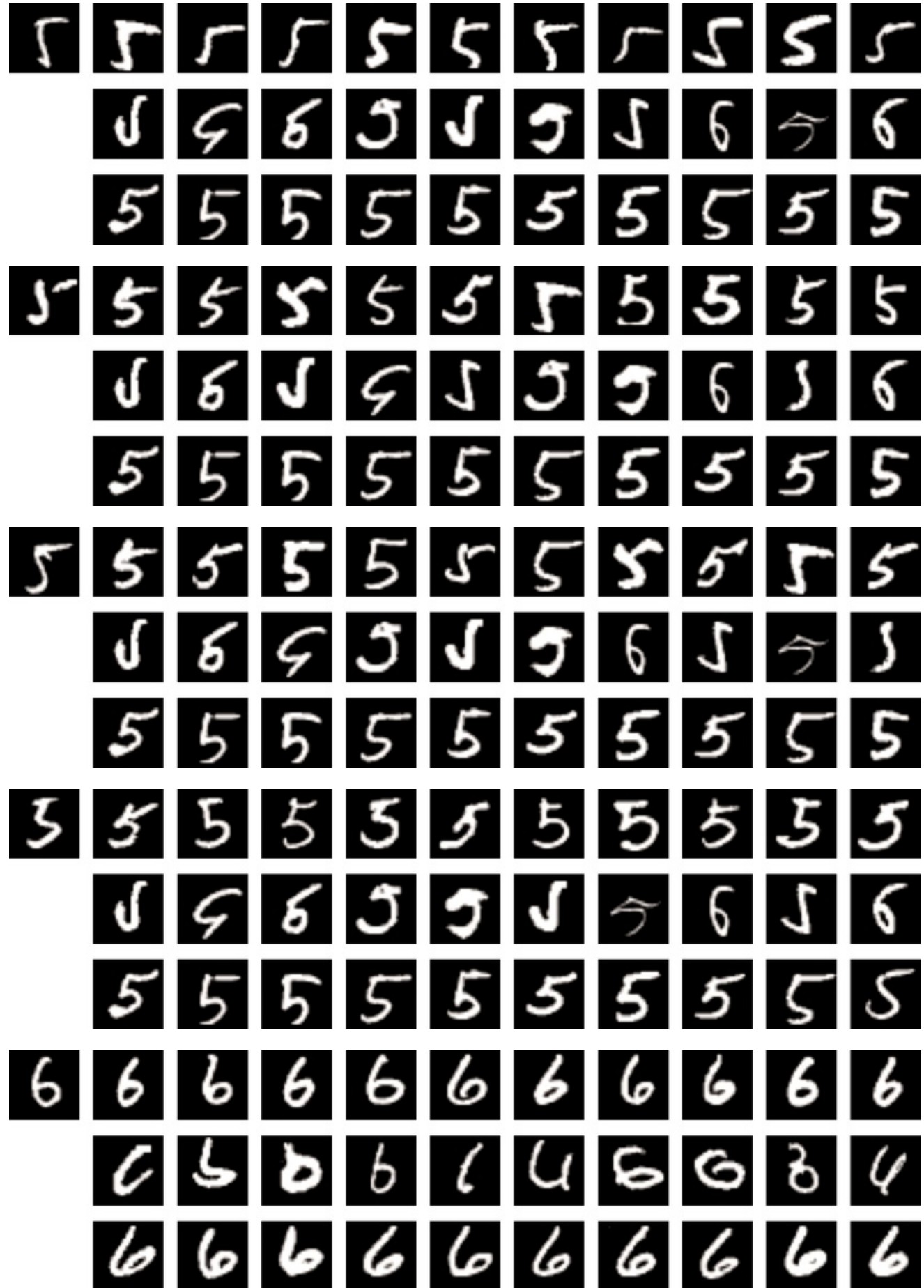

Fig. S53: Comparing the proposed GPEX with representer point selection [33] (set 4).

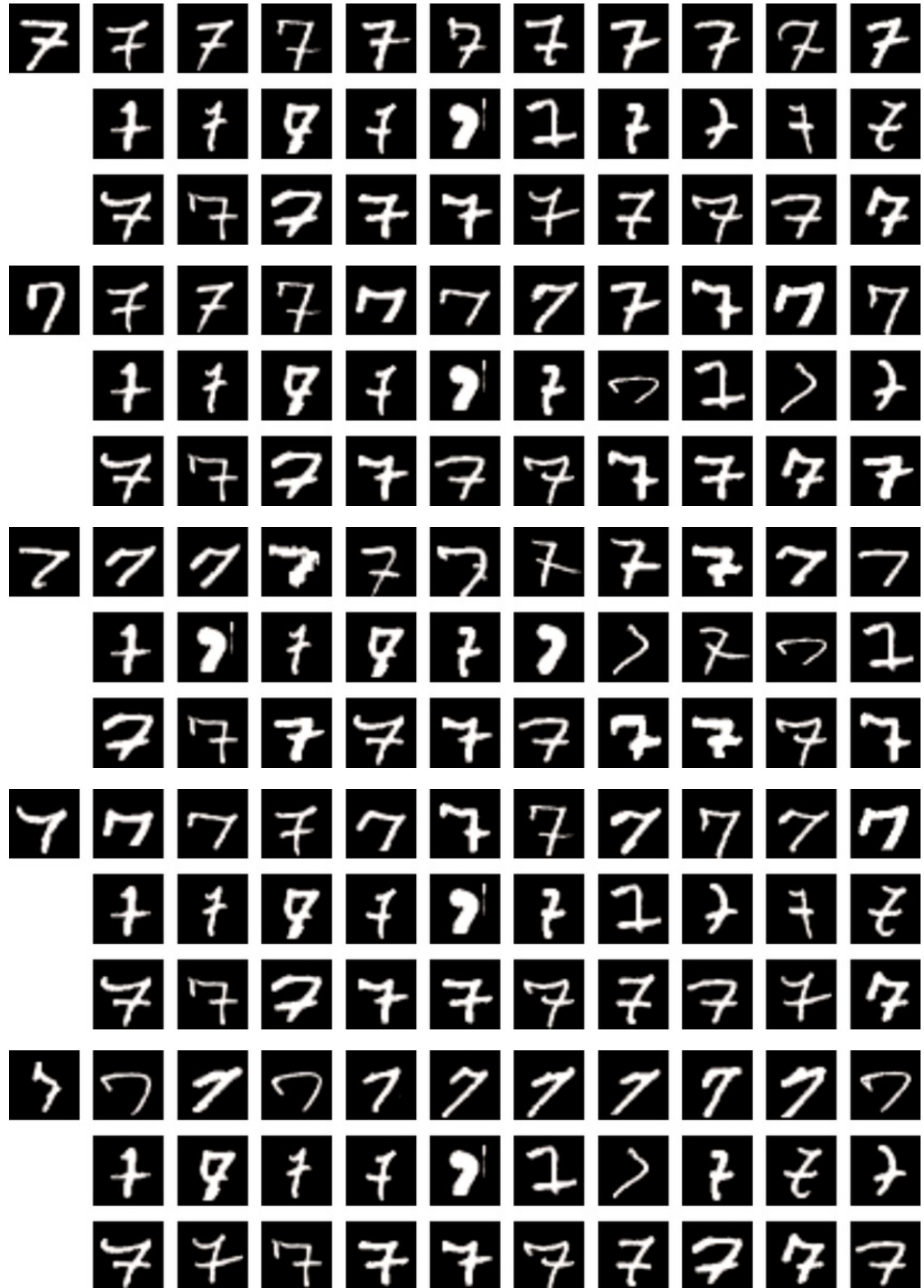

Fig. S54: Comparing the proposed GPEX with representer point selection [33] (set 5).

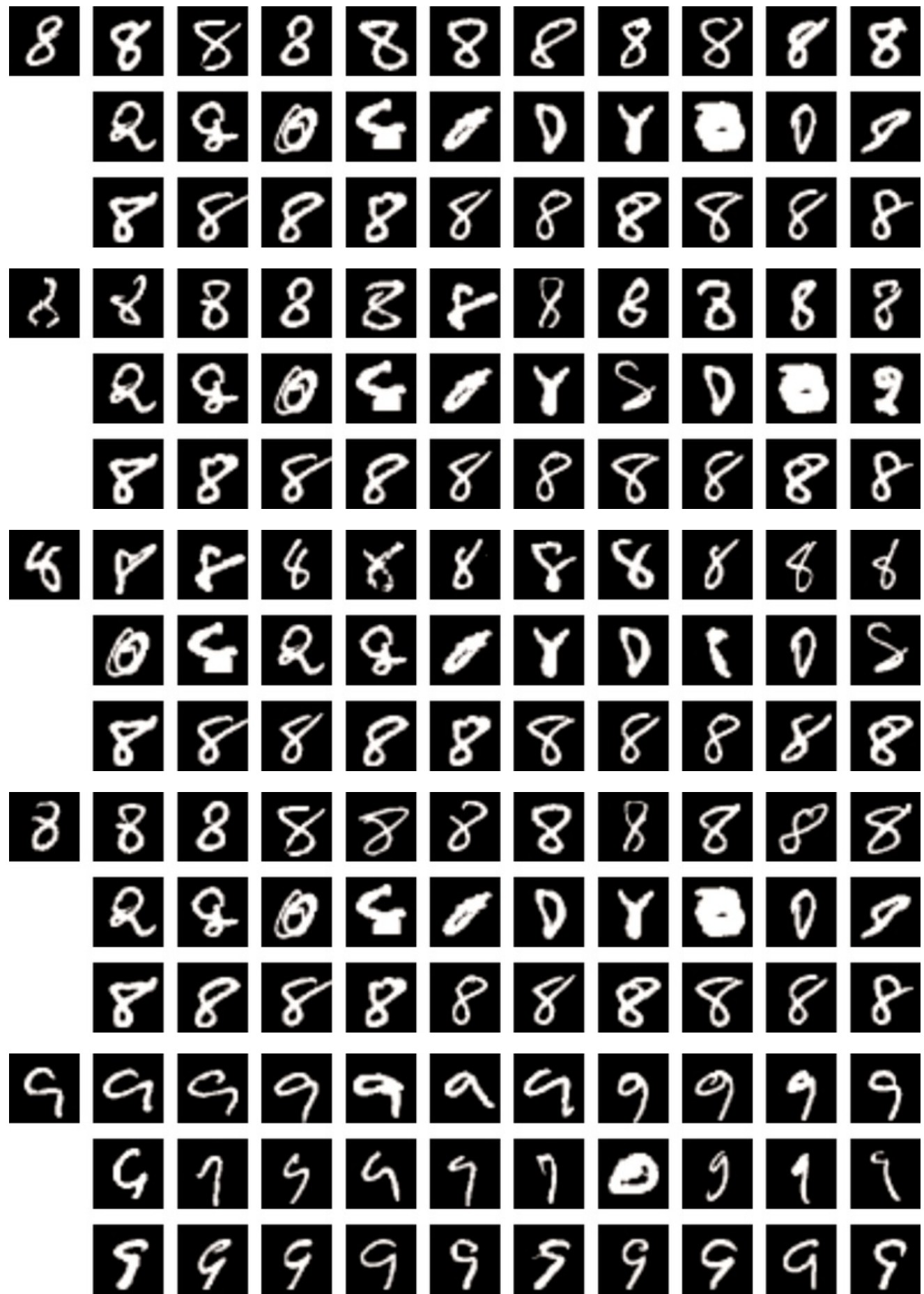

Fig. S55: Comparing the proposed GPEX with representer point selection [33] (set 6).

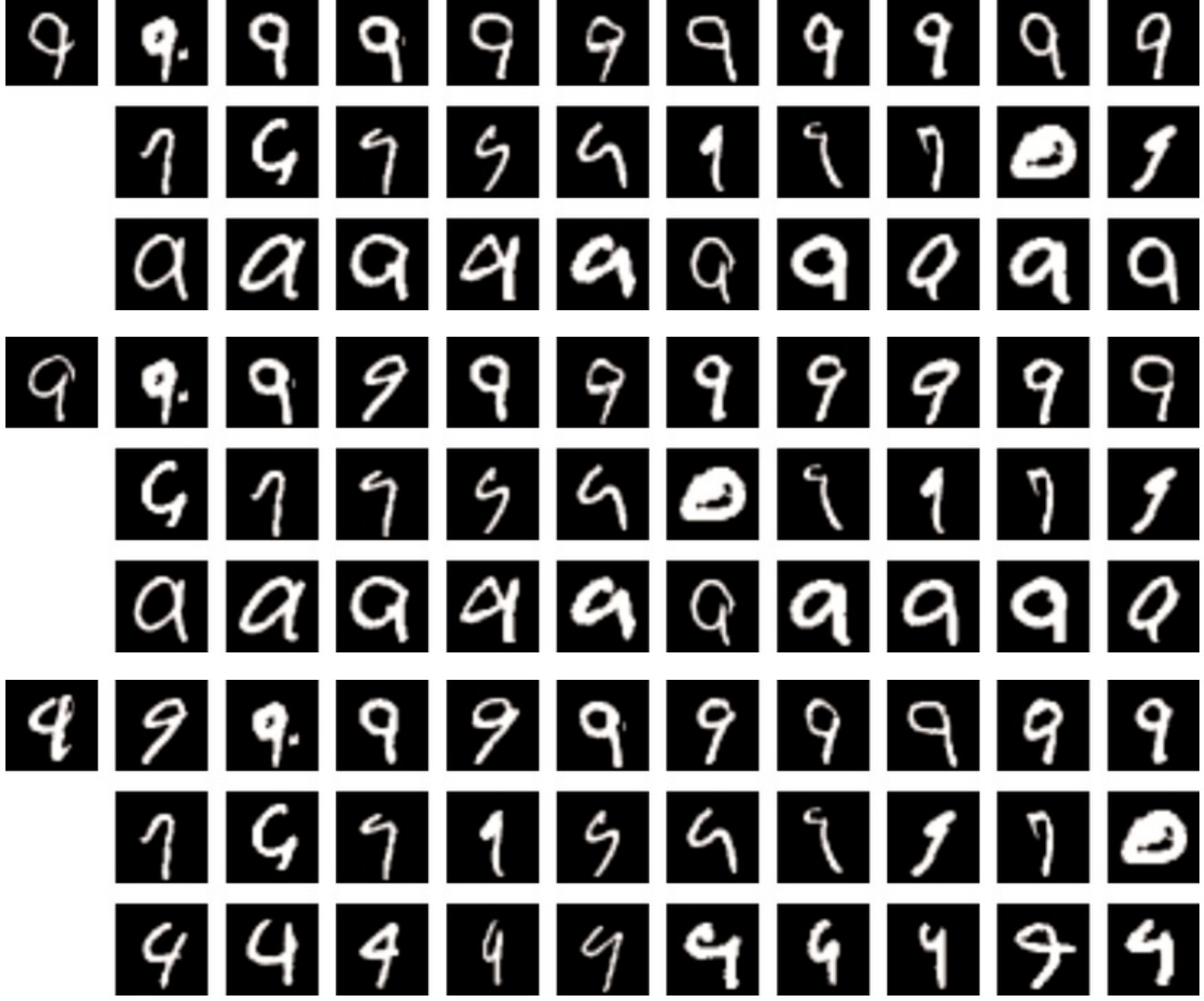

Fig. S56: Comparing the proposed GPEX with representer point selection [33] (set 7).

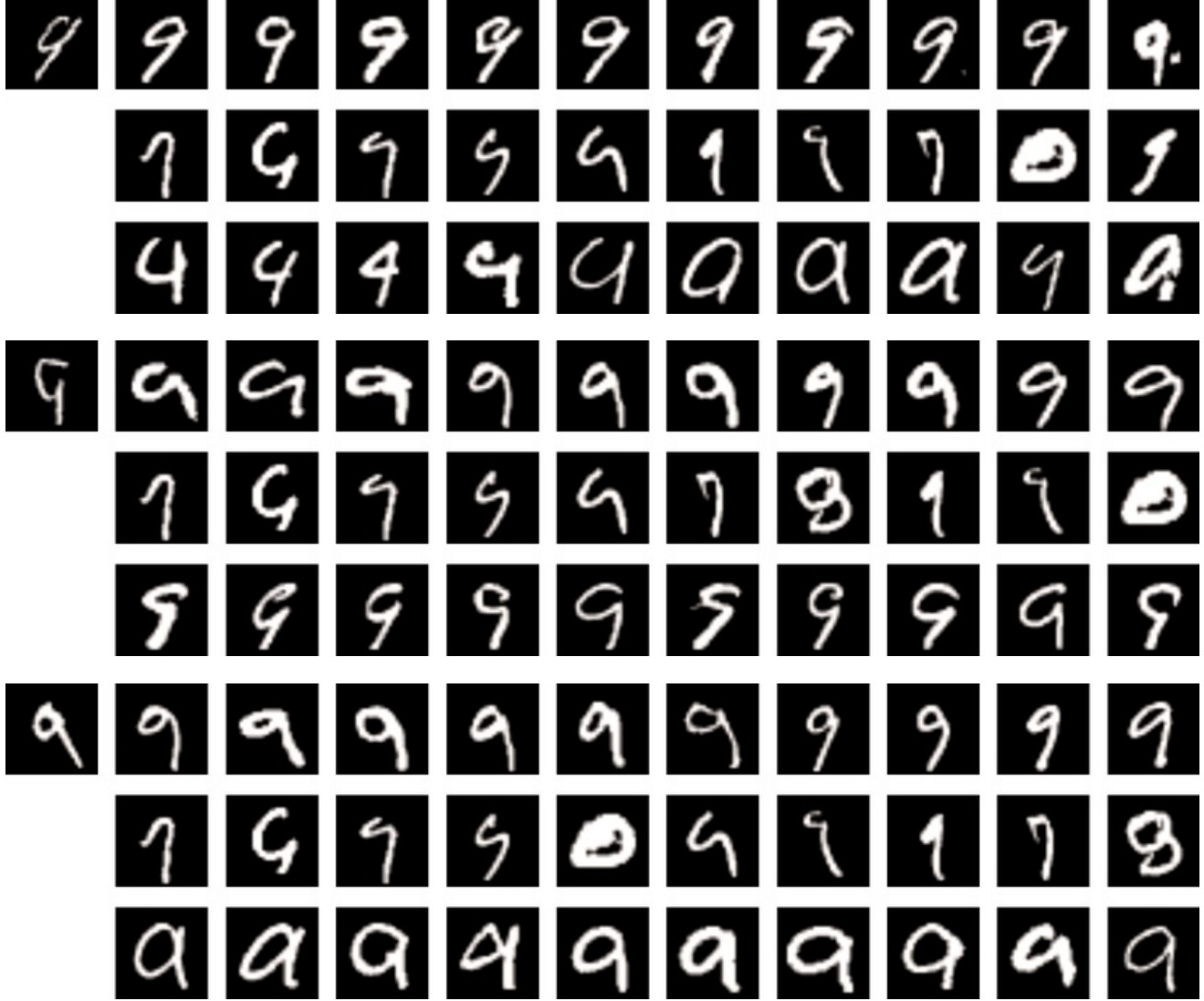

Fig. S57: Comparing the proposed GPEX with representer point selection [33] (set 8).

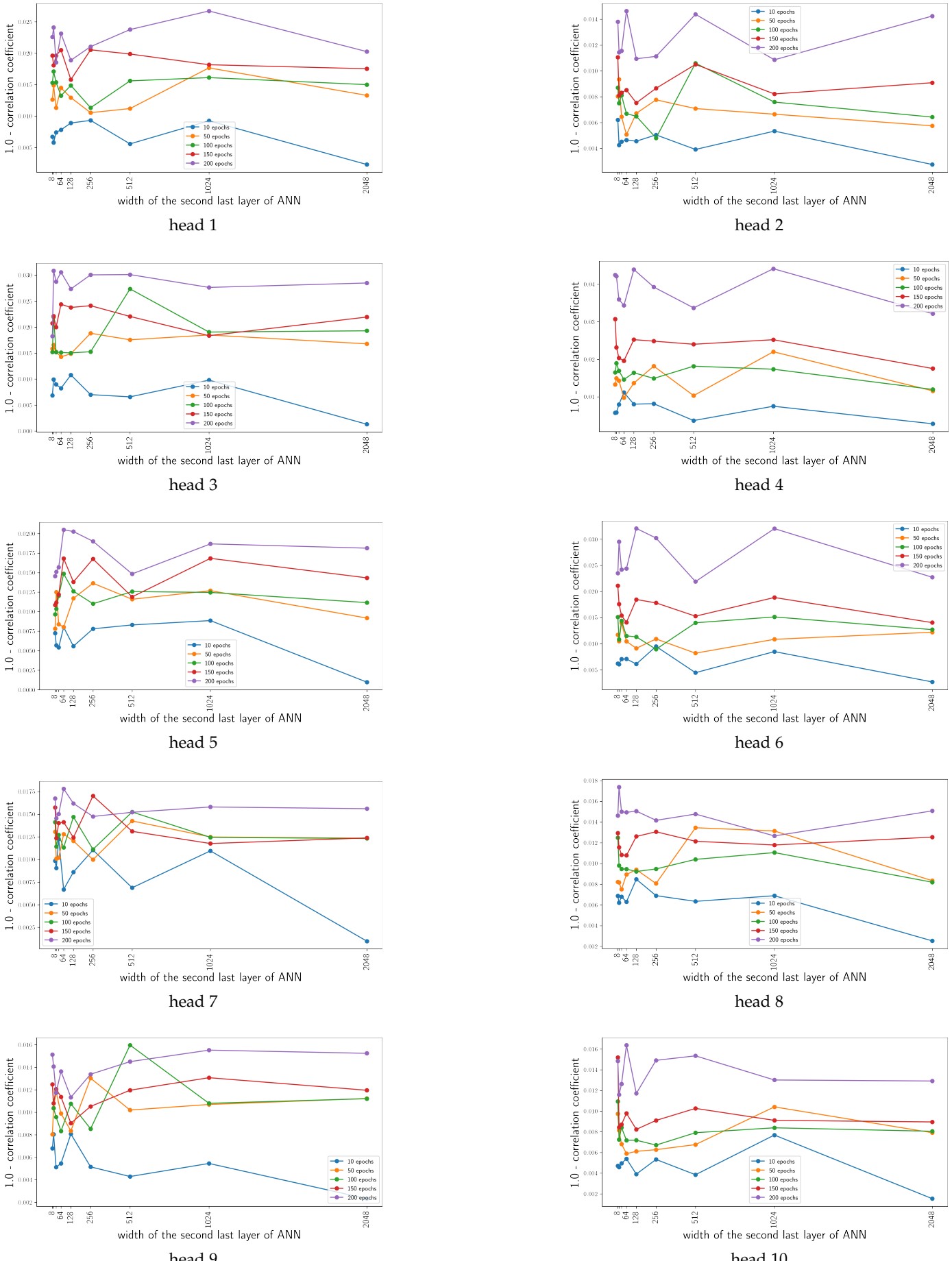

Fig. S58: Parameter analysis of Sec.S7.

|  | Cifar10 [15] | MNIST [6] | Kather [12] | DogsWolves [30] |
|---|---|---|---|---|
| ANN accuracy | 95.43 | 99.56 | 96.80 | 80.50 |
| GPs accuracy | 92.26 | 99.41 | 93.60 | 78.75 |

TABLE S1: Accuracies of ANN classifiers versus the accuracies of the explainer GPs on four datasets.

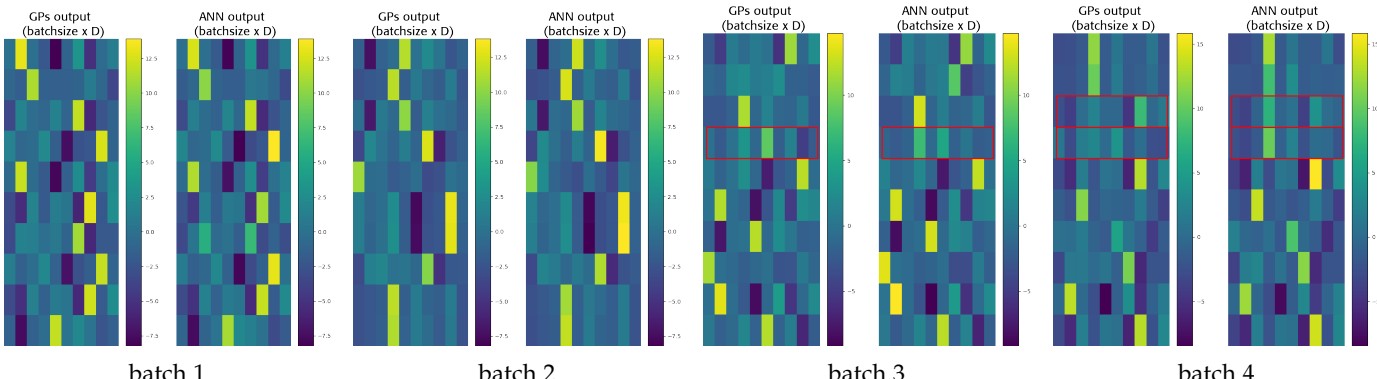

Fig. S59: Comparing GP and ANN outputs for four batches of Cifar10 dataset [33]. The red rectangles highlight the instnaces for which the predictions of GP and ANN (i.e. the class with maximum score) are different.

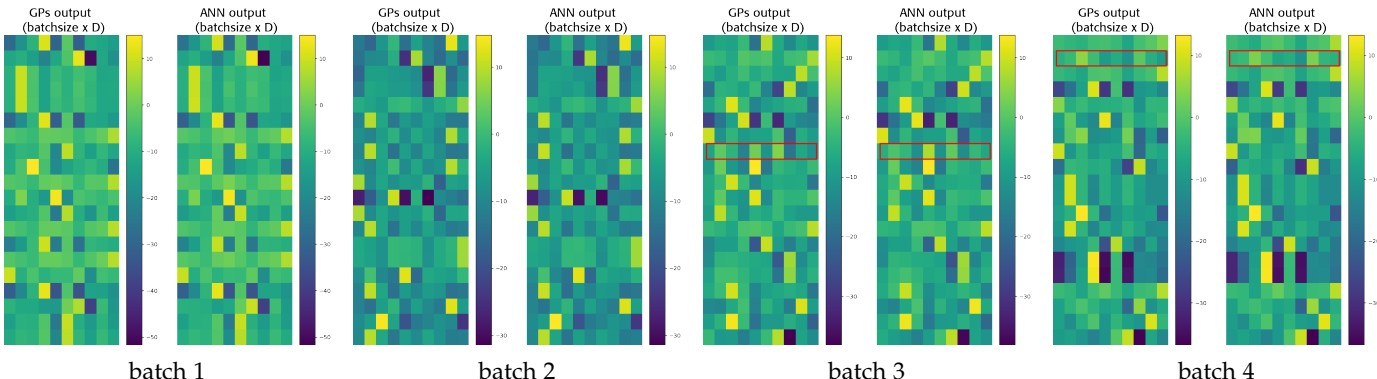

Fig. S60: Comparing GP and ANN outputs for four batches of MNIST dataset [32]. The red rectangles highlight the instnaces for which the predictions of GP and ANN (i.e. the class with maximum score) are different.

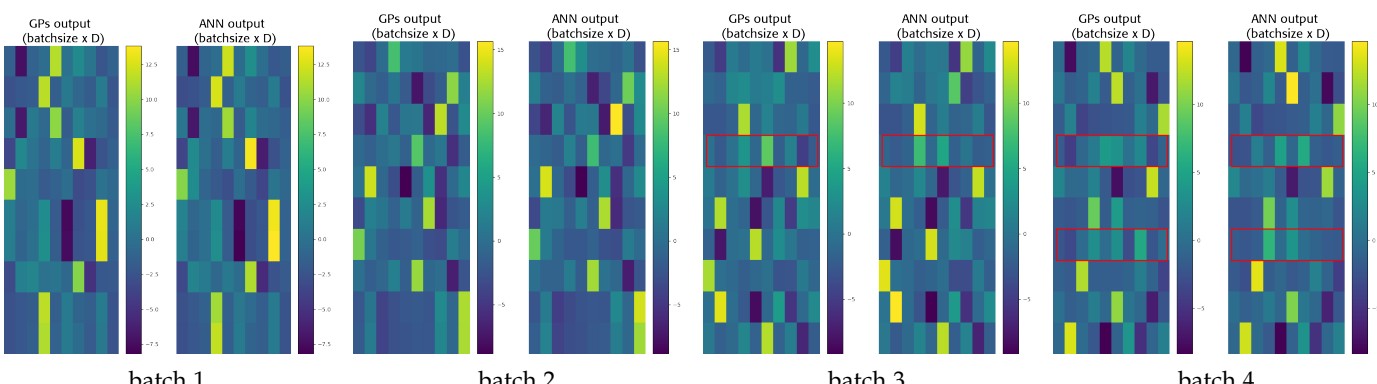

Fig. S61: Comparing GP and ANN outputs for four batches of Kather dataset [34]. The red rectangles highlight the instnaces for which the predictions of GP and ANN (i.e. the class with maximum score) are different.

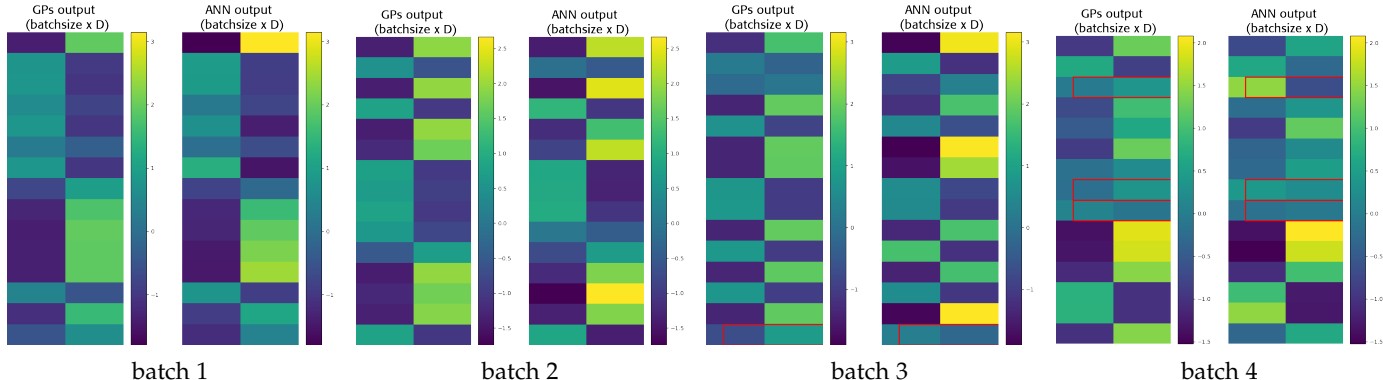

Fig. S62: Comparing GP and ANN outputs for four batches of DogsWolves dataset [35]. The red rectangles highlight the instnaces for which the predictions of GP and ANN (i.e. the class with maximum score) are different.

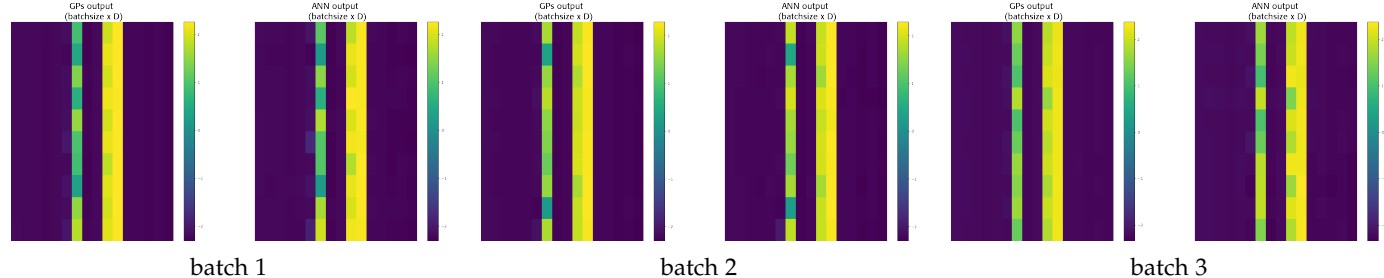

Fig. S63: Comparing GP and ANN (attention submodule) outputs for 3 batches of Cifar10 dataset [15].

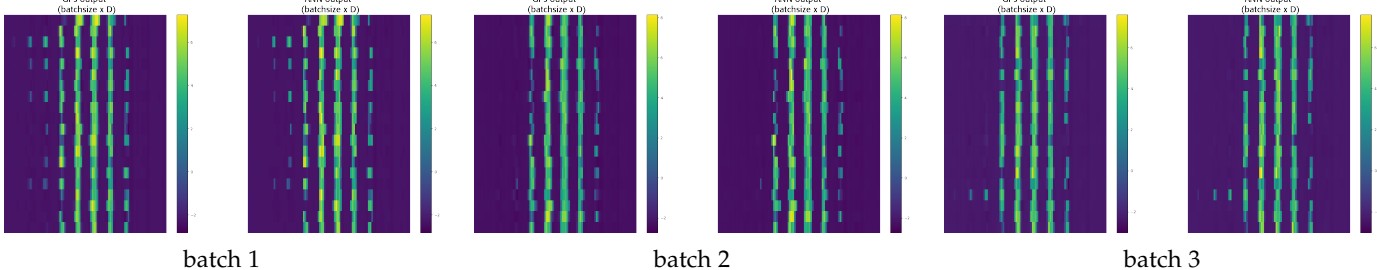

Fig. S64: Comparing GP and ANN (attention submodule) outputs for 3 batches of MNIST dataset [6].

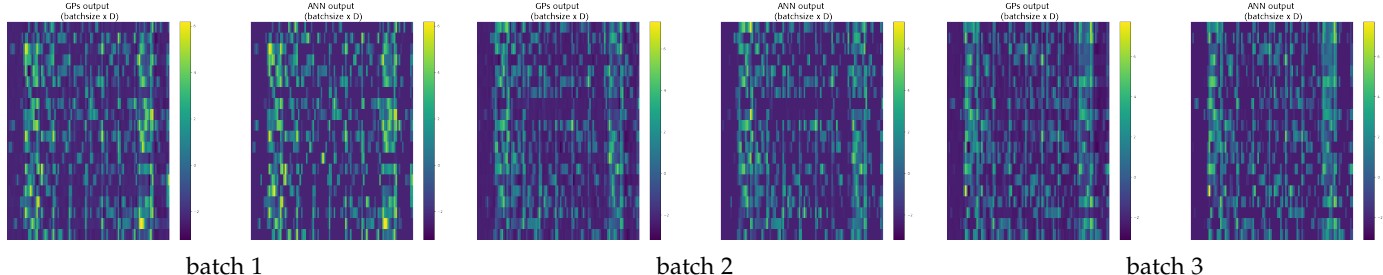

Fig. S65: Comparing GP and ANN (attention submodule) outputs for 3 batches of Kather dataset [12].

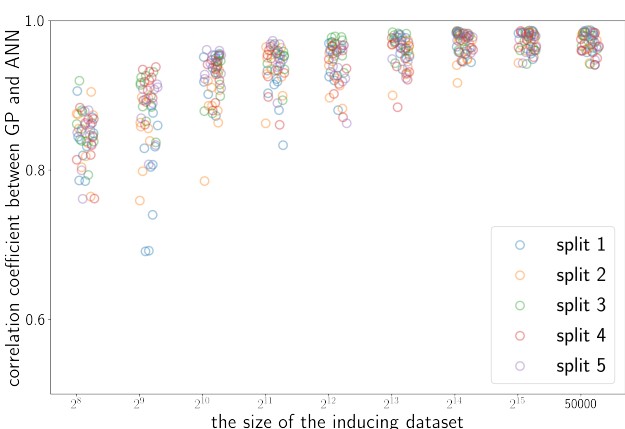

Fig. S66: Analyzing the effect of the size of inducing dataset.

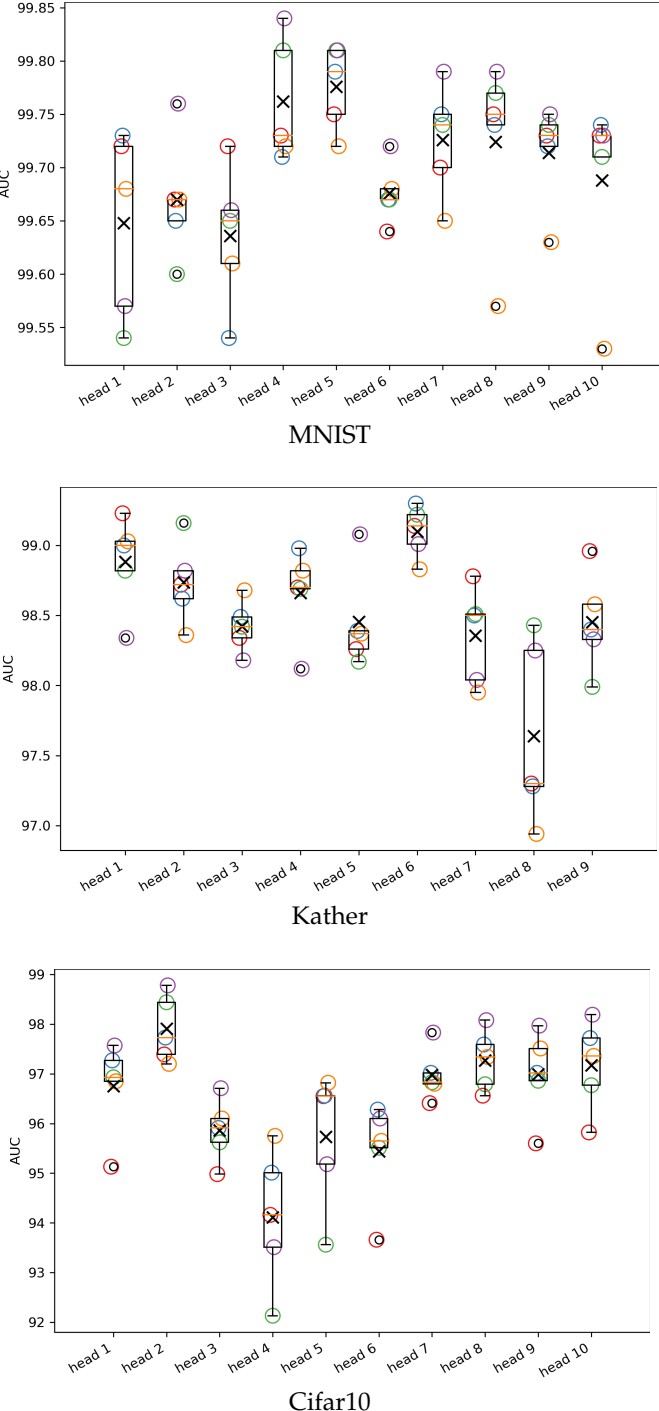

Fig. S67: Correlation coefficients for 5 splits.