# OpenReview forum: "GPEX, A Framework For Interpreting Artificial Neural Networks"
_NeurIPS.cc/2023/Conference — NeurIPS 2023 poster_

### Official Review · Reviewer_ycEM · 2023-06-26

**Soundness:** 3 good
**Presentation:** 3 good
**Contribution:** 3 good
**Rating:** 5
**Confidence:** 3

**Summary:**

This paper uses Gaussian Processes trained to match Neural Networks to subsequently "explain" the NN outputs by finding the nearest neighbors to any given test point in the training samples. An evidence lower-bound is derived that encourages the GP’s posterior to match the NN’s output. Scalability is obviously an issue for GPs. The authors use GPU acceleration techniques to enable training GPs with O(100k) inducing points. Various example experiments are shown where the 10 nearest-neighbors in training illuminate why the NN responds the way it does to a test example.


**Strengths:**

The method seems straightforward and intuitive. The GP matching is less aggressive where the uncertainty is large (far from any training points). The examples shown seem to demonstrate that the method appears to work on these examples.


**Weaknesses:**

I didn't find any mention about using the GP error bands in the explanations. Presumably the explanations are not as good in regions where the GP and NN are less closely matched, which should be the case where GP error bands are large since the method doesn't weight these regions as much in the loss. I.e., far from any training points, it's not obvious that the nearest neighbors chosen by the GP kernels are really driving the NN response. This isn't discussed at all as far as I can see.


**Questions:**

Have you looked at examples in regions where the GP error band is large?

Assuming it's true that explanations there are less reliable, can you return info using the size of the GP error at x_test that flags this for the user?

It's nice that you can interrogate the NN using your GP to explain specific examples, but it's not obvious (to me) how one would systematically study a NN in this way, since a human can only visually inspect a small number of examples as is done in the paper. Any comments / thoughts on this?


**Limitations:**

The Limitations paragraph is super concise (presumably due to space constraints), but it'd be good to expand it a bit and make it more clear. E.g., it talks about how the method runs well with O(1M) points, but that the GP fails to match the NN. As written this is hard to interpret.

---

> ### Author Rebuttal · Authors · 2023-08-10
>
> Please see the attached global pdf.

---

> > ### Author Response · Authors · 2023-08-21
> >
> > Thank you for your technical comments.
> >
> > === "it's not obvious (to me) how one would systematically study a NN in this way, since a human can only visually inspect a small number of examples as is done in the paper. Any comments / thoughts on this?"
> >
> > - It may so happen that in the GP's kernel-space, the test instance is equally close to many training instances (which seem to be the case for, e.g., rows 1-5 of Fig. 2a). Interestingly, for some test examples we see a different behavior as follows: $x_{test}$ is classified by finding one or a few training samples which are very similar (almost identical) to the test instance. As if the model "memorizes" the entire training set and when it sees, for example, the white dog in row 8 of Fig. S43 (in the supplementary) intuitively the model thinks it is very similar to one of the training instances that it has seen before (i.e. the very similar white dog in the 8th row and 2nd column of Fig. S43), therefore it classifies the image as dog. We see this behavior in many cases: rows 6 and 15 of Fig. 2(a), row 8 of Fig. S43, row 7 of Fig. S42, row 7 of Fig. S44, rows 7 and 8 of Fig. S48, and row 2 of Fig. S49. This "memorization behavior" and its connection to generalization has received recent research interest [43, 44, 45] and is clearly observable in our GP's explanations.
> > - Further quantitative experiments are needed to analyze the number of nearest neighbors which have a significant effect, and to support for example the above intuitive discussion. We have implemented the proposed GPEX as a tool (with all inference formula's happening under the hood) so other researchers can answer this and many other interesting questions. In the paper (in particular the supplementary material) we have analyzed many factors: the effect of the last layer's width, the number of inducing points, the number of epochs, etc, which has already made the article too long. Moreover, due to limited rebuttal time we could only perform some of the experiments suggested by reviewers. So we suggest this and many other interesting questions can be answered in future researches and by using our method/tool.
> >
> >
> > === "Have you looked at examples in regions where the GP error band is large? "
> >
> > We have analyzed many factors in the supplementary material (which is already long) and due to limited rebuttal time we could only perform some of the experiments. So the use of GP's uncertainty can be analyzed in future researches and by our tool. In particular, uncertainty quantification methods like Monte-Carlo dropout [7] have known issues, and GP's uncertainty may be a better choice. As a very preliminary experiment, we tried to relate GP's uncertainty to neural network's failures with the hope that for misclassified cases the GP shows a higher uncertainty. But at least in that preliminary experiment we failed to do so, that's why we didn't include that experiment in the paper.
> >
> >
> > === "The Limitations paragraph is super concise (presumably due to space constraints), but it'd be good to expand it a bit and make it more clear. E.g., it talks about how the method runs well with O(1M) points, but that the GP fails to match the NN. As written this is hard to interpret."
> >
> > Thank you for this comment. In the final version we will definitely consider this point. In particular, the page limit of the final version "may" be extended to 10 pages (maybe like previous year) which gives us even more space to accommodate the comments.
> >
> > [43] Feldman, Vitaly, and Chiyuan Zhang. "What neural networks memorize and why: Discovering the long tail via influence estimation." Advances in Neural Information Processing Systems 33 (2020): 2881-2891.
> >
> > [44] Adam Pearce, Asma Ghandeharioun, et al. "Do Machine Learning Models Memorize or Generalize?". website: https://pair.withgoogle.com/explorables/grokking/
> >
> >
> > [45] Mahajan, Divyat, Shruti Tople, and Amit Sharma. "The Connection between Out-of-Distribution Generalization and Privacy of ML Models." arXiv preprint arXiv:2110.03369 (2021).
> >
> > [7] Y. Gal and Z. Ghahramani. Dropout as a bayesian approximation: Representing model uncertainty in deep 366 learning. In international conference on machine learning, pages 1050–1059. PMLR, 2016

---

### Official Review · Reviewer_efq3 · 2023-06-29

**Soundness:** 2 fair
**Presentation:** 2 fair
**Contribution:** 2 fair
**Rating:** 5
**Confidence:** 4

**Summary:**

This article proposes a new training approach for Gaussian Processes (GPs) which encourages their posterior to match a given Artificial Neural Network (ANN) output. Their approach adopt a scheme that permits a scalable training, which is an issue when it usually comes to GPs, and that is motivated by the derivation of an evidence lower bound (ELBO) encouraging the faithfulness of the GP to the target ANN. The authors then  explain the decisions made by an ANN with the help of a matching GP with two different methods : explanations by similar examples (similar in the kernel space of the GP), and using saliency map to determine why some examples where similar.

**Strengths:**

1. It is, to the knowledge of the authors, truly original to use a GP in order to explain a matching ANN.

2. The approach for training GPs tackles a main issue of traditional approaches : scalability with regards to the number of inducing points.

**Weaknesses:**

_Major_

1. My first concern, which could probably summarise many flags I have to raise, for it leads to several problems : the article is too dense. One consequence is that there are too many things putted in Supplementary material that are necessary to understand the article in its whole. For example, the way the CAM-like explanations (line 282) are computed is not even slightly described in the main article; or, « According to the experiments and detailed discussions of Sec. S6 in the supplementary, the GP’s kernel that we find in this paper is superior due to a technical point in the formulation of representer point selection » (lines 299-300); or the fact that the derived  ELBO is a contribution in itself, and isn’t even presented in the main paper, only pieces of it. The Supplementary material should not contain information that are vital to understanding the methods, the idea and the approaches, but further readings.

2. When manipulating concepts such as explainability and interpretability (which seem, in the article, to be used interchangeably; lines 20-21 :  « After matching the GPs to the ANNs, we used the GPs’ kernel functions to explain the ANNs’ decisions. »; the title says « interpreting artificial neural networks », and on lines 75-76 : « GPEX can be used by machine learning researchers to interpret [...] their artificial neural networks. »), since they do not have a clear and universal definition, it is important to state what these terms refer to in the article, just like, in Section 2.1, some notations are defined. This lack of clearness makes it harder to grasp the goal of the article.

3. If I understand correctly, as for the explainability approach, the way the nearest neighbours are found is by looking at the nearest neighbours, but in the kernel space. How does that differ from only computing the nearest neighbours in the output space of the network? How does that makes the neighbours that are found more relevant? Both the kernel and the network can be seen as an embedding. Plus, the GP's job is to imitate the network. It seems to me like this approach combines two explainability approaches : finding a « more interpretable » predictor imitating the predicting behaviour of the former, and finding examples that are similar to one another. Why is combining those two approaches, both having their flaws, would lead to any better results when it comes to explainations?

4.1. I find the title misleading; explaining (interpreting, as the title says) ANNs would seem to be the main point of the article, but in reality, explaining ANNs concerns only page 8 and half of the last page, whereas training a GPs faithfully to a trained ANN really is the main point of the article. Plus, both the saliency map and looking at the most similar examples (which corresponds to approximate the decision function of the network by a more interpretable predictor : a k-nearest neighbours predictor) from the training set are not novel ideas.

4.2. As for the related works, explaining using GPs is only discussed on ~3 lines, citing a single work.This might be the first time an explanation is based on a bridge between GPs and ANNs, but the authors stated that « Gaussian processes are highly interpretable » (lines 37-38) and that « The analogy between Gaussian processes (GPs) and deep artificial neural networks (ANNs) has received a lot of interest, and has shown promise to unbox the blackbox of deep ANNs » (lines 1-3); it is therefore expected to see « what kind of interest they have received », especially in the discussion / related work sections of the article.

5. One of the contribution is that the proposed approach is more scalable when compared to previous approaches. Without any empirical comparison with state-of-the-art for training GPs, both in terms in training time and performances, it is hard to state whether the proposed approach works particularly well or if it is just have effective as what is found in the literature. That is also why the experiments ran in Section 4.1 could be misleading. Since no benchmark is used to compare the proposed approach to, how can one know if the obtained results are good or not? This lack of information makes it hard to state whether or not the results are positive.

6. I found some claims to be vague, doubtful or at least are highly debatable, but most importantly sometimes not supported at all; lines 37-38 : « Gaussian processes are highly interpretable. » How are they more interpretable? In what way? Doesn’t that depend on the kernel that is used? The same goes for lines 207-208 : « Because according to theoretical results on GP-ANN analogy, the second last layer of ANN should be wide. » This might be true, but there is a need to cite some works here. Also : « Our method scales very well, and Alg. 2 runs fine even on imagenet with more than 1M inducing points » (lines 343-344); what’s « fine » mean? (See also Weaknesses – Major – 5)

7. It is said that experiments were conducted on « 5 datasets (4 image datasets, and 1 biological dataset) » (lines 17-18), but nothing in the article is discussed about the biological dataset.

_Typos / Minor_

1. Line 85 : « The number of GPs is equal to the number of $\underline{the}$ outputs from the ANN. »

2. Some nomenclature and terminology is unclear. For example, line 91-92 : « we consider a general feed-forward pipeline that contains an ANN as a submodule »; what’s a feed-forward pipeline? What is a submodule? Figure 1a) somewhat doesn’t help understanding. (See also Weaknesses – Major - 2)

3. Equation 5 : Is there a missing term?: what is the expectation computed with regards to? (?~q)

4. Line 164 : « procecdure »

5. Line 167 : « matrcies »

6. Line 179 : « from linear algebra if follows that »

7. Line 327 : requries

8. Having the Related Work at the end of the article is peculiar; it seems to me like precious information for better understanding the relationship between ANNs and GPs during the reading of the article (and better understanding the relevance of the work) are put at the end of the article.

9. It should be explicitely written that the kernel function $\mathcal{K}(\cdot, \cdot)$ outputs a matrix whose dimensions correspond respectively to the cardinality of the first and the second input, for it is implicit, but not explicit yet important.

**Questions:**

1. Aren’t those two sentences (« The analogy between Gaussian processes (GPs) and deep artificial neural networks (ANNs) has received a lot of interest, and has shown promise to unbox the blackbox of deep ANNs » (lines 1-3) and the third contribution, « With the best of our knowledge, our work is the first method that performs knowledge distillation between GPs and ANNs. ») contradictory?

2. Can some details be provided concerning the kernel choice $f(\cdot)$?

3. Why choosing Pearson correlation coefficient for comparing GPs and ANNs? Is it what’s often seen in the literature? Why not comparing the coherence of the predictions, for two predictors can have similar prediction function but predict differently most of the time?

**Limitations:**

The authors did not discuss the limitations of their explainability approach. Nowadays, many problems with (for example) saliency maps are well-known (not robust to adversarial perturbations [1], simply unreliable [2], etc.), thus one would expect a few words on those matters.



[1] Ghorbani, Amirata, Abubakar Abid, and James Zou. “Interpretation of neural networks is fragile.” Proceedings of the AAAI Conference on Artificial Intelligence. Vol. 33. 2019.

[2 ]Kindermans, Pieter-Jan, Sara Hooker, Julius Adebayo, Maximilian Alber, Kristof T. Schütt, Sven Dähne, Dumitru Erhan, and Been Kim. “The (un) reliability of saliency methods.” In Explainable AI: Interpreting, Explaining and Visualizing Deep Learning, pp. 267-280. Springer, Cham (2019).

---

> ### Author Rebuttal · Authors · 2023-08-10
>
> Please see the attached global pdf.
>
> \textcolor{blue}{
> The article is too dense.
> }
> \\The paper is dense, because it contains two main ideas. 1. Scalability and 2. Knowledge distillation. Without scalability, successful knowledge distillation wouldn't have happened  (as underlined in the analysis of Fig. S66). So we had to include both scalability and knowledge distillation in one paper.
>
>
> \\\textcolor{blue}{
>  When manipulating concepts such as explainability and interpretability ... This lack of clearness ...
> }\\
> We added the following paragraph to the paper, that describes how the GP can explain the ANN.
> \\\textit{
>     ... After successful distillation, both the ANN and the GP correspond to a single function which is parameterized in two different ways. One is the parameterization by ANN weights, which is not necessarily understandable to humans. The other is the parameterization by GP's posterior which is understandable to humans. ...
> }
> \\As you mentioned, "interpretability" and "explainability" are used to convey different meanings in the literature. But in this paper we use the terms interchangeably. We added the following to the paper
> \\\textit{
> ... In this paper we used the terms interpretability and explainability interchangeably. ...
> }
>
>
> \\\textcolor{blue}{
>   How does that differ from only computing the nearest neighbours in the output space of the network?
> }\\
> If the ANN is a classifier with $C$ classes, in the final layer instances from different classes are supposed to be close to one another. For example all instances of class 0 are near the vector $[1.0, 0.0, ..., 0.0]$,  all instances of class 1 are near the vector $[0.0, 1.0, ..., 0.0]$, ..., all instances of class $C$ are near the vector $[0.0, 0.0, ..., 1.0]$. So the final layer is not a good candidate to compute the similarities. But an interesting question is, what is someone computes the similarity using intermediate layers of the neural network?\\
> We cannot answer all questions in one paper (the paper is already 80 pages long with supplementary), so we implemented GPEX as a tool that researchers can use to answer these questions. In the paper we added the below paragraph about future directions to answer more questions.
> \\\textit{
> ... In this paper we analyzed the effect of number of inducing points, ... One can use the proposed tool to answer other questions, like, is ... . How the similarities provided by GP kernel compares to the similarities obtained from intermediate layers of the ANN? ...
> }
>
>
> \\\textcolor{blue}{
>    Explaining ANNs concerns only page 8 and half of the last page, whereas training a GPs faithfully to a trained ANN really is the main point of the article.
> }\\
> This is because Gaussian process is a white-box model with well-known behaviour. Therefore, the hard part is finding a GP which is equivalent to the ANN and the rest (obtaining explanations by looking at the GP) is straightforward.\\
>
>
> \\\textcolor{blue}{
>    As for the related works, explaining using GPs is only discussed on $\approx$3 lines, citing a single work.This m
> }\\
> Thank you. We extended the related works section by providing more details about SV-DKL [23]. Moreover, we added another method (ref [35] below) to the paper.
> \textit{"
> ... SV-DKL [23] derives a lower-bound for training a GP with a deep kernel. In that method, a grid of inducing points are considered in the kernel-space (like the vectors $\lbrace(\tilde{\akvec{u}}^{(\ell)}_{m}, \tilde{v}^{(\ell)}_m) \rbrace_{m=1}^{M}$ with the notation of this paper).
> Afterwards, each input inst acne is firstly mapped to the kernel-space and the output is computed based on similarities to the grid points in the kernel-space. Since the GP posterior is computed via the grid points, SV-DKL [23] is scalable. But unfortunately the number of grid points cannot be increased to above 1000 even for Cifar10 \cite{ds_cifar10} and with a RTX 3090 GPU. Therefore, this may limit the flexiblity of the GP's posterior [32]. ..."
> }
>
>
> \\\textcolor{blue}{
>    Gaussian processes are highly interpretable. How are they more interpretable?
> }\\
> Gaussian process is definitely a white-box model.
>
>
> \\\textcolor{blue}{
>    Aren’t those two sentences (...) contradictory?
> }
> In summary,
>  - sentence 1: GP-ANN analogy has shown promise before.
>   - sentence 2: we are the first method applying knowledge distillation to achieve GP-ANN anology.
>
> The promise in sentence 1 has been in other ways than knowledge distillation. So there is no contradiction between the two sentences. For example, in neural tangents [20] the neural network is transformed into a GP by putting each and every single layer wide and training on a specific loss (not knowledge distillation).
>
>
> \\\textcolor{blue}{
>    Can some details be provided concerning the kernel choice $f(.)$?
> }
> Thank you. We added the following sentence to the paper to clarify
> \\\textit{
> ... A feature point like $\akvec{x}$ is first mapped to the kernel-space as $\akvec{u}^{(\ell)} = f_{\ell}(\akvec{x})$. Note that the kernel functions $\lbrace f_\ell(.)\rbrace_{\ell=1}^L$ are implemented as separate neural network, or for the sake of efficiency as a single neural network backbone with $L$ different heads.
> Afterwards, the GP's posterior on $\akvec{x}$ depends on the kernel similarities between. ...
> }
>
> \\\textcolor{blue}{
>    The authors did not discuss the limitations of their explainability approach.
> }
> We added the following point to highlight the limitations of the proposed method.
> \\\textit{
> ... . Although the obtained GP-kernel is faithfull globally, the number of nearest neighbours that equally affect the predictions may prevent a human from understanding the expalanations. Moreover, the CAM-like explanations that we obtain might be prone to the known problems of attribution-based methods. ...
> }.

---

> > ### Comment · Reviewer_efq3 · 2023-08-10
> > **Response to rebuttal**
> >
> > -Can some details be provided concerning the kernel choice? / "Gaussian processes are highly interpretable." How are they more interpretable?
> >
> > So, if I understand correctly, the kernel function is a neural network? (More details than the 1-2 lines provided in the rebuttal would be necessary in order to explain exactly what kind of network we are talking about.) In the end, this leads to a Gaussian Process that actually isn't transparent, for a complicated transformation of the input (kernel function) is necessary. It is therefore arguable whether or not a step toward a model that is more "interpretable" is made. I could be misunderstanding, but I feel like the argument here is similar to "the last layer of my black-box neural network is a simple linear layer, so my network is a white-box".
> >
> > -How does that differ from only computing the nearest neighbors in the output space of the network?
> >
> > I actually meant to compute the similarity before applying a hard-max function at the end of the ANN. After what would probably be a soft-max function. Example with 3 classes: data points with outputs [0.1, 0.3, 0.6] and [0.09, 0.28, 0.63] for the ANN should be quite similar. How does proceeding like that is less interesting than computing the similarity in the kernel space? Considering that the GP, in the end, approximates the ANN, and especially since the kernel function actually also is a neural network (as you mentioned in the rebuttal).
> >
> > This is joint with the following: Concerning the "future work avenue" that is mentioned in the rebuttal ("But an interesting question is, what is someone computes the similarity using intermediate layers of the neural network?"), I don't see it as a future work avenue, but a question that the authors should be discussing, for it justify the use of the GP for obtaining explanations. Indeed, if the answer to that question is "computing the similarity using intermediate layers of the neural network works just fine", then the relevance of using the GP could be called into question.
> >
> > -The article is too dense.
> >
> > I understand that the scope of the article requires many information and details, but this is not a sufficient justification as to why important information (see Weakness - Major - 1) is in the supplementary material.

---

> > > ### Author Response · Authors · 2023-08-12
> > >
> > > We hope the following points cover the questions
> > > - GP can ultimately unbox neural networks, as underlined in ref [37]:
> > > "We invite everyone to ... Neural Tangents and help us open the black box of deep learning. "[37]
> > > - GP is a white-box model. This fact has motivated 10 years of research on GP-ANN analogy ([19],[7],[5],[20]).
> > > - Regarding the use of similarities in the output space of neural network, let's say we compute the representations
> > > of a test point $x_{test}$ and training instances in the output space. One CANNOT say $x_{test}$ is classified as such because
> > > it is similar to some training instances in that space, because we don't know if neural network exactly works that way.
> > > But for GP "we know" it uses the similarity function (the kernel function) to make predictions.
> > > So, unlike the NN's output space, from GP it is completely accurate to say that "$x_{test}$ is classified as such because it is similar to $x_{i1}$, ..., $x_{iM}$ in the kernel space".
> > > - Regarding the GP's kernel being a neural network itself:
> > > Although the kernel is a neural network, with the GP it is still completely accurate to say that "$x_{test}$ is classified as such because it is similar to some training instances in that space".
> > > The question comes down to understanding, why the model thinks $x_{test}$ is similar to a training instance.
> > > To answer the latter question, 1. We tailored the CAM idea to our kernel modules which highlight the regions
> > > which have contributed the most to the similarities. 2. In some cases the similarities are highly interpretable to human, even without using the CAM idea.
> > > To see such examples, please refer to Fig. 2. (in particular rows 1 to 5 of Fig. 2-a) or more examples in the supplementary.
> > >
> > >
> > > [37] https://ai.googleblog.com/2020/03/fast-and-easy-infinitely-wide-networks.html
> > >
> > > [5] A. G. de G. Matthews, J. Hron, M. Rowland, R. E. Turner, and Z. Ghahramani. Gaussian process behaviour
> > > 362 in wide deep neural networks. In International Conference on Learning Representations, 2018.
> > >
> > > [7] Y. Gal and Z. Ghahramani. Dropout as a bayesian approximation: Representing model uncertainty in deep
> > > 366 learning. In international conference on machine learning, pages 1050–1059. PMLR, 2016
> > >
> > > [19] R. Neal. Bayesian Learning for Neural Networks. Lecture Notes in Statistics. Springer New York, 2012.
> > >
> > > [20] R. Novak, L. Xiao, J. Hron, J. Lee, A. Alemi, J. Sohl-dickstein, and S. Schoenholz. Neural tangents: Fast
> > > 390 and easy infinite neural networks in python. 2020.

---

> > > > ### Comment · Reviewer_efq3 · 2023-08-14
> > > >
> > > > Thank you for all of the insightful answers. I like the third point that has been raised in the previous answer ("Regarding the use of similarities...").  Nevertheless, overall, no comment has been made on the following: the biological dataset, which is mentioned in the abstract and the introduction, but not discussed elsewhere in the manuscript; the scalability aspect of the approach, which is presented as a contribution, but is not empirically demonstrated or discussed a posteriori. When it comes to the fact that too many things that are necessary to understand the article as a whole are being put in Supplementary material (e.g. CAM-like explanations, ELBO bound), I don't see how the current form of the manuscript, considering 1. the new experiments that have been made for the rebuttal, 2. the explanations not only on the form of the kernel function (really small ANN? as big as the ANN we want to approximate?) but also how this new network is trained and 3. the discussion concerning the limitations of the approach, could deal with this problem.
> > > >
> > > > I feel like the statement "GP can ultimately unbox neural networks" is bold, for it might have $\underline{\textup{helped}}$ doing so, or given $\underline{\textup{insights}}$ on how networks (theoretically) behave, but not more. I disagree with the statement that "GP is a white-box model", for it definitely depends on the form of the similarity function (which here, is a black-box whose ultimate form hasn't been discussed), just like all decision trees and GLMs aren't white-box.
> > > >
> > > > The manuscript contains interesting ideas. Following our discussions, I will raise my score from 3 to 4. I feel like the core ideas have potential, but I think a rewriting (in light of our discussion and the points all of the reviewers have raised) is necessary. Also, submitting to a conference/journal where the tolerated length of the main paper is longer might be well-suited for this work.

---

> > > > > ### Author Response · Authors · 2023-08-15
> > > > >
> > > > > We **strongly disagree** about the comment "GPs and decision trees are not white-box".
> > > > > - According to the Wikipedia page on decision trees [38]: "... have several advantages. Decision trees: ... Use a white box model.".
> > > > > - Decision tree has been used to interpret CNNs [40].
> > > > > - GP4ML book [39] in the preface highlights that
> > > > > 1.  "Gaussian processes provide a principled, practical, probabilistic approach to learning in kernel machines. This gives advantages with respect to the interpretation of model predictions and..."
> > > > > 2. "... neural network algorithms have been used extensively as black-box function approximators in machine learning, but to many statisticians they are less than satisfactory, because of the difficulties in interpreting such models ..".
> > > > > 3. "... Under the Gaussian process viewpoint, the models may be easier to handle and interpret than their conventional counterparts, such as e.g. neural networks. ...".
> > > > >
> > > > > A human observer may fail to track what's happening due to, e.g. the large depth of the decision tree or the GP's kernel space being too crowded (as underlined by reviewer ycEM "human can only visually inspect a small number of examples"). But the underlying prediction mechanism is still known.
> > > > >
> > > > > **In the paper we can can change the wording to "the GP explanations may contribute to understanding the predictions", if the wording is the issue.**
> > > > >
> > > > > **The Wikipedia page [38] and the provided references [40,38,39,37] cannot be all wrong. If you are fair reviewer, please do not deduct score for this matter and increase your score accordingly. Thank you.**
> > > > >
> > > > >
> > > > >
> > > > > [38] https://en.wikipedia.org/wiki/Decision_tree
> > > > >
> > > > > [39] C. E. Rasmussen & C. K. I. Williams, Gaussian Processes for Machine Learning, the MIT Press, 2006, ISBN 026218253X. ©c 2006 Massachusetts Institute of Technology. www.GaussianProcess.org/gpml
> > > > >
> > > > > [40] Zhang, Quanshi, et al. "Interpreting cnns via decision trees." Proceedings of the IEEE/CVF conference on computer vision and pattern recognition. 2019.
> > > > >
> > > > > [37] https://ai.googleblog.com/2020/03/fast-and-easy-infinitely-wide-networks.html
> > > > >
> > > > > [5] A. G. de G. Matthews, J. Hron, M. Rowland, R. E. Turner, and Z. Ghahramani. Gaussian process behaviour 362 in wide deep neural networks. In International Conference on Learning Representations, 2018.
> > > > >
> > > > > [7] Y. Gal and Z. Ghahramani. Dropout as a bayesian approximation: Representing model uncertainty in deep 366 learning. In international conference on machine learning, pages 1050–1059. PMLR, 2016
> > > > >
> > > > > [19] R. Neal. Bayesian Learning for Neural Networks. Lecture Notes in Statistics. Springer New York, 2012.
> > > > >
> > > > > [20] R. Novak, L. Xiao, J. Hron, J. Lee, A. Alemi, J. Sohl-dickstein, and S. Schoenholz. Neural tangents: Fast 390 and easy infinite neural networks in python. 2020.

---

> > > > > > ### Comment · Reviewer_efq3 · 2023-08-15
> > > > > >
> > > > > > My comment actually was that "not $\underline{\textup{all}}$ GPs and decision trees are white-box". I did not imply that decision trees couldn't be used to explain the decision process of a neural network; they can be used, as long as they manipulate concepts familiar to human beings and that their form remains quite simple, just as in [40]. My comment and "... Under the Gaussian process viewpoint, the models $\underline{\textup{may}}$ be easier to handle and interpret than their conventional counterparts, such as e.g. neural networks. ..." [39] actually aren't contradictory.
> > > > > >
> > > > > > Our discussion so far concerning the pros of GPs made me raise the score from 3 to 4; regardless of the kernel that it used, I agree that research on GP-ANN analogy is promising. I feel like my actual rating (4: Borderline reject) accurately reflects what I shared in my previous comment (some of the points have been raised in my original review and haven't been discussed so far):
> > > > > >
> > > > > > "[N]o comment has been made on the following: the biological dataset, which is mentioned in the abstract and the introduction, but not discussed elsewhere in the manuscript; the scalability aspect of the approach, which is presented as a contribution, but is not empirically demonstrated or discussed a posteriori. When it comes to the fact that too many things that are necessary to understand the article as a whole are being put in Supplementary material (e.g. CAM-like explanations, ELBO bound), I don't see how the current form of the manuscript, considering 1. the new experiments that have been made for the rebuttal, 2. the explanations not only on the form of the kernel function (really small ANN? as big as the ANN we want to approximate?) but also how this new network is trained and 3. the discussion concerning the limitations of the approach, could deal with this problem. [...] I feel like the core ideas have potential, but I think a rewriting (in light of our discussion and the points all of the reviewers have raised) is necessary."
> > > > > >
> > > > > > The first two points (biological dataset, scalability) are in themselves problematic, whereas the third point (the form of the manuscript) simply makes me feel too uncomfortable to lean toward acceptance.
> > > > > >
> > > > > > [39] C. E. Rasmussen & C. K. I. Williams, Gaussian Processes for Machine Learning, the MIT Press, 2006, ISBN 026218253X. ©c 2006 Massachusetts Institute of Technology. www.GaussianProcess.org/gpml
> > > > > >
> > > > > > [40] Zhang, Quanshi, et al. "Interpreting cnns via decision trees." Proceedings of the IEEE/CVF conference on computer vision and pattern recognition. 2019.

---

> > > > > > > ### Author Response · Authors · 2023-08-20
> > > > > > >
> > > > > > > We first thank you for being a committed reviewer and actively participating in the discussion with authors.
> > > > > > >
> > > > > > > ==="I agree that research on GP-ANN analogy is promising."
> > > > > > >
> > > > > > > The google-blog post [37] mentions that: “... We invite everyone to explore **the infinite-width versions of their models** …, and help us open the black box of deep learning. …” Indeed, they [37] invite  researchers to implement an infinitely-wide version of any architecture separately, which is too much to ask. Our contribution is that we lift this stringent requirement to unbox neural networks. **The rest (the paper being too dense, the format, not describing the biological dataset, etc.) are minor points compared to our contribution.** We believe we demonstrated the proof of concept of our method, and hope you will give a chance to this manuscript to be shared in our community.
> > > > > > >
> > > > > > >
> > > > > > > === "I think a rewriting … is necessary."
> > > > > > >
> > > > > > > We have dedicated Sec. 3 to explain the algorithm step-by-step. Following the algorithm requires knowing all details about learning GPs as well as details about backpropagation. These days not many people have the background knowledge about **both** GPs and neural networks, **but we cannot reiterate all the prerequisites (like GP training, backpropagation, etc.) in the paper and have to presume that the reader is already fluent in both topics.**
> > > > > > >
> > > > > > >
> > > > > > > === "The scalability aspect of the approach, which is presented as a contribution, but is not empirically demonstrated or discussed a posteriori."
> > > > > > >
> > > > > > > - We have dedicated Sec. 3 to explain the algorithm step-by-step. Following the algorithm requires knowing all details about learning GPs as well as details about backpropagation. These days not many people have the background knowledge about **both** GPs and neural networks, **but we cannot reiterate the prerequisites (like GP training, backpropagation, etc.) in the paper and have to presume that the reader is already fluent in both topics.**
> > > > > > > - The scalability is indeed demonstrated experimentally by increasing the number of inducing points to O(100K), which is impressive in GP literature. For example, in the sample notebook of GPytorch [25] which is available in [41], the number of inducing points can be increased to 900 (30x30 grid)  for Resnet18 classifier, even with Cifar10, batchsize=1, and using RTX3090 GPU. But we can go up to above 100K as opposed to 900.  As another example of the limitations of GPs, the work published last year in NeurIPS by Cohen et al [4] applies GPs to datasets containing roughly 100 dimensional features (like the UCI dataset) with pre-designed kernels.
> > > > > > > **All in all, this explains why we haven’t compared GPEX’s scalability to other methods, because previous methods simply cannot scale to O(100K) inducing points on, e.g., image datasets and with deep kernels.**
> > > > > > >
> > > > > > > === "I don't see how the current form of the manuscript,  (really small ANN? as big as the ANN we want to approximate?) …"
> > > > > > >
> > > > > > > - Our algorithm uses mini-batches of training/inducing instances, so if one can load two instances of the bigger ANN into memory and do the forward-backward passes without  issues, one can definitely apply our GPEX method to this bigger ANN as well. Two instances of the ANN have to fit into GPU memory: one for the ANN and one (potentially) for the kernel of the GP.
> > > > > > > - ResNet18 that we used in the experiments is already a big architecture and is widely used. For example, resnet18 has more parameters than densenet-121 and densenet-169.
> > > > > > >
> > > > > > > ==="the biological dataset, which is mentioned in the abstract and the introduction, but not discussed elsewhere in the manuscript"
> > > > > > >
> > > > > > > - That dataset is used in the sample notebook of scArches [16], which is available online [28]. In our paper we have cited both [16] and [28] so the reader can refer to it. For the other four datasets we did exactly  the same thing (citing the references instead of introducing the dataset ourselves), because the biological dataset isn’t an exception.
> > > > > > > -  Indeed as already mentioned, this manuscript is already dense and its main focus is the proposal of a novel approach to explore more easily so we didn’t include details about the datasets.
> > > > > > >
> > > > > > >
> > > > > > > [4] M. K. Cohen, et al. Log-linear-time gaussian processes using binary tree kernels. NeurIPS 2022.
> > > > > > >
> > > > > > >
> > > > > > > [41] https : / / docs . gpytorch . ai / en / latest / examples/06_PyTorch_NN_Integration_DKL/ Deep _ Kernel _ Learning _ DenseNet _ CIFAR _ Tutorial.html, Accessed Aug 18. 2023.
> > > > > > >
> > > > > > > [25] J. Gardner et al. Gpytorch: Blackbox matrix-matrix gaussian process inference with gpu acceleration. 401 Advances in Neural Information Processing Systems, 2018-December:7576–7586, 2018.
> > > > > > >
> > > > > > > [16] M. Lotfollahi,  et al. Mapping single-cell data to reference atlases by transfer learning. 384 Nature Biotechnology, pages 1–10, 2021.
> > > > > > >
> > > > > > > [28] scArches CVAE notebook. https://docs.scarches.org/en/latest/expimap_surgery_pipeline_basic.html
> > > > > > >
> > > > > > > [37] https://ai.googleblog.com/2020/03/fast-and-easy-infinitely-wide-networks.html

---

> > > > > > > > ### Comment · Reviewer_efq3 · 2023-08-20
> > > > > > > >
> > > > > > > > The discussion concerning the biological dataset and the scalability aspect of the approach makes me satisfied. In this sense, because 2 of the 3 remaining concerns I had were resolved, I'm willing to raise my score from 4 to 5 (borderline accept).
> > > > > > > >
> > > > > > > > What remains is the form of the article, which is such that much information currently misses in the main article. Here is a non-exhaustive list of things that should be discussed in the main article:
> > > > > > > >
> > > > > > > > -The way the CAM-like explanations (line 282) are computed.
> > > > > > > >
> > > > > > > > -The derived ELBO, which is stated as a contribution.
> > > > > > > >
> > > > > > > > -The new set of experiments that have been made for the rebuttal, at least a discussion of it.
> > > > > > > >
> > > > > > > > -The explanations not only on the form of the kernel function, but also how this new network is trained.
> > > > > > > >
> > > > > > > > -The discussion concerning the limitations of the approach.
> > > > > > > >
> > > > > > > > -Sentences supporting claims such as « Because according to theoretical results on GP-ANN analogy, the second last layer of ANN should be wide. » (lines 207-208) or « Our method scales very well, and Alg. 2 runs fine even on imagenet with more than 1M inducing points » (lines 343-344); what’s « fine » mean?
> > > > > > > >
> > > > > > > > -The many points that have been clarified, not only by our discussion but the points raised by the other reviewers as well.
> > > > > > > >
> > > > > > > > Those (many) points don't concern the "[reiteration of] all the prerequisites (like GP training, backpropagation, etc.) in the paper", but what supports and explains the contribution stated at the beginning of the article. The magnitude of the contributions or the relevance of the work is not a good reason not to take into account what actually misses from the paper. I restate that the Supplementary Material is a place for material that is supplementary; not necessary. These details concern not only the methods and the results in themselves but actually why the work is relevant. For it to have an impact on the community and in order to ensure the sustainability of the quality of the article presented in NeurIPS, this shouldn't be seen as a "minor point compared to the contribution".

---

> > > > > > > > > ### Author Response · Authors · 2023-08-21
> > > > > > > > >
> > > > > > > > > We again appreciate your efforts to improve our paper, and assure you that your points will be considered in the final version. In particular, the page limit of the final version "may" be extended to 10 pages (maybe like previous year) which gives us even more space to accommodate the comments.

---

### Official Review · Reviewer_6BV2 · 2023-06-29

**Soundness:** 2 fair
**Presentation:** 2 fair
**Contribution:** 3 good
**Rating:** 5
**Confidence:** 3

**Summary:**

This paper derive an evidence lower-bound that encourages GP's posterior to match ANN's output without any requirement on ANN. And the uses the GPs' kernel functions to explain the ANNs' decisions.

**Strengths:**

1) this paper provides a theoretical way for the algorithm

2) implementation is publically available

3) it is good to see that GP output matches ANN well

**Weaknesses:**

1) The main point of this paper is to explain ANN. However, Figure 2 is the only experiment results that are about explanation, which only contains some sample explanations. There is no comparison to existing explanation methods and there is not quantitative results about the evaluation performance.



**Questions:**

None

---

> ### Author Rebuttal · Authors · 2023-08-10
>
> Please see the attached global pdf.
>
>
> \textcolor{blue}{
> The main point of this paper is to explain ANN. However, Figure 2 is the only experiment results that are about explanation, which only contains some sample explanations. There is no comparison to existing explanation methods and there is not quantitative results about the evaluation performance.
> }
>
>
> \\In Sec. 4.3 of the original submission, we compare against reprenter point selection[33] both quantitatively and qualitatively. Moreover, in the global pdf we added comparison to another method called influence function [36].

---

> > ### Comment · Reviewer_6BV2 · 2023-08-12
> > **thanks for your rebuttal**
> >
> > Given the new results, I'd like to increase my score to 5.

---

### Official Review · Reviewer_nub9 · 2023-07-02

**Soundness:** 2 fair
**Presentation:** 3 good
**Contribution:** 2 fair
**Rating:** 5
**Confidence:** 4

**Summary:**

The paper derives an ELBO to end-to-end distill a deep neural
network into a set of Gaussian processes (one per output) with deep neural
network kernels and the full training set used as inducing points.
By using a low output-dimensionality for the feature map neural network,
the computational cost of the inversion/decomposition of the kernel matrix
is kept constant while arbitrarily scaling the number of inducing points.
The paper empirically validates the discrepency between the teacher neural
network and the student DNN-GP on 5 datasets with 2 different models (5 on
ResNet-18, 2 on some custom attention-based architecture).
Two approaches are introduced to attempt to explain the prediction decisions of
the distilled neural networks based on the DNN-GP: through similarity in the
DNN-kernel's feature space, and a pixel-wise contribution based on the
similarity to the nearest neighbors in the DNN's kernel space.
An accompanying framework is provided, which claims to distill any DNN in Pytorch.

**Strengths:**

- The ELBO and distillation process from GP to DNN-GP is a novel contribution.

- Code is provided for all experiments, which is a major strength.

- Important derivations are provided in the supplement.

**Weaknesses:**

- The paper reports on GPs, but is not very verbose in the fact that
  DNN-kernels are used, although they seem to be strictly necessary for the
  distillation (e.g., the low-dimensional, explicit features space). Prior work
  on DNN-GPs is only partly discussed (e.g. [23]).

- The paper claims to introduce a novel approach to scale the number of inducing points.
  However, the computational trick comes down to using a low-dimensional, explicit
  feature space, leading to a low-rank, high-dimensional noise-free kernel
  matrix, which is only useful for low-dimensional explicit feature spaces as
  in DNN-GP.

- The DNN-kernels are significantly more complex (resnet50) than the teacher
  DNN (ResNet18). This explainability-complexity trade-off is not explicitly
  discussed, and seems like a significant limitation.

- The experiments are not presented very well: Figure 2 (in particular a,c,d) is
  quite hard to understand without jumping to the respective text passage.
  Figures descriptions in general are lacking in detail (i.e., which are
  nearest neighbours, and which is the analyzed sample in Figure 2).
  Fig 2(b) has very small captions.

- In the experiment in Figure 3: while the classifiers on MNIST/CIFAR10/Kather
  are almost completely above 0.95, a PCC of 0.90 and slightly below for the
  other datasets and models are not very insignificant, especially given that
  these models are distilled in order to be explained. This should be discussed
  in more detail.

- The qualitative analysis of the explanations, both based on the
  kernel-similarities, which show similarities and a CAM-based approach, is
  difficult to interprete. While the samples generally seem to show other
  samples from the same class somehow semantically related, these results are
  not directly compared to any approach, making it hard to argue in favor of
  supporting the explainability claim. The focus should be more on the
  experiment in Fig 2b, ideally with more trials and different data sets.

- The explainability approach is only compared to a single baseline, which is
  too few.

- While the experiment in Fig 2b presents a quantitative analysis of the
  explainability of the method compared to one other baseline, it is only
  conducted a single time, on only a single data set. Since the paper claims
  explainability of the models, there should be more experiments in order to
  support this claim.

- While the paper claims to provide a tool, the linked software framework has
  close to no docstrings, provides no documention, features no tests, and does
  not provide a plug-and-play package (no setuptools).
  It is good practice to publish accompanying code, and the paper goes further
  by also including code for the corresponding experimental results.
  However, at the reviewed state, the *tool* simply presents code to the
  paper, rather than providing the software framework suggested in the
  introduction.

**Questions:**

- The paper could be improved by being more verbose about using DNN-kernels.

- It might be better for the paper to instead of stretching the
  scalability-claim, openly discuss how a small, explicit feature space leads
  to a constant rank of the noise-free kernel matrix, creating a situation
  where GPs do not scale beyond the explicit feature space's size, thus arguing
  for low-dimensional DNN-kernel-spaces.

- Given the increased complexity of the kernel's DNN to the teacher model, a
  discussion on the complexity-explainability trade-off would be very interesting.
  Further experiments with smaller kernel DNN's in the distillation-process
  could also bring insight in this regard.

- For the experiments in Figure 3, comparing to some baseline might help in
  understanding the PCC values better. While a PCC of 0.9 sounds good, it would
  help to also highlight the differences between student and teacher, although
  I do not have a concrete idea how this could be done.

- To put the gained explainability into context, especially in the qualitative
  experiments, it would really help to compare to some baselines.

- The quantitative experiment comparing to Representer-Point selection is very
  valuable, but would be even more valuable with multiple tries, and on
  different data sets. It feels somewhat incomplete.

- The software claim in the beginning of the paper suggests a well-maintained
  software framework for the GP model distillation, which does not seem to be true.
  I would suggest to soften this claim (i.e., simply provide a footnote stating
  "Code for methods/experiments provided at...")


### Minor

- Alg. 2 title is misleading, a better name may be: training loop for GP
  distillation.

- Alg. 1 can be moved to the supplement, as it is only a step of
  gradient descent for the DNN-kernels and does not add much to the manuscript

- l.126 nominator -> numerator

- figures should be described more in detail, especially the qualitative one

**Limitations:**

The paper discusses some limitations.

---

> ### Author Rebuttal · Authors · 2023-08-10
>
> Please see the attached global pdf.
>
>
>
> \textcolor{blue}{
> The paper reports on GPs, but is not very verbose in the fact that DNN-kernels are used, ....
> }
> \\In Sec. 2.2. we added the following sentence
> \\\textit{
> .Note that the kernel functions $\lbrace f_\ell(.)\rbrace_{\ell=1}^L$ are implemented as separate neural network, or for the sake of efficiency as a single neural network backbone with $L$ different heads.
>
> }
>
>
> \\\\\textcolor{blue}{
> Prior work on DNN-GPs is only partly discussed (e.g. [23]).
> }
> \\In the related work section, we added the following details about the SV-DKL method
> \\\textit{"
> ... SV-DKL [23] derives a lower-bound for training a GP with a deep kernel. In that method, a grid of inducing points are considered in the kernel-space (like the vectors $\lbrace(\tilde{\akvec{u}}^{(\ell)}_{m}, \tilde{v}^{(\ell)}_m) \rbrace_{m=1}^{M}$ with the notation of this paper).
> Afterwards, each input inst acne is firstly mapped to the kernel-space and the output is computed based on similarities to the grid points in the kernel-space.  ..."
> }
> \\\\Besides SV-DKL, in the paper we have cited and discussed GPytorch [25], KISGP [31], and binary-tree kernels [4]. We also added the work of Wilson et. al. (ref. [35] at the end of this document) which actually is based on KISS-GP [31].
>
>
> \\\\\textcolor{blue}{
> The paper claims to introduce a novel approach to scale the number of inducing points. However, the computational trick comes down to ...
> }
> \\\begin{itemize}
>     \item The core idea of our method is storing the kernel-space representaitons in the matrix $\mathbf{U}$ and involving only one row of $\mathbf{U}$ in the computation graph (as done in line 6 of Alg. S1). This way of handling GP kernel is completely novel.
>     \item Also doing knowledge-distillation between ANN and GP is novel.
>     \item Although the low-rank approximation is a simple idea, it hasn't been applied to GP training before.
> \end{itemize}
>
>
> \textcolor{blue}{
> The DNN-kernels are significantly more complex (resnet50)....
> }
> \\It might be the case that the GP kernel has to have more parameters than the ANN, but this is not a limitation at all. Instead, it is a question to be answered.
> Please note that we cannot answer all questions in one paper, and that's why we have implemented GPEX as a publicly available tool with a simple API .
> \\We added the following part to the paper.
> \\\textit{
> ... In this paper we analyzed the effect of number of inducing points, .... One can use the proposed tool to answer other questions, like, is the GP kernel required to have more parameters than the ANN itself, as observed in the experiments of this paper? Is the uncertainty provided by the GP correlated with the understandability of the explanations to humans or the ANN's failures? ...
> }
>
>
> \\\\\textcolor{blue}{
> The experiments are not presented very well: Figure 2 (in particular a,c,d) is quite hard to understand without jumping to the respective text passage. Figures descriptions in general are lacking in detail (i.e., which are nearest neighbours, and which is the analyzed sample in Figure 2). Fig 2(b) has very small captions.
> }
> \\Thank you. We added caption to all figures.
>
>
> \\\\\textcolor{blue}{
> The explainability approach is only compared to a single baseline, which is too few.
> }
> \\We added another baseline "influence-function". For details please refer to Sec. 1 of this document. Note that unlike attribution-based methods, there are not many similarity-based explanation methods. So we compared against representer point selection [33] and influence function [36].
>
>
> \\\\\textcolor{blue}{
> While the paper claims to provide a tool, the linked software framework has close to no docstrings, provides no documention, features no tests, and does not provide a plug-and-play package (no setuptools). It is good practice to publish accompanying code, and the paper goes further by also including code for the corresponding experimental results. However, at the reviewed state, the tool simply presents code to the paper, rather than providing the software framework suggested in the introduction.
> }
> \\We added documentation via "readthedocs" as well as sample notebooks in the github repository. Please refer to the the anonymous github repo (link available in the submitted paper).
>
>
> \\\\\textcolor{blue}{
> Given the increased complexity of the kernel's DNN to the teacher model, a discussion on the complexity-explainability trade-off would be very interesting. Further experiments with smaller kernel DNN's in the distillation-process could also bring insight in this regard.
> }
> \\Great suggestion. We have done some parameter analysis on the effect of width of the last layer, number of inducing points, and the number of epochs for which the ANN is trained. Unfortunately we cannot answer all questions in one paper (the paper is already 80 pages long including the supplementary). We added this to future directions
> \\\textit{
> ... In this paper we analyzed the effect of number of inducing points, the width of the second last layer, and the number of epochs for which the ANN is trained. One can use the proposed tool to answer other questions, like, is the GP kernel required to have more parameters than the ANN itself, as observed in the experiments of this paper? Is the uncertainty provided by the GP correlated with the understandability of the explanations to humans or the ANN's failures? ...
> }
>
>
> \\\\\textcolor{blue}{
> To put the gained explainability into context, especially in the qualitative experiments, it would really help to compare to some baselines.
> }
> \\In Sec. S6 of the supplementary we have qualitatively compared our explanations to those of representer point selection [33]. Please refer to Sec. S6 for more details.
>
>
> \\\\\textcolor{blue}{
> l.126 nominator -> numerator
> }
> \\corrected.
>
>
> \\\\\textcolor{blue}{
> figures should be described more in detail, especially the qualitative one
> }
> \\Thank you. We added captions to all figures.

---

> > ### Comment · Reviewer_nub9 · 2023-08-21
> > **Thank you for the rebuttal**
> >
> > Thank you for the detailed responses and the new results.
> > The introduction of influence functions make the manuscript a lot stronger, although they look somewhat unstable and may require a few more trials.
> > I am somewhat unsatisfied about the response to the size of the Kernel-DNNs, but acknowledge the restrictive length of the manuscript and hope a sentence/paragraph will be added.
> > Thank you for adding documentation and tutorials to the software.
> >
> > Given the new additions, I feel comfortable to increase my score to 5.

---

### Author Rebuttal · Authors · 2023-08-10

the global response

---

### Author Response · Authors · 2023-08-20

Dear reviewers, ACs, and SACs,

We want to share a concern. This week, a co-author’s github repositories were visited three times by an individual from “openreview.net”. We are concerned that the knowledge of authors’ identities could have created some biased evaluations which would violate blind-review process guidelines. It would be appreciated to know that this scenario did not happen, or if it did happen please reconsider your scores by forgetting the bias.

GP-ANN analogy  is viewed as a high priority by the most prominent AI researchers to guide the future of our field [1] to unbox neural networks, and our work solves many issues of the aforementioned research direction. We believe therefore, that this work should be given the opportunity to be fostered in our research community.

Best regards,

[1] https://ai.googleblog.com/2020/03/fast-and-easy-infinitely-wide-networks.html

---

### Author Response · Authors · 2023-08-21

Dear ACs,
3 out of 4 reviewers accepted the paper, but reviewer "nub9" hasn't replied to our rebuttal as of now.
We would appreciate it if you could please send a reminder to him/her (reviewer "nub9").
Best regards

---

### Decision · Program_Chairs · 2023-09-21

**Decision:**

Accept (poster)

**Comment:**

The results of this paper should be of interest to the NeurIPS community. Having read the reviews, rebuttal and extensive discussion, there are a number novel and technically significant contributions, from distilling an NN to GPs, to interpretability applications and the inclusion of comparisons to influence functions in the rebuttal. The experiments + code contribution will also be a useful resource for the community to build on.